# Leveraging ecosystems responses to enhanced rock weathering in mitigation scenarios

Yann Gaucher [1,2] ✉, Katsumasa Tanaka [1,3], Daniel J. A. Johansson [4], Daniel S. Goll [1] & Philippe Ciais [1]

Carbon dioxide removal (CDR) is deemed necessary to attain the Paris Agreement's climate objectives. While bioenergy with carbon capture and storage (BECCS) has generated substantial attention, sustainability concerns have led to increased examination of alternative strategies, including enhanced rock weathering (EW). We analyse the role of EW under cost-effective mitigation pathways, by including the CDR potential of basalt applications from silicate weathering (geochemical CDR) and enhanced ecosystem growth and carbon storage in response to phosphorus released by basalt (biotic CDR). Using an integrated carbon cycle, climate and energy system model, we show that the application of basalt to forests could triple the level of carbon sequestration induced by EW compared to an application restricted to croplands. EW also reduces the costs of achieving the Paris Agreement targets as well as the reliance on BECCS. Further understanding requires improved knowledge of weathering rates and basalt side-effects through field testing.

Parties to the Paris Agreement committed to keeping global warming well below 2 °C, and to continuing their efforts to aim for 1.5 °C of warming relative to the preindustrial level. Meeting this goal requires reducing emissions at an unprecedented pace to reach carbon neutrality by the middle of the century. Carbon dioxide removal (CDR) is known to be required to compensate for both temporary overshoots of carbon budgets and for residual emissions that may persist after emission reductions from all sectors. The longer the delay in emission reductions, the greater the need for CDR[1].

While bioenergy with carbon capture and storage (BECCS) and afforestation are so far the most widely explored carbon dioxide removal solutions[2], concerns have been raised regarding the technical feasibility and sustainability of BECCS at the scales envisaged in the mitigation scenarios assessed by the IPCC[3]. In particular, the amounts of biomass needed to reach sufficient CDR levels could have high impacts on water, land and nutrient use[4–6]. Sustainable deployment of large-scale CDR thus requires the examination of alternative CDR portfolios[7,8]. Enhanced weathering of basalt (EW) is an emerging and

promising CDR that consists in amending soils with basalt dust[9–12]. Basalt is an abundant volcanic rock containing less harmful trace elements than alternative feedstocks[13]. As basalt erodes, the minerals released react with $CO_2$ and sequester carbon for at least several hundred years[14], a process called 'geochemical CDR'. Current research focuses on basalt for demonstration purposes of EW. Basalt encompasses a wide range of rock material with varying CDR potential and other feedstocks might be more suited. Unlike BECCS, EW does not disrupt existing land use and is usually assumed to be deployed on croplands[1,9–12,15], that are accessible for transporting and spreading basalt. Furthermore, co-benefits of crop yields from amendment with basalt have been studied in previous work, including dedicated experiments, showing improvement of soil quality[13] which could reduce fertilisers use[16,17], plant health, and yields[18–22]. Moreover, EW stimulates biomass production through nutrients released during basalt dust dissolution, thereby increasing carbon sequestration and storage in natural ecosystems. This additional $CO_2$ removal process from EW called 'biotic CDR' has only recently been quantified. Biotic

[1]Laboratoire des Sciences du Climat et de l'Environnement (LSCE), IPSL, CEA/CNRS/UVSQ, Université Paris-Saclay, Gif-sur-Yvette, France. [2]CIRED, Ecole des Ponts, Nogent-sur-Marne, France. [3]Earth System Division, National Institute for Environmental Studies (NIES), Tsukuba, Japan. [4]Department of Energy and Environment, Chalmers University of Technology, Gothenburg, Sweden. ✉e-mail: yann.gaucher@enpc.fr

CDR could potentially double the global CDR of EW[23] (2-5 $GtCO_2$ per year[4,12,15]), yet non-agricultural application is not part of CDR portfolios in existing assessments[1,2,8]. While EW presents co-benefits for soils and ocean pH[10], emissions associated with extracting, crushing and spreading basalt could partly offset the CDR potentials, depending on the underlying energy mix[23,24].

Here we explore how the application of EW on suitable forests and crop fields could affect mitigation pathways using a hard-linked carbon-cycle, climate, and energy system model that considers geochemical and biotic CDR and associated energy requirements within a single framework. We quantified the potential CDR from EW for ambitious climate mitigation pathways and the subsequent reduction in reliance on BECCS. The addition of EW to the CDR portfolio of mitigation technologies could make ambitious climate targets achievable with lower mitigation costs[25]. We thus examined how EW affects mitigation costs, energy consumption, and temperature pathways over the 21st century in four climate target cases: 1.5 °C scenarios with medium overshoot (up to 0.2 °C) and high overshoot (no limit) and 2.0 °C scenarios with no overshoot and with high overshoot (no limit). All three overshoot scenarios achieve the respective temperature target by 2100. In our 1.5 °C medium overshoot scenario, we allow a higher overshoot than what is defined as low overshoot scenarios in IPCC AR6 (up to 0.1 °C), as the latter would lead to very large unrealistic short-term demand reductions in our model[26,27]. By taking advantage of our energy-climate modelling framework, we further highlight key uncertainties that influence the role of EW in climate mitigation.

We developed a version of the partial-equilibrium energy model GET7.1[28,29], which we integrated with the aggregated carbon cycle, atmospheric chemistry, and climate model (ACC2[30,31]). The carbon cycle component of the resulting GET-ACC2 model was coupled to an EW module, in which the dissolution of basalt directly removes atmospheric $CO_2$ and delivers phosphorus to the soil, fertilising forest growth and stimulating additional $CO_2$ sequestration in forest biomass (Fig. 1). We emulate ORCHIDEE-CNP[32], a global biosphere model that resolves the phosphorus cycle, to quantify both geochemical (abiotic) and biotic carbon dioxide removal (CDR) from basalt applied to forest ecosystems. GET-ACC2 is thus a forward-looking model accounting for this biotic CDR pathway of EW. GET-ACC2 quantifies least-cost pathways where low-carbon technologies, CDR, and abatement measures for $CH_4$ and $N_2O$ are deployed to mitigate climate change. The net present value of the social surplus (i.e., the sum of consumer surplus minus the energy costs, discounted at a 5% rate) is maximised with perfect foresight, leading to a preference for late spending, including late abatement.

The spatial heterogeneity of the response of ecosystems to basalt application and the local factors driving additional biological carbon sequestration have been discussed in Goll et al. [23]. The biotic CDR was found to be highly variable across regions, strongly dependent on ecosystem type, and most effective where the natural background phosphorus availability was insufficient for plants to benefit from increasing atmospheric $CO_2$ and warming, notably in tropical and boreal forests[23]. GET-ACC2 prioritises the most responsive areas for basalt application, but this spatial heterogeneity is only implicitly captured in the emulator (see Methods), and that we only accounted for the phosphorus fertilisation effect on plants, not other effects of basalt weathering on soil microbes and soil biota, or other biotic

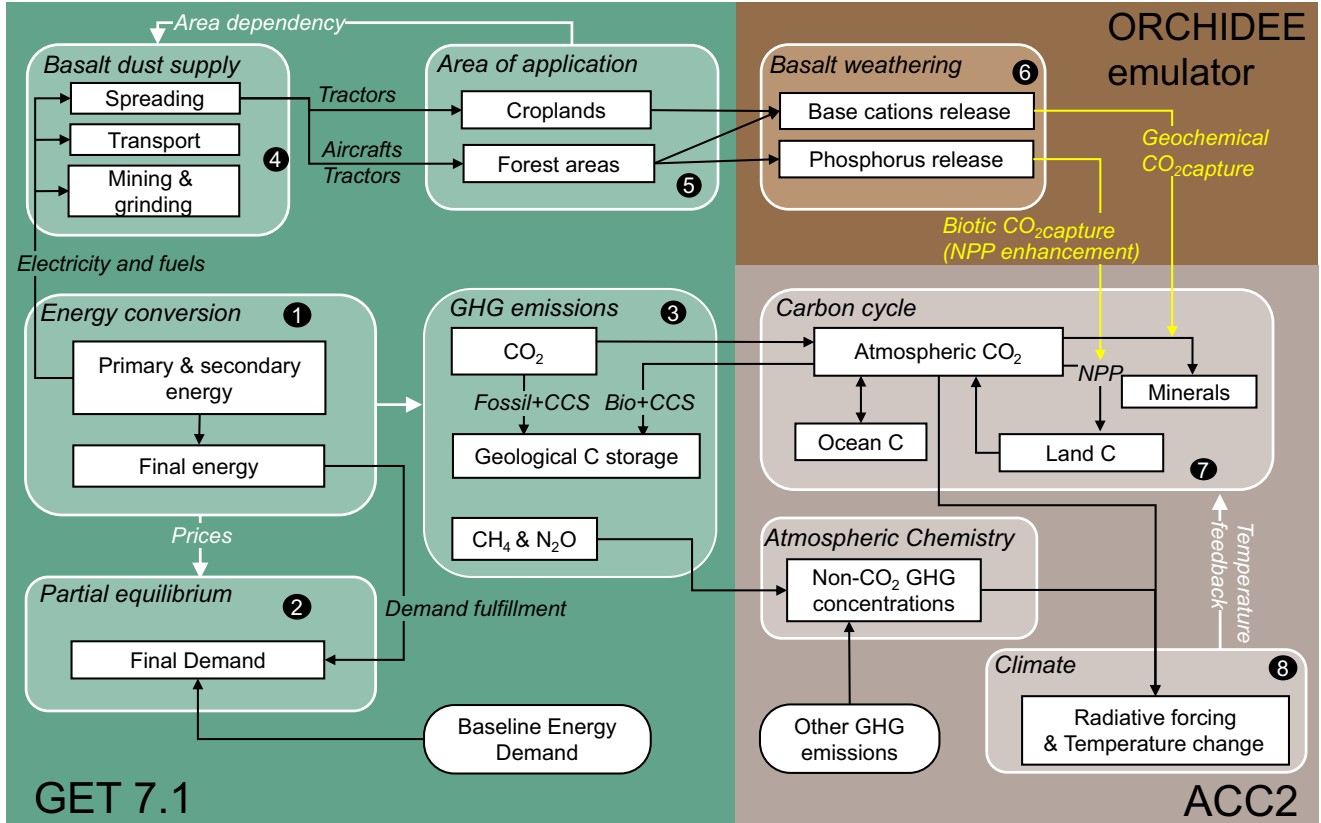

**Fig. 1 | Integrated model of climate, carbon, and energy economics** This diagram highlights the key processes resolved in the model related to bioenergy with carbon capture and storage (BECCS) and enhanced weathering (EW), and their interactions with the energy and climate systems. Round boxes are exogenous projections. Key outputs are: (1) Energy production, mix and associated costs. (2) Price-responsive energy demand (3) Resulting net greenhouse gas (GHG) emissions from the energy sector, including negative emissions from BECCS. (4) Costs and energy requirements of EW. (5) Quantity of basalt applied on croplands or forest areas. (6) Geochemical $CO_2$ capture from basalt weathering. (7) $CO_2$ capture from phosphorus-driven net primary production (NPP) increases (the biotic effect). (8) Global temperature pathway.

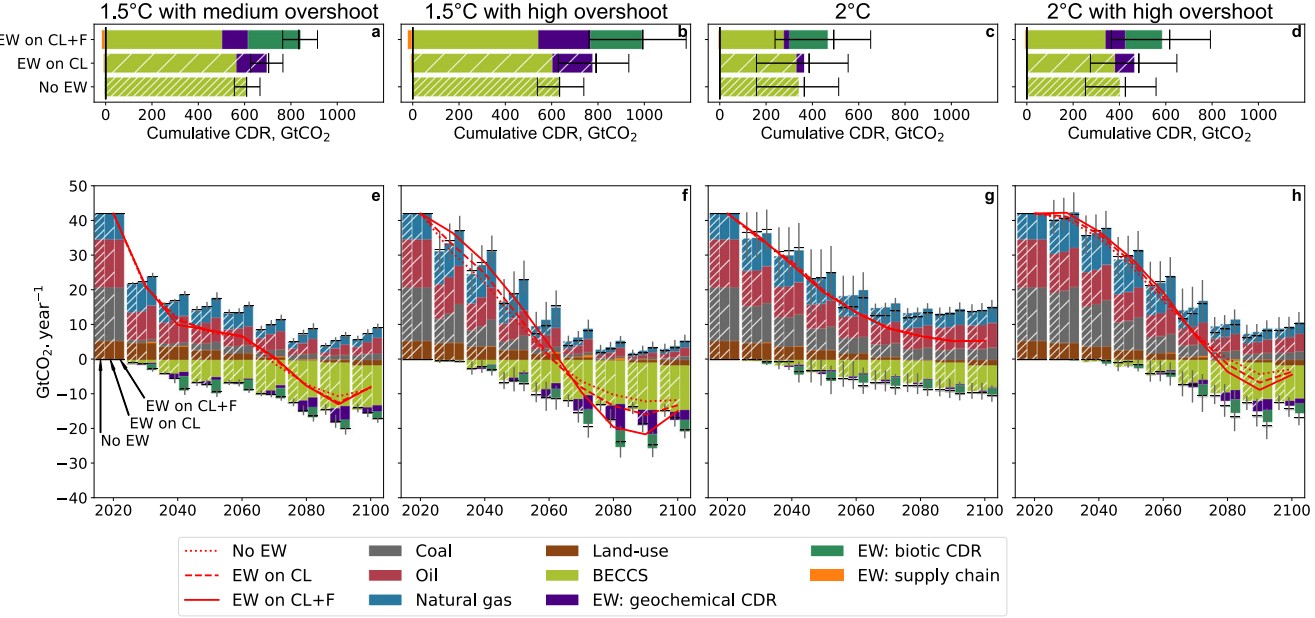

**Fig. 2 | Carbon dioxide emissions from three carbon dioxide removal (CDR) portfolios for different climate targets across the 21st century.** Three CDR portfolios are assessed: No EW: bioenergy with carbon capture and storage (BECCS) only. EW on CL: enhanced weathering (EW) deployed on croplands (CL) only, and BECCS. EW on CL + F: EW deployed on croplands and forest areas (F), and BECCS. The bars indicate the mean, the black dashes the median, and the 25–75% range. Four climate policy targets are compared (one per column): 1.5 °C with medium overshoot (**a, e**): The temperature change is limited to 1.5 °C after 2100, with a possible overshoot of up to 0.2 °C before 2100. 1.5 °C with high overshoot (**b, f**): the temperature change is limited to 1.5 °C after 2100. 2 °C with no overshoot (**c, g**): the temperature change is limited to 2 °C. 2 °C with high overshoot (**d, h**): the temperature change is limited to 2 °C after 2100. **a–d** Displays cumulative CDR over the period 2020–2100, while **e–h** display annual emission and CDR. "EW: supply chain" (in orange) represents the emissions from fossil fuels used to apply EW. The red lines represent the net $CO_2$ emissions.

effects such as interactions with the nitrogen cycle, plant health and resistance to pathogens[33].

## Results

### Enhanced weathering deployment

Under each climate target case, we assessed the following three CDR portfolios: i) BECCS only (No EW), ii) BECCS with EW on croplands only (EW on CL), and iii) BECCS with EW on croplands and forest areas (EW on CL + FA). In the latter, we assumed that basalt could be spread over forest areas, yet with a significant energy and cost penalty compared to croplands. The cost-effective magnitude of CDR from EW deployment is very contrasted across climate target scenarios (Fig. 2): EW is more used to achieve 1.5 °C than to achieve 2 °C, and it is also used more in high-overshoot than in medium and no-overshoot scenarios. EW is applied when the net present value of future carbon removals, occurring in the years following basalt application and extending over decades for biotic sequestration, outweighs application costs. Cropland application costs range from $43 to $132 per ton basalt, increasing at higher application levels due to prioritising accessible fields first, leading to higher transport costs for more remote areas. It corresponds to $116 to $242 per ton of $CO_2$ removed, within the range of existing assessments[12,15]. Possible co-benefits could increase the cost-effectiveness of EW[34] if they can act as fertilisers, but our model does not include food systems and land use and these effects were not considered in the study.

We assume no absolute limit to the production of basalt dust although its growth rate is limited (to 10–20% per year). The maximum CDR potential by EW on croplands thus depends on the application rate of basalt (15 kg/m²), the area of suitable croplands with sufficiently warm and rainy climate, 7.9 Mkm², i.e., a third of global croplands[15], and the weathering rate (1–26% per year). The maximum cropland CDR is 4.9 GtCO₂ per year, consistent with other estimates assuming unlimited basalt supply[12,35]. Here, the EW application only on croplands

approaches its full potential at 1.5 °C with a high overshoot scenario with an annual CDR peak of 4.4 GtCO₂ per year and a cumulative removal of 173 GtCO₂.

Forest application is more expensive, with costs varying from $146 to $364 per ton of basalt, hinging primarily on the expenses associated with airborne application, and on carbon price-sensitive energy costs, constituting 20–40% of the total. However, the phosphorus effect enhances $CO_2$ removal efficiency, resulting in substantially reduced removal costs of $20–$166 per ton $CO_2$, especially at low application levels (see Supplementary Material for a detailed analysis of removal costs). Allowing basalt application in forests reduces the carbon price threshold above which EW becomes cost-efficient and increases the CDR potential in two ways: by expanding the area for basalt application, increasing the geochemical CDR potential, and enabling biotic CDR in forests. As a consequence, the EW-induced CDR is almost tripled when basalt can be applied to forests, with a peak of 12.4 GtCO₂ per year and 446 GtCO₂ cumulatively in the 1.5 °C with high overshoot scenario. The relative contributions of geochemical and biotic removals vary depending on scenarios; for example, in the 1.5 °C with high overshoot scenario, the increase in geochemical CDR due to the additional area available is more pronounced than in other scenarios because cropland application is at full potential. Furthermore, the share of the biotic CDR is proportionally lower at high application levels because the phosphorus stimulation of forest production gradually saturates, thereby limiting the biotic CDR potential. This limit explains why biotic CDR by EW varies less than geochemical CDR by EW among different scenarios.

### Policy costs

EW provides flexibility not only by replacing more expensive mitigation measures but also by allowing abatement to be delayed. This reduces the costs of achieving climate objectives, here quantified as the net present value of policy cost (the energy system costs plus the

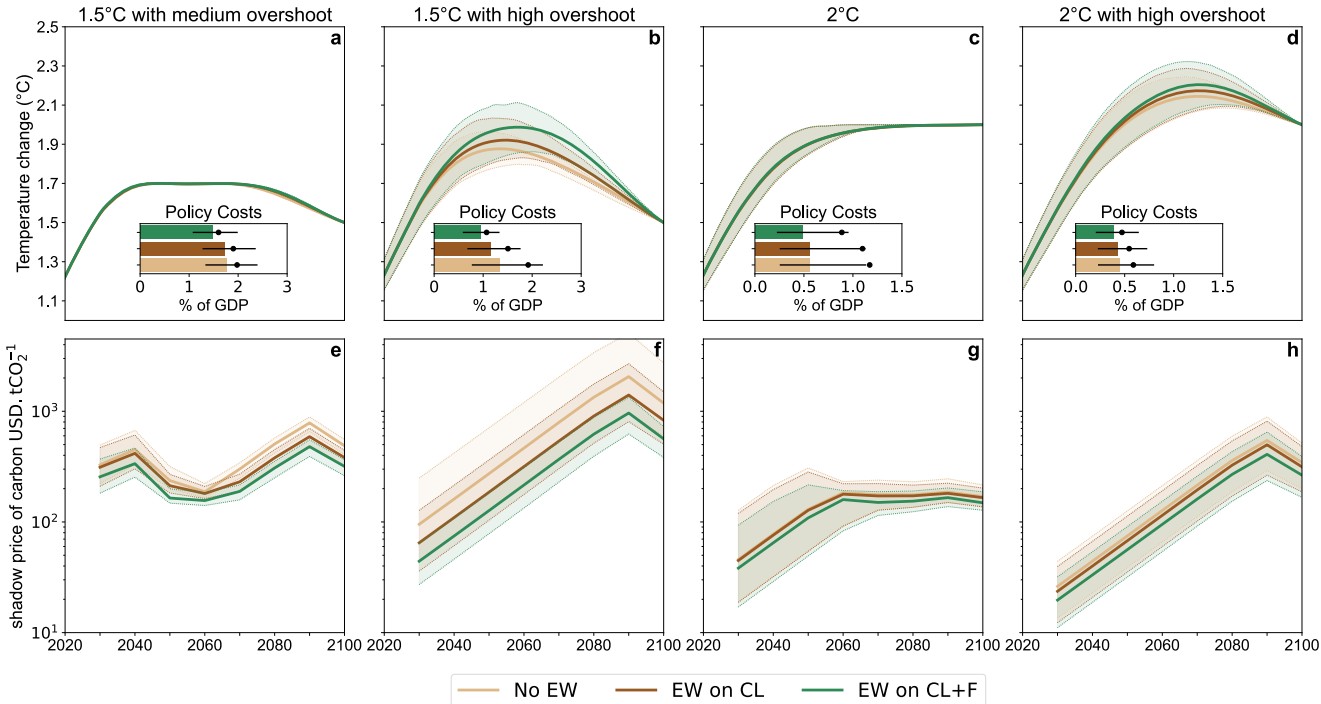

**Fig. 3 | Temperature, policy costs and carbon price pathways across the 21st century. a–d** Global-mean surface temperature change relative to the 1850–1900 mean. Policy costs are the net present values of future energy production costs and consumption losses as a percentage of GDP, compared to the no-policy scenario. The coloured bars indicate the median, the black dot the mean, and the error bars the 25–75% range. Policy costs scale exponentially with the equilibrium climate sensitivity, which can push the mean above the 25–75% range of the sample. **e–h** Median price of carbon across the 21st century for different climate targets (log scale). The shaded area represents the 25–75% range. The vertical scales are different between the panels.

loss of consumer surplus) compared to a baseline without climate policy (Fig. 3, see also Supplementary Fig. 22 for energy demand reduction and Supplementary Fig. 23 for annual costs). Abatement, including EW deployment (Fig. 2), is delayed in overshoot scenarios: in our forward-looking optimisation model, the greater the future opportunity for negative emissions, the lower the near-term abatement and the discounted costs. Therefore, applying EW on croplands and forest areas reduces policy costs most significantly in the high overshoot scenarios, by 44% in the 1.5 °C scenario with high overshoot, against 20% in the 1.5 °C scenario with medium overshoot. In the latter case, the rapid reduction required over the next decade is too early for basalt to be used on a large scale, and relies largely on a severe contraction in the demand. The application of EW on croplands only has a weak impact, in particular in the 2 °C scenario without overshoot where the median cost reduction is zero and the mean reduction is 6%.

The increasing stringency of climate policies across the 21st century is reflected by the endogenous carbon price (Fig. 3). EW reduces it, on average, by 67% if applied on croplands and forests in the 1.5 °C with high overshoot scenario, and by 31% when applied on croplands. When aiming at 2 °C, EW reduces it by 27%. But if application over croplands only is considered, the carbon price is only reduced by 10%. Thus, applying EW on crop fields helps to reduce the efforts required to achieve the most ambitious climate objectives but is not a game-changer when aiming at the 2 °C target. Overall, EW pays off more when aiming at 1.5 °C than when aiming at 2 °C, and more with overshoot than without.

As a consequence of delaying abatement and changing the net emission pathways, EW increases the peak temperature level in high overshoot scenarios: from 1.88 °C without EW to 1.98 °C with EW in the 1.5 °C case, and from 2.13 °C to 2.20 °C in the 2 °C case (Fig. 3). The change in peak warming depends on the assumed discount rate, and is strongly reduced with a lower discount rate of 2% (Supplementary Figs. 20 and 21).

## Reduction of BECCS

Reducing the reliance on BECCS for achieving negative emissions could limit the deployment of bioenergy crops and alleviate the threats they pose to food security, water and nutrient resources and biodiversity[36]. Therefore, it is of interest to analyse if EW and BECCS are complementary or in competition with each other. EW does not directly compete with BECCS for resources: BECCS provides energy while EW uses energy, and EW could be applied on bioenergy crops areas. BECCS are used in all our mitigation scenarios to supply electricity and heat, but also hydrogen (for transportation, and industrial processes) when high levels of negative emissions are required. We found that adding EW to the CDR portfolio increases the total CDR level but reduces the use of BECCS which becomes partially unnecessary (Fig. 2). The reduction in BECCS per tonne of EW-induced CDR is higher in medium or no overshoot scenarios than in high overshoot scenarios, where BECCS and EW rather add up.

By reducing the dependence on bioenergy, the EW could also reduce the pressure on food prices and lower land rents and the market incentive to cultivate food or bioenergy crops in pristine areas. Due to the competition for land[37], food prices are expected to increase with biomass prices[38], which reflect the willingness to pay for bioenergy and increase with carbon prices. Applying EW on croplands and forests reduces biomass prices and cuts the use of bioenergy, but limiting EW to croplands reduces these effects (Fig. 4). In summary, EW only partially replaces BECCS, but reduces the demand for bioenergy.

## Impacts on energy use

EW generally requires smaller energy input than other CDR options, such as direct air carbon capture and storage. The energy use of EW can be divided in three components: i) mining and grinding of basalt (here, a size of 20 μm was assumed[15]) which consumes fuel and electricity, ii) transport to application sites which increases the freight

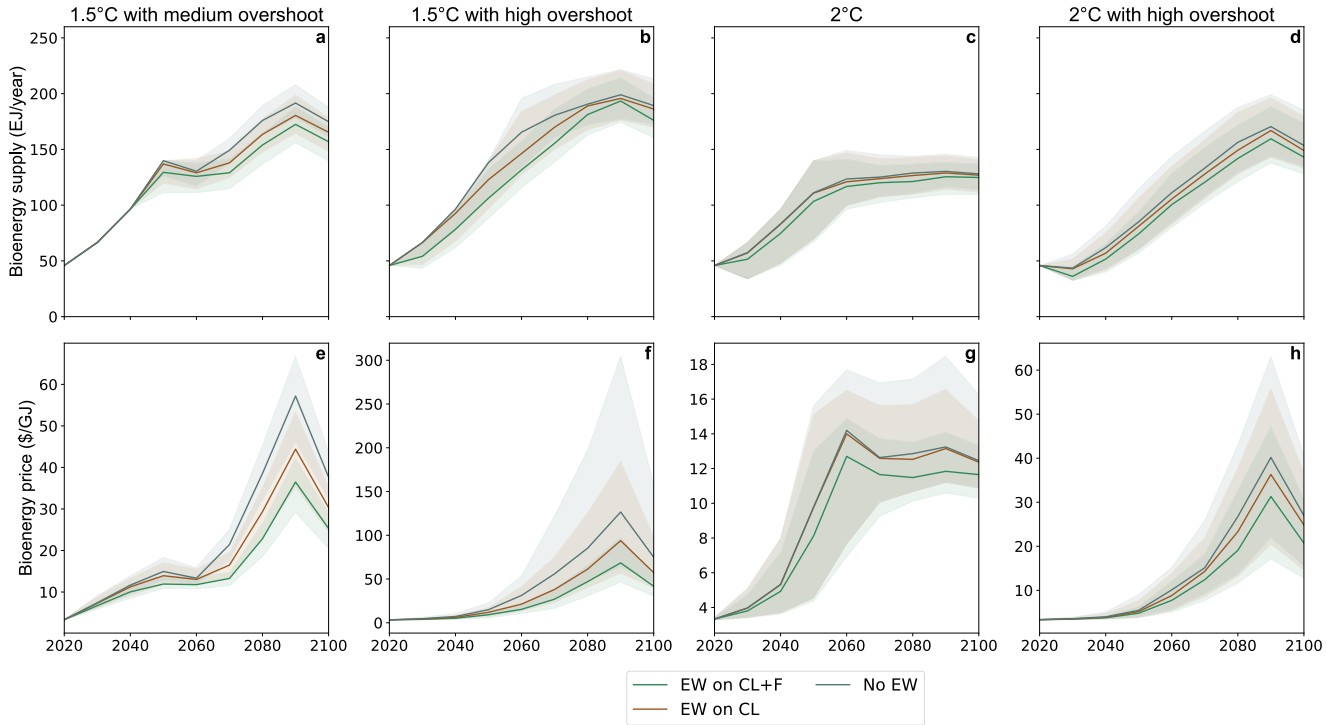

**Fig. 4 | Price and use of bioenergy. a–d** Median primary bioenergy consumption. Top, right y-axis (dotted line): Median primary bioenergy supply in the no enhanced weathering (EW) scenarios. **e–h** Median biomass price. Shaded area: 25–75% range.

demand, and iii) spreading of basalt dust requiring fuel if tractors are used (for crop fields, for instance) and aviation fuel if basalt is spread on forests (see Methods).

The average final energy intensity of EW per ton of $CO_2$ sequestered depends on the level of EW deployment, and thus on the scenario (Fig. 5). In crop fields, the final energy input increases with basalt application as transport distance increases. Therefore, if basalt is only applied on crop fields, the final energy use per $tCO_2$ removed varies between 1.2 $GJ/tCO_2$ in the 2 °C without overshoot where EW is less applied, and 2.2 $GJ/tCO_2$ in the 1.5 °C scenario with high overshoot. In forests, $CO_2$ removal per ton of basalt is higher than in croplands due to the biotic effect but it decreases with increasing application, whereas energy use per ton of basalt varies little because the areas chosen first are those most stimulated by phosphorus, rather than the nearest ones. The application on forest areas, therefore, reduces the energy intensity in the 2 °C scenario without overshoot (0.4 $GJ/tCO_2$), but increases it in the 1.5 °C scenario with high overshoot (2.3 $GJ/tCO_2$) due to the saturation of the biotic effect. For comparison, direct air carbon capture and storage would typically require 4–12.4 $GJ/tCO_2$[39,40] depending on the technology used.

Electricity and aviation fuel are the predominant energy carriers used for EW, which uses 8% of the projected total electricity production, and 23% of the projected aviation fuel until 2100 in the 1.5 °C with high overshoot scenario. The emissions from EW depend on the energy sources used, and thereby on the carbon price. In the medium and no-overshoot scenarios, the share of kerosene among the aviation fuel used for basalt application is higher than in the high overshoot scenarios where higher carbon prices at the time of basalt application on forests lead to a switch to hydrogen. Moreover, since a share of this hydrogen is produced from BECCS, the net basalt supply emissions are negative in the 1.5 °C scenario with high overshoot. Conversely, in the 2 °C scenario, the carbon price is lower, kerosene continues to be used, and the EW-related emissions offset 2.6% of the CDR (Fig. 5c).

## Uncertainty analysis
CDR plays a critical role in mitigation pathways developed by IAMs despite the low technology readiness of the majority of CDR technologies. Thus, uncertain costs and scalability[41] call for an analysis of the impact of related model assumptions on our results. Besides, costs of competing technologies[42] and discount rates[43] have been shown to affect uncertainties in the role of CDR.

To gain insight into uncertainties related to key model parameters, we use the Morris method[44,45]. It quantifies the mean of the variations of an output variable resulting from an increase in the value of a single parameter over a representative sample of all parameter values (see Methods). A positive mean indicates that the output increases with a higher parameter value. This method is applied to analyse the sensitivity of EW and BECCS deployment to key model parameters, as shown in Fig. 6a–d.

The parameters that increase the efficiency of EW tend to reduce the use of BECCS, and vice versa, so that EW and BECCS appear as competing technologies or substitutes. The use of CDR also increases with the costs of other mitigation technologies, such as wind, solar or nuclear energy: a decrease in their costs reduces the carbon price and therefore disincentivises the use of CDR[42]. Similarly, increasing the system flexibility such as the maximal rate at which the installed capacity of technologies can grow, or the price-elasticity of the demand, generally reduces the use of CDR. The use of CDR further depends on the climate uncertainty, as assessed through the ECS[46].

We found that the uncertainty related to the physical processes of EW (weathering rate, geochemical capture rate, and the phosphorus content of basalts) strongly influence the magnitude of EW CDR, it is less sensitive to the uncertainties of parameters surrounding the energy requirements of EW, in particular the electricity use for grinding the rocks.

Weathering rates in soils remain highly uncertain[47–49] (see Methods). The weathering rates used in modelling studies are calibrated on laboratory experiments[9,12,15,48,50]. Field or pot conditions experiments generally provide lower estimates than laboratory experiments[47,51–53],

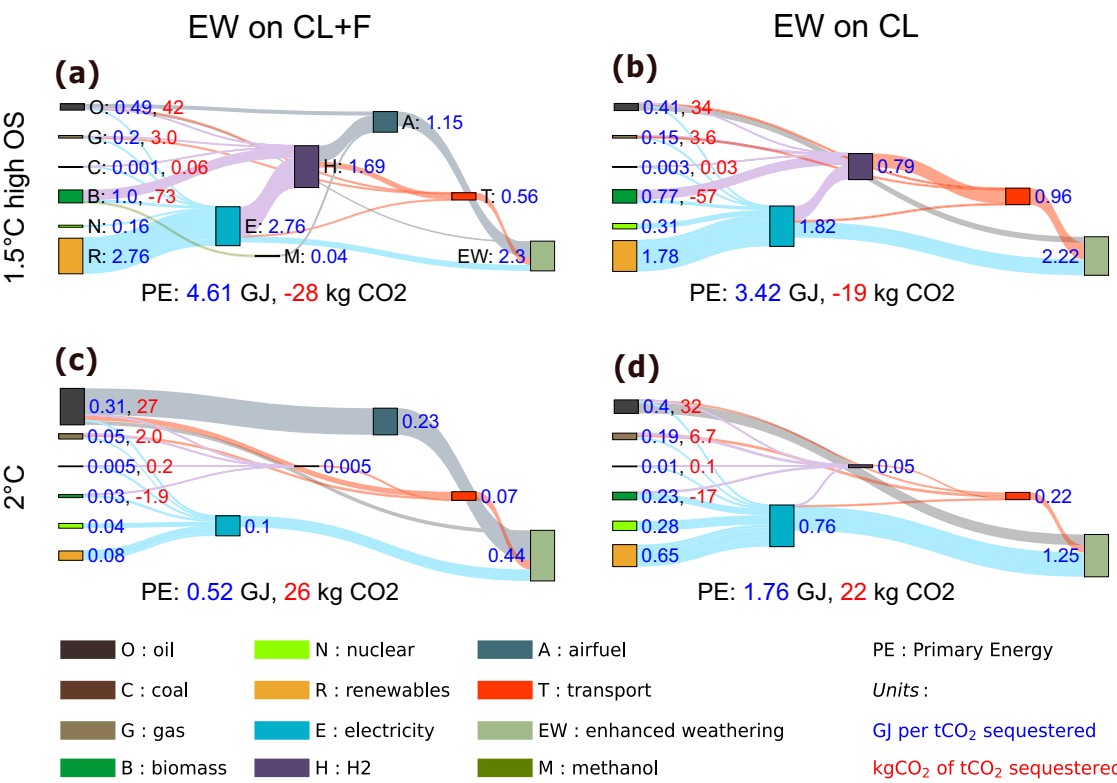

**Fig. 5 | Average energy use and average CO$_2$ emissions associated with the use of enhanced weathering (EW) per tCO$_2$ sequestered (average across the 21st century). a, c** Enhanced weathering is applied on forest areas and croplands. **b, d** Enhanced weathering is applied only on croplands. **a, b** 1.5 °C case with high overshoot. **c, d** 2 °C case without overshoot. Blue labels: The energy used to apply EW, for each energy vector, expressed in GJ per ton of CO$_2$ that is sequestered through EW. Red labels: The emissions associated with the use of each primary energy source, expressed in kg of CO$_2$ emitted per ton of CO$_2$ captured through EW. Note that the vertical scale is different in each panel.

because the weathering rate depends on complex interplay of soil pH, temperature, hydrological conditions, and biological activity[33,53]. Soil column[51] and mesocosm[47] experiments have suggested surface-normalised weathering rates respectively two and three orders of magnitude slower than those used in ref. 15, which defines the high range of the weathering rates used in the present work (25% per year). However, field[18,19,54,55] and forest[56] studies have shown promising CDR rates corresponding to weathering rates exceeding this range. For instance, ref. 18 reports a 16 ± 6% loss of cations from basalt applied on agricultural crops over 4 years. This corresponds to a mean weathering rate of 3–6% per year. As they report a grain size of 267 μm, 20 μm-sized grains could weather around ten times faster, because weathering speed scales with the reactive surface. The wide variations across experiments indicate that the weathering rate is a critical source of uncertainty. As shown in Fig. 6e–l, the lower the weathering rate, the less basalt is applied and the less carbon is captured. For weathering rates below 1% per year, EW can become a viable cost-effective option only if basalt is applied on natural areas, because the supply of even very low quantities of phosphorus to phosphorus-depleted soils yields a significant biotic CDR. Thus, the high efficiency of basalt application over forest is maintained even with low weathering rates (see also the sensitivity results using weathering rates of 1% and 25% in Supplementary Information 1.1.1).

## Discussion

We showed that the CDR potentials of EW under cost-effective mitigation pathways can be larger than previously thought by additionally considering the potentials associated with the phosphorus fertilisation, or 'biotic' effect of EW, while our results align with existing studies on application costs and geochemical removal potential over croplands[3,22–24]. EW neither accelerates climate change mitigation nor

reduces temperature overshoot in our cost-effectiveness analysis, yet its potential for lowering peak temperatures to mitigate near-term climate damage could be further assessed elsewhere through cost-benefit analyses. Deploying EW in addition to BECCS reduces the willingness to pay for biomass and could thereby lower the pressure on land conversion as well as on food prices, although the reliance on bioenergy remains significant.

We further demonstrated that under mitigation pathways, in particular, for the 1.5 °C warming target, the use of EW reduces the total mitigation cost, lowers the peak carbon price, and replaces a larger amount of BECCS when the 'biotic' effect is included, even if we account for the high costs for EW application over forest areas by aeroplanes. These findings are robust under a range of uncertainties considered, unless weathering rates are in the 0–1%/year range. Such benefits of EW were found to be more pronounced under pathways with high-temperature overshoot than those with medium or no overshoot. Nevertheless, in high overshoot pathways, EW is used to compensate for higher emissions for the upcoming decades, which is a risky strategy, given the increased likelihood of climate disasters at high overshoot levels.

This global, centennial-scale assessment of the technical and physical potential of enhanced weathering (EW) does not capture all the local factors that could constrain its real-world efficiency, scalability, and sustainability[57]. Specifically, the model does not account for lower-than-expected geochemical carbon dioxide removal (CDR) resulting from incomplete basalt weathering or the formation of secondary minerals[58], socio-cultural and institutional barriers[59], or potential impacts of EW on human health or ecosystems[52,60]. Monitoring, reporting, and verification (MRV) can be expected to increase marginal costs[61]. The cost of MRV will depend on the specific technologies under development, which may vary between biotic and

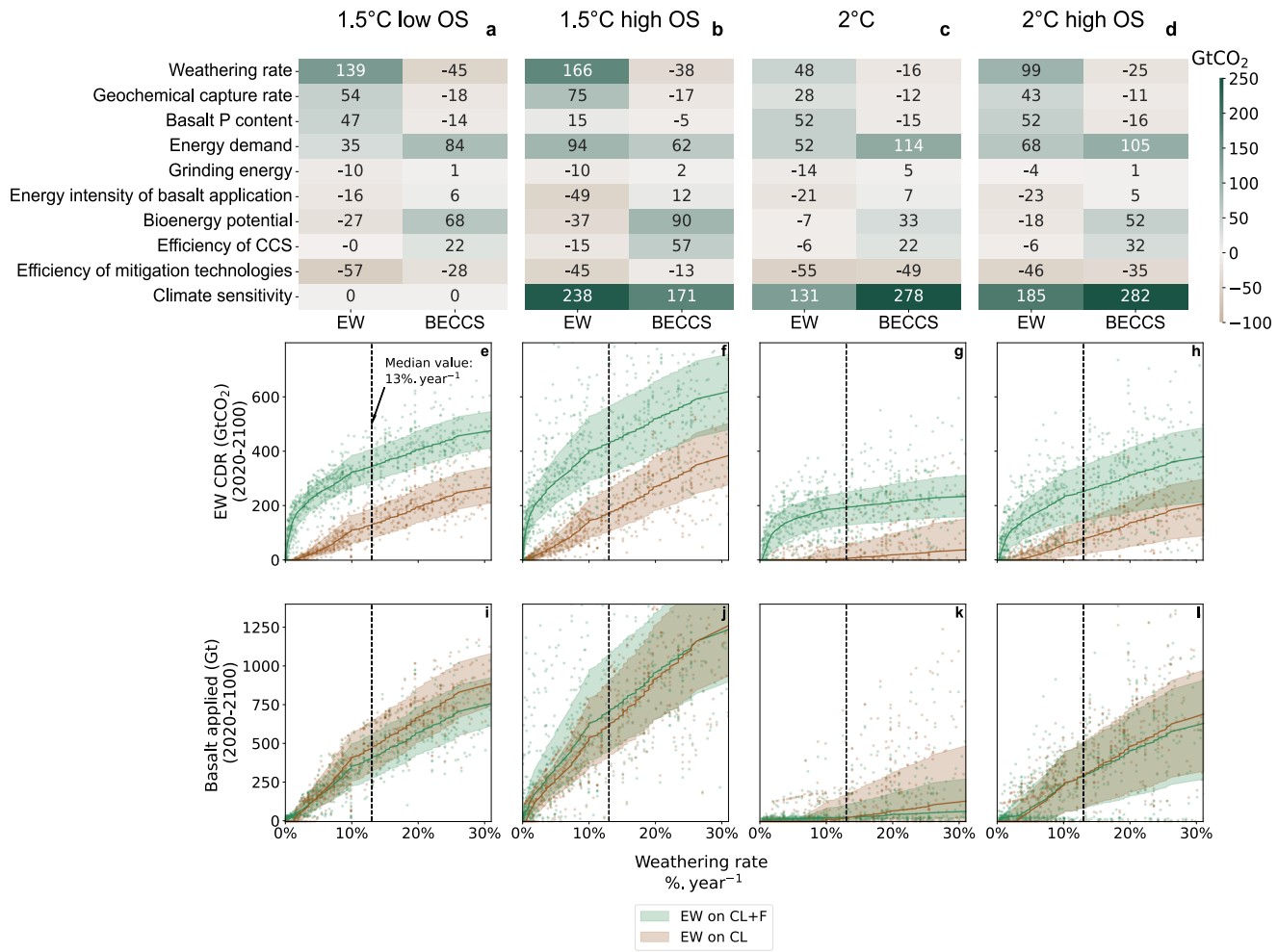

**Fig. 6 | Sensitivity analysis. a–d** Morris screening: mean variation of the output (columns), when the input (rows) is increased by half of its uncertainty range. The sources of uncertainty assessed are: the weathering rate, the geochemical capture rate, the phosphorus content of basalt, the baseline energy demand, the energy intensity of basalt grinding, the energy intensity of basalt application, the annual bioenergy potential, the efficiency of carbon capture and storage (CCS), the efficiency of other mitigation technologies and the climate sensitivity. The outputs displayed are: enhanced weathering ('EW'), the cumulative carbon dioxide removal (CDR) from EW when EW is applied on croplands and forests; 'BECCS', the cumulative CDR from bioenergy with carbon capture and storage (BECCS) when EW is applied on croplands and forests. **e–h** CDR from EW depending on weathering rate. **i–l** Application of EW depending on weathering rate. Each dot is a simulation in the sample. Solid line: median. Shaded area: 25–75% range for a given value of the weathering rate. The vertical dotted line shows the mean weathering rate considered in the rest of the paper (13% per year). Additional runs were performed to cover a wider range of weathering rates.

geochemical CDR pathways. However, how MRV costs could propagate to the marginal costs depends on the local regulatory framework. At the local scale, behavioural and institutional barriers, along with low social acceptance, may limit EW deployment as with other land-based CDR practices[59]. EW also faces low acceptability compared to other CDR methods, notably because of increased extractive activities[62,63]. For instance, the needed basalt extraction in the 1.5 °C with high overshoot case reaches 46 Gt/year (Fig. 7), which is half of the current global material footprint and ten times the global cement production today. This scale of mining could drive deforestation, disrupt ecosystems, and pose significant ecological and societal challenges[64]. The dust pollution associated with the aerial application of finely milled basalt could lead to silicosis and other respiratory diseases[65], and must therefore be prevented, for example by mixing the dust with water to form aggregates or by pelletisation[56]. The release of metals in basalts causing toxicity for humans must also be avoided in agricultural settings by choosing carefully the right material, and long-term studies on metal bioavailability and accumulation in soils, crops, and water systems are needed to assess potential health risks and inform regulatory guidelines[66]. Basalt dust potential impacts on tree canopy, possibly blocking leaf's stomata and reducing tree

growth[65,67] as well as potential impacts on riverine chemistry[68] must also be anticipated. The application of basalt in forests could alter soil geochemistry for centuries, possibly disrupting natural systems and impacting organisms among all trophic levels[69]. However, wisely exploited, these geochemistry side-effects could increase the potential for biotic CDR in addition to phosphorus fertilisation, as observed in an acid-rain impacted forest where the release of calcium through weathering of added silicate led to a biotic CDR of 3.2–3.5 tCO2 per ton of wollastonite applied[56]. Ultimately, biotic effects may either offset the net carbon removal in the case of soil carbon leaching to rivers[70], or enhance it by increasing soil carbon sequestration[33,71]. More experiments are therefore required to explore the side-effects of EW, and to determine the most suitable areas for basalt application particularly as rock material cannot be removed from the soil after its application.

At face value, a life cycle analysis comparing EW with other mitigation technologies showed that EW has the advantage to use less land than BECCS or afforestation, less energy than for direct air capture, and less water than for those three technologies[24]. The application of basalt in forests is, therefore, a promising method for mitigating climate change, but it requires the deployment of an appropriate

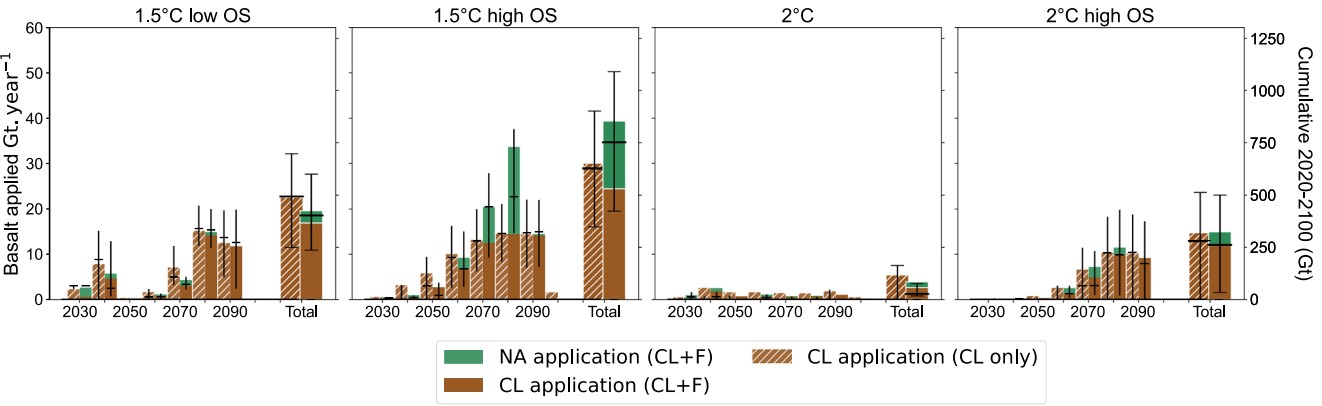

**Fig. 7 | Basalt applied on crop fields and forest areas.** Left y-axis: predicted annual application rate (Gt/year). Right y-axis: cumulative application across the 21st century (Gt). The coloured bars indicate the mean, the black horizontal dashes the median and the error bars the 25–75% range.

regulatory framework, to ensure that EW helps ecosystems sequester more carbon while minimising adverse side effects. Furthermore, even if we explored uncertainties as comprehensively as possible, the true uncertainties cannot be wholly captured inherently and certain classes of uncertainties cannot be assessed via quantitative means, indicating a need for careful interpretation and dissemination of our results for stakeholders.

## Methods
### Modelling framework

We developed an energy-economy-climate model by hard-linking GET7.1 with the reduced-complexity carbon cycle, atmospheric chemistry and climate model ACC2. GET7.0 is a bottom-up, cost-minimising energy system model, with a focus on energy supply and transformation. GET7.1 derives from GET7.0 with updated techno-economic parameters and a price-responsive energy demand following the SSP2 baseline[72]. The coupled model allows to assessment of least-cost emission pathways directly considering the temperature target (and not a carbon budget target) with a detailed representation of the energy system. Such a feature is important for an analysis under overshoot pathways involving several different greenhouse gases and EW as a CDR option, where the carbon budget approach may not necessarily work.

The coupled model produces internally consistent social-surplus maximising pathways to meet a reference energy demand in five end-use sectors (transportation, electricity, heat for industrial processes, space heat and industrial feedstocks), with perfect foresight, while respecting a given climate target as well as resource constraint for a range of primary energy source (oil, gas, coal, uranium, wind power, solar power, biomass and hydropower). Figure 1 shows the structure of the model. Primary energy is transformed into secondary and then final energy through investments and operations in order to satisfy a demand-supply equilibrium. CCS can abate emissions from fossil fuel power plants, or directly remove $CO_2$ from the atmosphere if combined with bioenergy (BECCS). The maximum achievable carbon capture by BECCS is limited by the deployment of carbon storage infrastructures, and bioenergy supply, as BECCS are competing with other bioenergy uses. Land-use is not explicitly modelled: the primary bioenergy supply is exogenously limited to 50 EJ in 2020 and to 260 EJ per year in 2100 following a supply curve (see SI.3). The growth rates of energy conversion technologies and $CO_2$ storage are limited under assumed upper bounds. There is no constraint on emission levels reduction rate as long as the energy demand is met. Since energy demand is price-responsive, stringent climate targets are achievable in an optimisation model sense. The energy module has a 10-year timestep, while ACC2 has an annual resolution.

The anthropogenic $CH_4$, $N_2O$ emissions and net energy-related $CO_2$ emissions are calculated in GET7.1 and are transferred to ACC2 for temperature calculations. The temperature calculations also use exogenous non-energy-related $CO_2$ emissions and other greenhouse gas and pollutant emissions, which are assumed to follow SSP1-1.9 and SSP1-2.6 for the 1.5 °C and 2 °C target cases, respectively. In ACC2, a box model represents oceanic and terrestrial carbon cycles. The ocean $CO_2$ uptake is represented by a four-layer box model, with the uppermost representing the atmosphere and ocean mixed layer and the others the ocean's inorganic carbon storage capacity, and the land $CO_2$ uptake model consists of four reservoirs connected with the atmosphere. The atmospheric concentrations of multiple greenhouse gases respond dynamically to their emissions and removals e.g., chemical sinks. The resulting radiative forcing is an input to a heat diffusion model, which further calculates the global temperature. The temperature change in turn affects the carbon cycle through soil respiration and ocean-atmosphere carbon flux. ACC2 is comprehensively described in ref. 30.

GET-ACC2 is fully coupled and optimised with perfect foresight, therefore the biotic CDR, which is the net increase of land carbon stock due to phosphorus fertilisation, the geochemical CDR, the basalt application, the energy system and the climate system are optimised simultaneously, reaching a global least-cost solution achieving the temperature target. No revenue flows are explicitly considered in the model. Carbon fluxes related to afforestation and deforestation are not optimised in the model.

### Enhanced weathering module: basalt supply

The enhanced weathering module has two main components: the basalt dust supply and the biogeochemical module calculating the removal rate of $CO_2$. The costs and energy requirements of basalt supply are integrated in GET. We followed ref. 15 for the parameterisation of the extraction, grinding and tractor application costs. The electricity for grinding basalt is 0.2 EJ/Gt (central value) and ranges from 0.07 to 0.6 EJ/Gt in the uncertainty analysis, which is the range provided in ref. 15 for grain size of 20 μm. Tractors used to spread basalt in agricultural fields and mining machinery were assumed to use petroleum products[73]. The energy requirements of transport from mines to application areas are based on ref. 9 and increase the energy demand in each transport subsector (road, train, or water freight). Transport modes are substitutable but an assumed minimum share (70–90%) must be transported on the road. The transport distance for basalt applied on croplands follows ref. 15, the mean distance for the basalt applied on forest areas ranges from 350 to 550 km. It was estimated by comparing a map of basalt resources[52], airports[74] and suitable application sites (see SI.1). A share of the basalt spread on forests is assumed to be applied with aircrafts (70–90%),

since 80% of the global surface is more than 1 km away from the nearest road[53]. This share can be expected to decline in the future, with the expansion of new roads[75]. On the other hand, applying basalt on forests by means of land transport can be challenging, even when a road is available. This share depends on the development of roads and on the share of forests that tractors can penetrate. Aerial application is likely to be more expensive and energy intensive than land-based alternatives. If the share of aircraft use were lower, the energy consumption would also be lower, and basalt application could become higher in a cost-optimisation model. Assuming a large part of aerial spraying is therefore pessimistic as far as costs are concerned.

### Enhanced weathering module: airborne basalt application

Global forests are divided into five land response classes based on the net primary productivity (NPP) response to basalt addition. Basalt is applied evenly in a given class, but the cost and energy intensity of application depend on the time taken to apply 1 ton of rock, which is inversely proportional to the desired application rate of rock in kg/m² over the fixed area of a given land response class.

We considered that small agricultural aeroplanes such as the AirTractor 802 (AT802), which can be equipped with a dust spreader, could be used to spread basalt dust. This kind of aircraft is commonly used to spread limestone[76–78] although issues with rock discharge have been reported, due to the wide range of the particle size distribution[76]. Details of rock dust discharge are beyond the scope of this analysis, and more research would be needed on how to spread large quantities of basalt dust by air. An AT802 burns 330 l of kerosene per hour, flies at 306 km/h, can carry 4.3 t of rocks (we assume that rock dust can be spread without mixing it with water, either as flee-flowing particles or as pellets) and costs USD 1.8 million. Using an open-source map of airports, we estimate that the mean distance per flight ranges between 160 and 240 km. If one adds 10 min for spreading operations, the average flight should last between 41 and 57 min, lasting longer if the application rate (in kg/m²) is lower, and thereby increasing the application cost per ton of rock. This represents an energy use of 1.8–2.5 GJ/t$_{rock}$ for a spreading duration of 10 min, but it could virtually be infinite for infinitely low application rates. Assuming 20 min of ground operations per flight, 10 h of use per day, five days out of seven, the capacity cost for spreading one ton per year is $170–210. Including ground operation and maintenance, pilot fees, insurance and housing (see SI 1.3), the non-energy cost per ton applied in forests is $110–170, and total costs (including energy costs) reach $142–355 per ton of basalt depending notably on the energy prices. We do not consider the non-CO$_2$ climate effect of aerial application[79], nor those of diesel combustion[80].

### Enhanced weathering module: ORCHIDEE emulator

The biogeochemical module is an emulator of the response of NPP to phosphorus fertilisation induced by the dissolution of phosphorus-rich basalt dust, as simulated by the land surface model ORCHIDEE-CNP model. Tailored simulations in which basalt dust is applied on all ice-free non-agricultural land in the year 2018 were used for the calibration. Once applied to soils, basalt is assumed to have a constant dissolution rate, referred to in this study as the weathering rate. This simplistic approach reflects current understanding and data availability to parameterise weathering rates, and does not account for certain phenomena, such as the reduction of reactive surface over time, as well as the soils, plants and hydrological processes that can potentially influence weathering. More detailed models that account for some of these processes have been proposed[12,81]; however, many processes are not yet well quantified, and consequently the weathering rates[52,82].

Here, the stock of basalt in soils $B$ [Gt] increases with the supply $S_B$ [Gt]. The dissolution of basalt follows a law of decay, parameterised

with the weathering rate $w_r$ [year$^{-1}$] as in Eq. (1).

$$\frac{dB}{dt} = - w_r B + S_B \tag{1}$$

As we assume that the grain size is 20 μm, a range of 1–25% per year is assumed for $w_r$. The high end is the global average weathering rate used in ORCHIDEE-CNP, where the pixel-level values are based on ref. 15 and on temperatures at a given model pixel. The low end follows ref. 48, which assumes similar grain size, temperature and pH as in ref. 15, but a lower dissolution rate per unit of specific surface area, and a lower specific surface area than in ref. 15. However, this uncertainty range is small compared to the variations in the observed weathering rates from different field and lab experiments[82], see SI 1.2.1 for a partial review of measured and simulated weathering rates.

Geochemical CO$_2$ capture happens when basalt dissolves: the released base cations (calcium potassium, natrium, and magnesium) are transferred to surface waters, where they are charge-balanced by the formation of bicarbonate ions[23]. The capture rate $p_B$ depends on the assumed concentration of these elements in rock material, and ranges between 0.24 and 0.37 t$_{CO2}$/t$_{rock}$. In GET-ACC2, the geochemical capture $G_{CO2}$ is assumed to be instantaneous and controlled by Eq. (2), but it should be noted that these values are not necessarily reached before minerals are leached to the ocean, and that the actual rate of in situ capture depends on local freshwater pH and alkalinity[83].

$$G_{CO_2} = p_B w_r B \tag{2}$$

The land carbon cycle component of ACC2 interacts with the enhanced weathering module. It consists of four carbon pools $C_i$ [Gt], with different turnover rates, which exchange carbon with the atmosphere. The inflow is the net primary production of the terrestrial biosphere: its magnitude is assumed to depend on the atmospheric CO$_2$ concentration and (to a lesser extent) on the global temperature change $\Delta T$. The outflow is the heterotrophic respiration (HR) (Eq. 3): it is proportional to the quantity of carbon in each pool and to their turnover rate $\frac{1}{\tau_i(\Delta T)}$ [year$^{-1}$], which increases with land surface temperature. The apparent NPP is thus the sum of the temperature-dependent NPP ($NPP^{climate}$), plus the CO$_2$ fertilisation effect $F^{CO2}$ (Eq. 4). The net land sink is thus the difference between the NPP and the heterotrophic respiration, and is zero at equilibrium (Eq. 5) (i.e., a quasi-steady state assumption at preindustrial). Note that land use CO$_2$ emissions are treated separately and do not directly influence the land biomass as typically assumed in many simple climate and carbon cycle models.

$$\Sigma_{i\in pools} NPP_i^{climate}(\Delta T) + F_i^{CO_2} = NPP^{climate}(\Delta T) + F^{CO_2}(\Delta CO_2) \tag{3}$$

$$HR_i(t) = \frac{C_i(t)}{\tau_i(\Delta T)} \tag{4}$$

$$\frac{dC_i}{dt} = NPP_i(t) - HR_i(t) \tag{5}$$

The dissolution of basalt releases phosphorus which is available for plant uptake, leading to an increase in the NPP by a fraction $\delta NPP(t)$. In the extreme case where 50 kg/m² of basalt dust is applied on all forests worldwide, global NPP over the next 40 years is 4.4 GtC/year higher on average than without basalt application, based on the results of ORCHIDEE-CNP (Daniel Goll, unpublished). The assumed phosphorus content in basalt is 0.161%-weight (with an uncertainty range of 0.036–0.28%), thus 50 kg/m² on 41 M km² would supply 70 times the current global use of phosphorus as a fertiliser.

Global NPP is higher in ACC2 than in ORCHIDEE. Therefore, in order to replicate the absolute magnitude of its increase in ORCHIDEE, we scale its increase by the ratio of its respective initial values in the two models as in Eq. (6). The $CO_2$ fertilisation term $F^{CO2}$ is not affected by phosphorus from basalt as it was calibrated based on predictions of models which omit nutrient constraints on the $CO_2$ fertilisation effect[30] and thus reflects an upper boundary of the stimulation of NPP by increasing $CO_2$[84] which cannot be further enhanced by phosphorus additions.

We limit basalt application to forest ecosystems, where the stimulation of NPP results in substantially more carbon sequestered for multiple decades compared to grasslands in simulations by ORCHIDEE-CNP.

$$NPP_i(t) = NPP_i^{climate}(\Delta T)\left(1 + \frac{NPP^{ORCHIDEE-CNP}(2018)}{\Sigma_{i \in pools}NPP_i^{climate}} \delta NPP(t)\right) + F_i^{CO_2}(t)$$

$$(6)$$

The increase in the NPP is followed by the increase of heterotrophic respiration, which releases a part of the sequestered carbon following the decay rate constant (E.3b). Our phosphorus cycle emulator quantifies $\delta NPP$, the fractional increase of NPP following basalt application: $\delta NPP = \frac{NPP_{EW}}{NPP_{Baseline}} - 1$.

In the spatially explicit land surface model ORCHIDEE-CNP, the increase of NPP due to phosphorus release depends on the soil, biome and climate and saturates with increasing basalt additions.

Application pixels are ranked according to their NPP stimulation from high to low, and grouped in M land response classes of areas $a_i$. In the current setting, $M = 5$ (more details on classes in the SM). A function of the rock application rate $c_{B,i}$ [Gt M km$^{-2}$] is used to fit the mean NPP response in each class $i$ during the forty years that follow basalt application, $\delta \bar{NPP_i}$ (Eq. 7). These classes are an implicit representation of the spatial heterogeneity of the response of forest ecosystems to phosphorus addition.

$$\delta \bar{NPP_i} = \delta N\bar{PP}_{i, max}\left(1 - e^{-\alpha_i c_{B,i}}\right)$$

$$(7)$$

The emulator is based on the following assumptions: the increase in NPP in class $i$ responds to the increase $\delta c_{P,i}$ in soil phosphorus concentration [Gt.Mkm$^{-2}$], which is proportional to the application rate of basalt, and decreases over time (Eq. 8).

$$\delta NPP_i(t) = \delta NPP_{i, max}\left(1 - e^{-\alpha_i \delta c_{P,i}(t)}\right)$$

$$(8)$$

The dynamic evolution of the soil phosphorus concentration $\delta c_{P,i}$ is designed to reproduce the results of ORCHIDEE-CNP. It is modelled with an auxiliary pool of phosphorus which is unavailable to plants, exchanges phosphorus with the soil concentration with exchange times $\tau_{p,i}$ and $\tau_{u,i}$, and is leached to inland waters with a time $\tau_{l,i}$ (Supplementary Fig. 10). Noting $B_i$ the undissolved basalt in class $i$, and $\delta u_{P,i}$ the concentration of unavailable phosphorus, we calibrate the exchange times on the ORCHIDEE-CNP outputs using the following system of equations (see SM for more details on the calibration procedure).

$$\frac{d\delta c_{P,i}}{dt} = \frac{\lambda w_r B_i}{a_{B,i}} - \frac{\delta c_{P,i}}{\tau_{p,i}} + \frac{\delta u_{P,i}}{\tau_{u,i}}$$

$$(9)$$

$$\frac{du_{P,i}}{dt} = \frac{\delta c_{P,i}}{\tau_{p,i}} - \frac{\delta u_{P,i}}{\tau_{u,i}} - \frac{\delta u_{P,i}}{\tau_{l,i}}$$

$$(10)$$

Supplementary Fig. 11 shows the comparison of the emulator with ORCHIDEE-CNP data. Finally, the total NPP increase is the sum of the

increase overall land response classes.

$$\delta NPP(t) = \sum_i \delta NPP_i(t)$$

$$(11)$$

## Uncertainty analysis

To assess the sensitivity of our results to the uncertainty of parameters, we apply a double uncertainty analysis.

### Quasi Monte-Carlo

First, we use a quasi-Monte Carlo sampling method to derive the distribution of outputs from the distributions of parameters. A Quasi-Monte Carlo method is similar to a Monte Carlo method but uses quasi-random sampling instead of random sampling to minimise errors. The Latin Hypercube Sampling method is used. On the supply side, we vary the costs and efficiency of new technologies, as well as their maximum diffusion speed and rates. The climate model uncertainty is also quantified by varying the equilibrium climate sensitivity[46]. More details about the parameters assessed, as well as their distribution, are described in SI.2.

### Morris method

Second, we apply the Morris screening method[44,45,85] to quantify the influence of each parameter on the outputs. Let $\boldsymbol{X} = \{x_1, \ldots, x_m\}$ be a vector of parameters which are normalised to [0,1], $Y = f(\boldsymbol{X})$ the output. A *trajectory* T is initiated by choosing an initial point $\boldsymbol{X_0^t}$ in $\left[\frac{1}{2*N-1}, \frac{2}{2*N-1}, \cdots, \frac{2N-1}{2(2N-1)}\right]^m$, and then iteratively increasing each parameter $i$ by $\frac{N}{(2N-1)}$ in random order $\{P^t(i)\}_{i \in [1,p]}$ where $P^t$ is a permutation, to obtain T $= \{\boldsymbol{X_0^t}, \boldsymbol{X_1^t} \ldots \boldsymbol{X_m^t}\}$. Computing the output along this trajectory yields the elementary effects for each parameter $i$: $d_i^t = f(\boldsymbol{X_{\sigma(i)}^t}) - f\left(\boldsymbol{X_{\sigma(i)-1}^t}\right) = f(x_1, \ldots x_i + \Delta \ldots x_m) - f(x_1, \ldots x_m)$. We produce $N = 20$ trajectories. The means $\mu_i$ of the elementary effects, their standard deviation $\sigma_i$ and the mean of their absolute values $\mu_i^*$ give useful information about the influence of these parameters.

Initial points are sampled following a Latin Hypercube method, and trajectories are chosen to maximise their dispersion and thus their coverage of the parameters space, following ref. 45, but we improve the sampling strategy by changing the dispersion measure: we maximise the sum over all parameters of the Euclidean pairwise distances of all the points used to compute elementary effects of this parameter. Additionally, we use a simulated-annealing algorithm instead of their brute force approach, which greatly reduces the computational burden (more details in SI.2).

## Data availability

The ORCHIDEE-CNP simulations used to calibrate the emulator, and a Jupyter notebook used to calibrate the emulator are available at https://doi.org/10.5281/zenodo.12787826. Source data are provided with this paper.

## Code availability

The GET-ACC2 model description can be found in the supplementary information. The model code is available upon request.

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

## Author contributions

P.C., K.T. and Y.G. conceived the study and designed the methodology. Y.G. developed the emulator and coupled ACC2 and GET, performed simulations of GET-ACC2, generated figures, and led the manuscript

writing. D.J. gave technical support and advice on the economic part. D.S.G. provided conceptual advice on the biophysical part, and produced the new ORCHIDEE-CNP simulations used to calibrate the emulator. All authors contributed to the analysis and edited the manuscript.

## Competing interests

The authors declare no competing interests.
