## [Peer Review File · Nature Communications]

Leveraging ecosystems responses to enhanced rock weathering in mitigation scenarios

Corresponding Author: Dr Yann Gaucher

Version 0:

Reviewer comments:

Reviewer #1

(Remarks to the Author)

As one of the ideas to mitigate climate change, a significant way is exploration and application of the Carbon Dioxide Removal (CDR) technology. Thus, EW (Enhanced Weathering) is widely concentrated. This study highlights the potential and side effect of the ESW (Enhanced Silicate Weathering) to quantify the CDR potential of ESW in forest and cropland by establishing the model, putting forward an interesting data. It not only provides new support for ESW on a large scale in theory but also assesses the costs and side benefits for ESW (phosphorus fertilization). However, the obstacle of silicate weathering is the real weathering rate. As a great number of studies show that Enhanced Silicate Weathering has a considerable lower CDR rate than carbonate (Enhanced Carbonate Weathering). However, it does not agree with the results of some studies in field and simulation experiments, which show that the CDR rate of ESW is several orders of magnitude lower than prediction. If this factor is not fully considered, the results are not reliable. In my opinion, this study provides some guideline for some studies, especially in ESW. But some result such as real CDR rate still needs to be demonstrated by some field experiments. To sum up, as engineering research, this study has established a systematic and comprehensive model and reached an impressive conclusion, especially in the part of the cost and land carbon cycle. That is of great reference value for the development of ESW. But it is not the only key factor for ESW promotion and application on a large scale. Thus, this study should focus some view on the field weathering rate of basalt to avoid the CDR rate of ESW in cropland and forest by model be overestimated.

In a word, I think this study needs major revision.

Detailed comments are as follows:

1. In this study, an integrated model, based on climate change, land carbon cycle and energy economics, has been established to predict the potential of ESW, including the CDR rate, the cost, and the increase of the land carbon sink. This model is comprehensive, especially in cost and the phosphorus fertilization on the terrestrial ecosystem. It is positive and provides a guideline for relevant research. In addition, the discussion of environmental effect in the study is not sufficient, please consider as much as possible.
2. The application area is one of the key issues for EW. In theory, due to the global food crisis and the decline of farmland quality, cropland has always been regarded as an ideal area for EW application to seek the double win of carbon capture and increase grain production. It has also been suggested in previous studies that the ideal area of EW application is not only on cropland but also on the forest. This study uses the model to predict the potential of EW application in the forest and provides data supportive to breaking down the geographical restrictions of EW. As proposed in the manuscript that applying basalt to forests could triple the level of carbon sequestration induced by EW compared to an application restricted to croplands, which is an interesting theoretical value to deploy the ESW. What do you think of increasing grain yield by ESW in cropland? I think it is also important.
3. As a CDR technology, the potential of EW compared with other CDR technologies (Such as BECCS in this study) and quantify the side effect on ecosystem are some noticeable points. This study makes a model not only estimation for the ESW and BECCS but also quantitative analysis of the phosphorus fertilization for forest system, it is also meaningful. As we know, phosphorus fertilization by ESW at terrestrial ecosystem is not to be sniffed at. However, in this study, it is important to distinguish the inorganic carbon by basalt weathering and organic carbon by phosphorus fertilization.
4. A large number of studies have also been carried out on Enhanced Silicate Weathering, which shows that the field weathering rate of silicate (basalt, etc) is lower than the model (several orders of magnitude). Thus, this magnitude of the model is needed to be discussed. Similarly, this study predicts a theoretical value for ESW also hard to convince to support the

result of such a large CDR efficiency. It is of great necessary to verify or adjust with field data.

5. Some issues, how to improve the ESW real weathering rate, become the hot points rather than to predict theoretical magnitude of ESW. It is the dilemma of the promotion and application of the ESW. Although a lot of work has been carried out (seek the suitable grain size, consider Carbonic Anhydrases and Organic acids of fungi). Thus, improving the weathering rate rather than calculating the CDR rate is the key to the large-scale application of the ESW. How do you think to increase the CDR rate of ESW?

6. It is a generally acknowledged truth that the weathering rate of silicate is lower than carbonate. What do you think of ECW (Enhanced Carbonate Weathering)?

Reviewer #2

(Remarks to the Author)

Reviewer #3

(Remarks to the Author)

Review of "Leveraging ecosystems responses to enhanced rock weathering in mitigation scenarios" by Gaucher et al.

The paper presents the results of calculations that show that adding basalt powder to forests could increase the efficiency of enhanced weathering as a method for carbon removal from the atmosphere. Using their integrated carbon, climate and energy system model they propose that this approach could remove 3 times the amount of carbon when compared to adding the same amount of basalt to agricultural lands.

The topic is timely and I found the manuscript interesting if not entirely convincing. Adding basalt to forests has been proposed before (see for example de Oliveira Garcia, Wagner, et al. "Impacts of enhanced weathering on biomass production for negative emission technologies and soil hydrology." *Biogeosciences* 17.7 (2020): 2107-2133.) so that the main contribution of this article is an attempt to calculate this potential impact. However, the analysis suffers from a number of limitations:

1. The method of calculation involves a complex global model, which the authors present no attempt to calibrate or verify. While the authors do perform an uncertainty analysis using a Monte Carlo method, the overall reliability of the approach remains unclear.

2. The authors present no analysis of the influence of local factors such as climate, forest type, soil types, or different kinds of basalts that would need to be accounted for in order to reliably calculate the impact of enhanced weathering on both global and local scales. The absence of this analysis means that even if they are correct in some environments, in others there may not be any advantage at all to adding basalt to forests.

3. Besides the addition of phosphorus to soils by enhanced weathering, which is assumed to enhance growth, there is no discussion of the possible additional mechanisms by which CO₂ removal may be enhanced. Also, how would this work in mature forests in which increased nutrient supply may not necessarily translate into enhanced weathering. As such, the model behaves very much like a black box and is not convincing, again, without experimental verification.

Comments on a technical level:

4. The authors use so many acronyms that the writing is dense and, at times, incomprehensible. For example, at one point they write: "Global NPP is higher in ACC2 than in ORCHIDEE. Therefore, in order to replicate the magnitude of the increase in NPP from ORCHIDEE, we scale the increase in NPP by the ratio of preindustrial NPP in ACC2 and the NPP in ORCHIDEE for the year 2018 (E.4)." If you read this aloud, which I did, it just sounds like gobbledygook.

5. Figure 5 and 6 are too dense, and contain too much information for the average reader to follow or understand.

Version 1:

Reviewer comments:

Reviewer #3

(Remarks to the Author)

In the new manuscript, the authors have satisfactorily answered the questions I raised and I think the paper is suitable for publication.

(Remarks on code availability)

Reviewer #4

(Remarks to the Author)

Note: This is not a de novo review of the paper. The editor requested an assessment of how well the authors had responded to the suggestions in the first round reviewer reports.

This study quantifies the global potential for the combination of geochemical CDR as enhanced rock weathering of basalt and biotic CDR (increased biomass production) using a coupled system of models: the global energy transition model, the aggregated carbon cycle, atmospheric chemistry, and climate model, and the ORCHIDEE-CNP land-surface model. The analysis covers a significant set of scenarios including four different climate targets and three CDR portfolios and includes an uncertainty analysis of ten parameters, many of which were used to quantify enhanced weathering. Most notably, the study suggests that potential for biotic CDR from the application of basalt to forests could be a significant source of CDR under all four climate targets. The revised manuscript adequately responds to most of the reviewers' comments. For example, the authors made significant additions in response to reviewer one's first and second comments that enhanced the quality of the manuscript.

One remaining concern is that comments five and three, from reviewers two and three respectively, about the justification of an aggregated global analysis versus a regional analysis could be more fully addressed in the revised manuscript. It is likely that the global potential presented by this analysis overestimates the reasonably achievable deployment of EW CDR when additional regional factors are considered. It is recommended that the authors consider addressing more directly some of the constraints that may arise at the local scale that could reduce the aggregate potential for EW. These constraints can arise from the potential health and ecosystem services concerns (briefly mentioned in the current draft on lines 326 et seq. - maybe the authors would want to cite Choi et al 2021 here), but also from institutional frictions in shifting forest management practices. We are not suggesting that modeling these things are within the scope of the current paper but that the paper acknowledges the magnitude of the potential affect that these concerns may have in constraining the potential for forest EW. What does the modeling effort in this paper suggest about specific issues that need to be addressed to understand how region and locality specific constraints may limit the potential for EW? Given the potential importance of these factors, some additional comment on this point is warranted.

One relevant paper the authors do cite (although for a different purpose) is:

Fuhrman, J. et al. Diverse carbon dioxide removal approaches could reduce impacts on the energy–water–land system. *Nat. Clim. Chang.* 13, 341–350 (2023)

This paper, and other mentioned in the references below are examples of modeling exercises exploring regional issues in CDR deployment. It is not that these papers need to be cited, but that the implications of regional analysis on aggregate potential be fairly (if briefly) explored so as not to give an overly optimistic assessment of actual potential for forest EW. This has implications for the marginal cost estimates given in the "Enhanced weathering deployment" subsection.

Two small edits: On line 288, "Field or pot conditions" should probably read "Field or plot conditions"

The comma on line 38 should probably be deleted.

References:

Environmental Risks in Atmospheric CO₂ Removal Using Enhanced Rock Weathering Are Overlooked WJ Choi, HJ Park, Y Cai, SX Chang. *Environmental Science & Technology* 55 (14), 9627-9629

Some regionalized modeling of CDR include:

Fuhrman et al. (2023) already discussed.

Regional implications of carbon dioxide removal in meeting net zero targets for the United States Chloé Fauvel et al 2023 *Environ. Res. Lett.* 18 094019

Signed,

William M. Shobe, Research Professor of Public Policy (Emeritus), University of Virginia

(Remarks on code availability)

Reviewer #5

(Remarks to the Author)

(Remarks on code availability)

Reviewer #6

(Remarks to the Author)

General Comments:

The manuscript analysed the role of EW under cost-effective mitigation pathways, by including the CDR potential of basalt applications from silicate weathering (geochemical CDR) and enhanced ecosystem growth and carbon storage in response to phosphorus released by basalt (biotic CDR). Using an integrated carbon cycle, climate and energy system model, the authors assessed the potential of considering enhanced weathering applications for achieving the Paris Agreement targets. The study showed that the application of basalt to forests could triple the level of carbon sequestration induced by EW compared to an application restricted to croplands. I reviewed the comments and suggestions from previous reviewers and the authors' responses. I agree with most of the comments. The authors have addressed some of the comments. Furthermore, I have some other concerns as follows:

1. The research involves coupling several complex models in very different ways. As previous reviewers mentioned, ensuring the reliability of the simulations from your coupled model is critical. In this revision, the authors have paid significant attention to the uncertainties of the model. Using Morris methods, they have discussed the uncertainties of key model parameters and calibrated the weathering rates. Beyond these efforts, if possible, I suggest evaluating their assessments based on the coupled model by comparing them to existing enhanced weathering application studies.
2. I suggest the author reorganize an easy-read Introduction section by moving sentences/review for methods to the method sections, highlighting importance of EW application at the begin, and stating your objectives and mainly methods used at the last of introduction. The paper is very multidisciplinary research, and thus this is important for readers from a broad field.
3. The authors said "application of basalt to forests could triple the level of carbon sequestration induced by EW compared to an application restricted to croplands". Do you mean for all forest types? From perspective of biogeography and climate, different forest types (or forests with much different hydrologic and climate conditions) exhibit distinct different biogeochemical effects. I did not see how did you consider this important problem.
4. In addition to the high costs, large-scale applications of EW for natural ecosystems may induce ecological risk. Discussing where is suitable for application EW is important.
5. Please revise the entire text according to the journal's formatting requirements, such as using abbreviations and full names for the journal titles and figures (e.g., "Fig." for figures in abbreviation and "Figure" in full).
6. In the results section, some figures do not fully demonstrate the data due to the attempt to maintain uniform coordinate axes, while others concentrate within a narrow range, making it difficult to discern trends, as exemplified in Figure 4g. Consider adjusting these figures to better illustrate the differences between the data.
7. The background section lacks detailed knowledge on the application of the model used in this paper to the study of rock weathering carbon sequestration. The suitability of using this model is not apparent from the context provided. In addition, what are the advantages of the applied model compared with other models?
8. Why choose basalt for rock weathering research, and what impact does it have on climate change?
9. The results section of this paper cites numerous similar studies. What distinguishes this section from the discussion? It is recommended that the results section solely present the findings of this paper, leaving in-depth exploration and discussion of the relevant results to the discussion section.
10. What is the conclusion of this paper and does it conform to the hypothesis? What are the implications of this study's findings for climate change mitigation.

(Remarks on code availability)

Version 2:

Reviewer comments:

Reviewer #6

(Remarks to the Author)

Enhanced rock weathering (EW) can consume atmospheric carbon dioxide. The authors analysed the role of EW under cost-effective mitigation pathways, 11 by including the CDR potential of basalt applications from silicate weathering and 12 enhanced ecosystem growth and carbon storage in response to phosphorus released by basalt. However, the side effects of basalt applications on croplands should also be fully considered. Especially, the risks of accumulation of heavy metals and large scale basalt mining should be fully considered.

(Remarks on code availability)

Please find in the following our answers to the reviewers' comments. We carefully addressed all the comments and made modifications in the text accordingly. Additionally, we made several minor modifications visible in the "track change" document to remove typos and wrong citation formats.

Reviewer #1:

General Comment

Comment #1

As one of the ideas to mitigate climate change, a significant way is exploration and application of the Carbon Dioxide Removal (CDR) technology. Thus, EW (Enhanced Weathering) is widely concentrated. This study highlights the potential and side effect of the ESW (Enhanced Silicate Weathering) to quantify the CDR potential of ESW in forest and cropland by establishing the model, putting forward an interesting data. It not only provides new support for ESW on a large scale in theory but also assesses the costs and side benefits for ESW (phosphorus fertilization). However, the obstacle of silicate weathering is the real weathering rate. As a great number of studies show that Enhanced Silicate Weathering has a considerable lower CDR rate than carbonate (Enhanced Carbonate Weathering). However, it does not agree with the results of some studies in field and simulation experiments, which show that the CDR rate of ESW is several orders of magnitude lower than prediction. If this factor is not fully considered, the results are not reliable. In my opinion, this study provides some guidelines for some studies, especially in ESW. But some results such as real CDR rate still need to be demonstrated by some field experiments. To sum up, as engineering research, this study has established a systematic and comprehensive model and reached an impressive conclusion, especially in the part of the cost and land carbon cycle. That is of great reference value for the development of ESW. But it is not the only key factor for ESW promotion and application on a large scale. Thus, this study should focus some view on the field weathering rate of basalt to avoid the CDR rate of ESW in cropland and forest by model being overestimated.

In a word, I think this study needs major revision.

Response #1

We thank Reviewer #1 for reviewing the article, and for the positive comments and suggestions on our manuscript. Reviewer #1 raises the important issue of the uncertainty on real weathering rates, their possible over-estimation in our study, which highlights the need to reduce uncertainty on CDR rates by more field experiments.

The weathering rates used in modelling studies are calibrated on laboratory experiments, like in refs^{13,14,17-19}. As explained by Reviewer#1, some field studies have reported lower weathering rates than those expected from the laboratory experiments, but measurement techniques for field application are still in their infancy and methodological biases could explain the difference between results obtained in the lab versus in field experiments. Therefore we mention uncertainty on weathering rates in the text L304-322 (see below), and took it into account in our analysis. First, we reported uncertainty ranges on figures 2, 3 and 4, obtained with a Monte-Carlo process, including on the assumed weathering rate. Second, we assessed the contribution of the weathering rate uncertainty on the total uncertainty of the cumulative EW CDR (see figure 5). Namely, the cumulative EW CDR is close to zero for weathering rates below 0.1%/year, and increases with the weathering rate, typically exceeding 100 GtCO₂ for weathering rates above 1 %/year if basalt is applied on forests, whereas weathering rate must exceed ~ 10% for geochemical CDR to reach 100 GtCO₂ if basalt is only applied on croplands (see figure 6). We have also changed the last sentence of the paragraph L304-322 in order to insist on the fact that low weathering rates could render EW ineffective and unprofitable.

Main text L304-322

Weathering rates in soils remain highly uncertain^{17,20,21} (see **Methods**). The weathering rates used in modelling studies are calibrated on laboratory experiments^{13,14,17-19}. Field or pot conditions experiments generally provide lower estimates than laboratory experiments^{1,20,22,23}, because the weathering rate depends on complex interplay of soil pH, temperature, hydrological conditions, and biological activity^{11,23}. Soil column²² and mesocosm²⁰ experiments have suggested surface-normalised weathering rates respectively two and three orders of magnitude slower than those used in Strefler et al. 2018¹³, which defines the high range of the weathering rates used in the present work (25% per year). However, field²⁴⁻²⁷ and forest⁴ studies have shown promising CDR rates corresponding to weathering rates exceeding this range. For instance, ref²⁴ reports a $16 \pm 6\%$ loss of cations from basalt applied on agricultural crops over 4 years. This corresponds to a mean weathering rate of 3-6% per year. As they report a grain size of 267 μm , 20 μm -sized grains could weather around ten times faster, because weathering speed scales with the reactive surface. The wide variations across experiments indicate that the weathering rate is a critical source of uncertainty. As shown in **Figure 6(e-l)**, the lower the weathering rate, the less basalt is applied and the less carbon is captured. For weathering rates below 1% per year, EW can become a viable cost-effective option only if basalt is applied on natural areas, because the supply of even very low quantities of phosphorus to phosphorus-depleted soils yields a significant biotic CDR. Thus, the high efficiency of basalt application over forest is maintained even with low weathering rates (see also the sensitivity results using weathering rates of 1% and 25% in SI 1.1.1).

While we are aware of the uncertainty surrounding weathering rates, we would like to draw the attention of Reviewer#1 on recent field experiments results that have measured the ESW rates. We reviewed recent literature^{1,13,15,17,20,24,26,28-34} on weathering rates in the field, in mesocosms and in the lab. This literature is summarised in the following text and table, that were added to the Supplementary information (1.1.1). When the weathering rates were not directly available, we derived them indirectly from the amount and type of rocks and the total CDR achieved. As can be seen in the table, the

weathering rates are highly dispersed because of different local conditions and grain sizes, but also because there is a range of basalt materials that can be accessed (e.g. waste) or mined for the purpose of ESW, with different efficiencies. The data reported in the table do not allow us to assess if lab experiments provide a higher weathering rate than the field experiments. In our study, we assumed an average idealised material with a given weathering rate. We used a range from 1% to 25% per year. This range encompasses the only field experiment with basalt which has succeeded in measuring weathering (4 %/year Beerling et al, 2024) and the mesocosms experiments with basalt (Kelland et al 2020, Reershemius et al 2023), with the exception of Buckingham et al 2023. Furthermore, when weathering rates are scaled to 20µm-size grains, they are often above the 1-25%/year range that we use in our study, which suggests that we do not overestimate weathering rates. In the revised manuscript we added in SI 1.1.1 a reproduction of figures 2 and 3 for weathering rates equal to 1% and 25%.

SI 1.1.1

The weathering rates are strongly influenced by the rock types, the in situ physicochemical and hydrological conditions as well as by biological processes, which leads to a large dispersion of the experiments results. Furthermore, robust and standardised methods for measuring CDR and rock weathering are still lacking, which may also explain the dispersion of the observed weathering rates. In the following table, we reviewed the recent literature ^{1,13,15,17,20,24,26-34} on enhanced weathering experiments. The reported weathering rates were scaled to a value that would correspond to a grain size of 20µm. More precisely, if $w_r(d_1)$ is the weathering rate of a feedstock of grain size d_1 , we have $\frac{w_r(d_1)}{w_r(d_2)} = \left(\frac{d_2}{d_1}\right)^\alpha$. It is important to note that the relationship between weathering rate and grain size is uncertain, and that the existing literature reports both sublinear ($\alpha < 1$, e.g. ref²⁸) and superlinear ($\alpha > 1$, e.g. ref ^{13,17}) relationships between weathering rates and the inverse of grain sizes. For simplicity, we assumed that weathering rates are proportionate to the inverse of grain sizes ($\alpha = 1$), as would be the case for perfect spheres, whose weathering rates depend on the reactive surface. For each mineral, we assume an exponential law of dissolution like in the main text, where the share of dissolved basalt x_t follows: $1 - x_t = \exp(-w_r t)$. For experiments during less than one year, we thus to obtain the share dissolved after 1 year as follows: $1 - x_T = \exp\left(\frac{T}{D} \ln(1 - x_D)\right)$ where D is the duration of the equation and T is one year.

Study type	Grain size and reported weathering rate (% per year)	Linearly scaled weathering rate for 20µm-grains	Rock type	Ref
Reactor simulating humid tropical soil.	size: 300µm rate: 12-19 %/year	173-283 %/year	Basaltic flow (“Arenal”)	Ryan et al 2024

Reactor simulating humid tropical soil.	size: 47.2 μm rate: 30-42 %/year	70-100 %/year	Basaltic flow (“Barva”)	Ryan et al 2024
Reactor simulating humid tropical soil.	size: 14.1 μm rate: 20-78 %/year	14-55 %/year	Basalt flow (“BHVO-1”)	Ryan et al 2024
Reactor simulating humid tropical soil.	size < 45 μm rate: 3-27 %/year	7-63 %/year	Basalt (BR-fine)	Vanderkloot et al 2023
Reactor simulating humid tropical soil.	size <45 μm rate: 6-41 %/year	14-96 %/year	Basalt (PV-fine)	Vanderkloot et al 2023
Planted Mesocosm	size ~ 100 μm rate: 30 %/year (1.6 t/ha)	120 %/year	Olivine	ten Berge et al. 2012
Planted Mesocosm	size ~ 100 μm rate: 2 %/year (204 t/ha)	10 %/year	Olivine	ten Berge et al. 2012
Planted Mesocosm	size: 43 μm rate: 0.03%/year	0.06 %/year	Dunite	Amann et al. 2018
Planted Mesocosm	size: 1020 μm rate: 0.01%/year	0.5 %/year	Dunite	Amann et al. 2018
Planted Mesocosm	size: 20 μm rate: 91%/year (10t/ha)	91%/year	Olivine	Dietzen et al. 2018
Planted Mesocosm	size: 20 μm rate: 40 %/year (50t/ha)	40 %/year	Olivine	Dietzen et al. 2018
Planted Mesocosm	size: 25 μm rate: 94 %/year	117 %/year	Wollastonite	Haque et al. 2019
Planted Mesocosm	size: 1250 μm rate: 17 %/year	600 %/year	Basalt	Kelland et al 2020
Planted Mesocosm	size: 128 μm rate: 21.7 %/year	139 %/year	Basalt	Kelland et al 2020
Planted Mesocosm	size: 35 μm rate: 23 %/year	40 %/year	Basalt	Reershemius et al 2023

Field Trial (oil palm plantation)	No weathering measured	No weathering measured	Basalt	Larkin et al 2022
Field trial (Potato field)	size ~ 10 μ m rate: 93%/year	41 %/year	Wollastonite	Haque et al 2020
Field trial (Soybean field)	size ~ 10 μ m rate: 43%/year	21 %/year	Wollastonite	Haque et al 2020
Field Trial (Corn plantation)	size: 367 μ m rate: 3-6 %/year	48-110 %/year	Basalt	Beerling et al 2024
Field Trial (Rice paddy)	size < 75 μ m rate: 91%/year	341%/year	Wollastonite	Wang et al 2024
Soil core study	size: 125-250 μ m rate: 0.04-0.05 %/year	0.5-0.8 %/year	Basalt	Buckingham et al 2022
Modelling (reactive transport model)	size: 714 μ m rate: 5.7 %/year	205 %/year	Basalt (Oregon)	Lewis et al 2021
Modelling (reactive transport model)	size: 1128 μ m rate: 5.5 %/year	308 %/year	Basalt (Craigmill)	Lewis et al 2021
Modelling (reactive transport model)	size: 1531 μ m rate: 4.7 %/year	362 %/year	Basalt (Tichum)	Lewis et al 2021
Modelling (reactive transport model)	size: 267 μ m rate: 0.8 %/year	11 %/year	Basalt (Blue ridge)	Lewis et al 2021
Modelling (reactive transport model)	size: 1767 μ m rate: 3.8 %/year	336 %/year	Basalt (Tawau)	Lewis et al 2021

Ryan et al, 2024: The share of minerals present in the rocks that were leached after 14 days are given in their table 4b. For each rock type, we report the range of weathering rates of Ca and Mg scaled by the grain size.

Vanderkloot et al 2023 reports a similar experimental design as Ryan et al, 2024. We do the same calculation to extrapolate annual weathering rates, considering only the cases with fine grain size (<45 μ m) to limit the errors when scaling to 20 μ m grains.

Kelland et al 2020 reports that 17% and 21.7% of the maximal CO₂ removal were obtained after 1 year, for relatively large grains (p80 = 1350 μ m, p50 \approx 700 μ m). The weathering rates for ten Berge et al.

2012, Amann et al 2018, Dietzen et al 2018 and Haque et al 2019 were obtained from the table 3 of Kelland et al 2020

Reershemius et al 2023 reports that 15.7% of the maximum CO₂ removal was obtained after 235 days for grain size close to 20µm (p80 = 35µm, p50 ≈ 20µm).

Larkin et al 2022 reports no difference of CO₂ drawdown through alkalinity export between basalt-amended and reference fields, which could be explained by the time-lag in the exports of weathered cations, which does not affect the overall amount of CDR that will occur²⁴.

Haque et al 2020 reports that in one field, the application of wollastonite at the rate of 1.24t/ha resulted in a CDR of 0.32tCO₂/ha after 5 months, which means that around 70% of the rocks have been weathered in 5 months assuming a stoichiometric rate of 1 mole of CO₂ sequestered per mole of wollastonite weathered after full precipitation of carbonates. The same calculation for another field, where coarser wollastonite was applied, indicates that around 20% of the initial input was weathered after 20 weeks.

Beerling et al 2024 reports a $16 \pm 6\%$ loss of cations from basalt applied on crop fields over 4 years. This corresponds to a mean weathering rate of around 4% per year. Since the reported grain size in this study is 267 µm, and the weathering speed is proportional to the reactive surface of the grains, we can extrapolate that 20µm-sized grains should weather more than ten times faster.

Wang et al 2024 report that applying 5t/ha of wollastonite yields a CDR of 1.2tCO₂/ha after 5 months.

Buckingham et al 2022 report the release rates (in mol.cm⁻².s⁻¹) of calcium and magnesium ions from basalt grains of size 150-250 µm. As they also report the mass composition of their basalt feedstock and its reactive surface, it is straightforward to derive the corresponding weathering rate in %/year. These results they report have been contested (ref³⁵, with a response in ref³⁶).

Lewis et al 2021 simulate the weathering of different basalt feedstocks depending on their mineralogy using a reactive transport model. They report the share of maximum CDR after 15 years, which we assume to be equal to the share of rock weathered.

Rinder et von Hagke 2021 simulate the weathering of 20-µm sized grains and obtain lower weathering rates than Strefler et al 2018, due to a lower reactive surface.

Due to the importance of weathering rates, we reproduced Figures 2 and 3 for weathering rates equal to 1%/year and 25%/year, which are the bounds of the assumed uncertainty range. These figures show that although the biotic and geochemical CDR are higher at high weathering rates, the biotic effect is, relative to the geochemical effect, more efficient at low weathering rates.

1.1.1.1 Weathering rate = 1% per year

Figure S1.1.1.a| Carbon dioxide emissions from three CDR portfolios for different climate targets across the 21st century. Discount rate = 5%. Weathering rate= 1% per year.

Figure S1.1.1.b Carbon dioxide emissions from three CDR portfolios for different climate targets across the 21st century. Discount rate = 2%. Weathering rate= 1% per year

Figure S1.1.1.1.c| Temperature, policy costs and carbon price pathways across the 21st century. Discount rate = 5%. Weathering rate= 1% per year.

Figure S1.1.1.1.d| Temperature, policy costs and carbon price pathways across the 21st century. Discount rate = 2%. Weathering rate= 1% per year.

1.1.1.2 Weathering rate = 25% per year

Figure S1.1.1.2.a Carbon dioxide emissions from three CDR portfolios for different climate targets across the 21st century. Discount rate = 5%. Weathering rate= 25% per year.

Figure S1.1.1.2.b Carbon dioxide emissions from three CDR portfolios for different climate targets across the 21st century. Discount rate = 2%. Weathering rate= 25% per year

Figure S1.1.1.2.c) Temperature, policy costs and carbon price pathways across the 21st century. Discount rate = 5%. Weathering rate= 25% per year.

Figure S1.1.1.2.d) Temperature, policy costs and carbon price pathways across the 21st century. Discount rate = 2%. Weathering rate= 25% per year.

Specific Comments

Comment #2

In this study, an integrated model, based on climate change, land carbon cycle and energy economics, has been established to predict the potential of ESW, including the CDR rate, the cost, and the increase of the land carbon sink. This model is comprehensive, especially in cost and the phosphorus fertilisation on the terrestrial ecosystem. It is positive and provides a guideline for relevant research. In addition, the discussion of environmental effects in the study is not sufficient, please consider as much as possible.

Response #2

Thank you for the positive comment about the scope of the and the model we present. We agree on the importance of discussing environmental effects and have modified the main text to mention different environmental side-effects and the risks associated with basalt dust application (L343 - 372).

Main text, L343-372

Potential impacts of EW on human health or ecosystems^{1,2}, possible scaling constraints on basalt supply, or lower-than-expected geochemical CDR from incomplete basalt weathering are not considered in the model, which may both limit the sustainability and scalability of basalt soil amendment and restrict EW efficiency. For instance, the needed basalt extraction in the 1.5°C with high overshoot case reaches 46 Gt/year (Fig. S1), which is half of the current global material footprint and ten times the global cement production today, potentially having a large ecological and societal impact. The dust pollution associated with the aerial application of finely milled basalt could lead to silicosis and other respiratory diseases³, and must therefore be prevented, for example by mixing the dust with water to form aggregates or by pelletisation⁴. The release of metals in basalts causing toxicity for humans must also be avoided in agricultural settings by choosing carefully the right material⁵. Basalt dust potential impacts on tree canopy, possibly blocking leaf stomata and reducing tree growth^{3,6} as well as potential impacts on riverine chemistry⁷ must also be anticipated. The application of basalt in forests could alter soil geochemistry for centuries, possibly disrupting natural systems and impacting the composition of plant communities⁸. Wisely exploited, these geochemistry side-effects could serve as an additional tool for biotic CDR in addition to phosphorus fertilisation, as observed in an acid-rain impacted forest where the release of calcium through weathering of added silicate led to a biotic CDR of 3.2-3.5 tCO₂ per ton of wollastonite applied⁴. Ultimately, biotic effects may either offset the net carbon removal in the case of soil carbon leaching to rivers⁹, or enhance it by increasing soil carbon sequestration^{10,11}. More experiments are therefore required to explore the side-effects of enhanced weathering particularly as rock material cannot be removed from the soil after its application.

At face value, a life cycle analysis comparing EW with other mitigation technologies showed that EW has the advantage to use less land than BECCS or afforestation, less energy than for direct air capture, and less water than for those three technologies¹². The application of basalt in forests is therefore a promising method for mitigating climate change, but it requires the deployment of an appropriate regulatory framework, to ensure that EW helps ecosystems sequester more carbon while minimising adverse side-effects. Furthermore, even if we explored uncertainties as comprehensively as possible, the true uncertainties cannot be wholly captured inherently and certain classes of uncertainties cannot be assessed via quantitative means, indicating a need for careful interpretation and dissemination of our results for stakeholders.

Comment #3

The application area is one of the key issues for EW. In theory, due to the global food crisis and the decline of farmland quality, cropland has always been regarded as an ideal area for EW application to seek the double win of carbon capture and increase grain production. It has also been suggested in previous studies that the ideal area of EW application is not only on cropland but also on the forest. This study uses the model to predict the potential of EW application in the forest and provide data supportive to breaking down the geographical restrictions of EW. As proposed in the manuscript that applying basalt to forests could triple the level of carbon sequestration induced by EW compared to an application restricted to croplands, which is interesting theoretical value to deploy the ESW. What do you think of increasing grain yield by ESW in cropland? I think it also important.

Response #3

Thank you for the comment. We agree that increased yields could be an important co-benefit of EW. The potential co-benefit of ESW to increase grain yield is an important feature of this CDR method and has been discussed in several publications, with first evidence from field studies²⁴. Therefore, the application of EW on croplands is studied in the literature, and it is one of the two cases of application studied in the paper. We did not account for the yield increase from EW in our model, however, because it does not represent agriculture, but we mention it in the introduction of the paper (L 45-47). We ignored as well the possibility to apply EW on BECCS which may increase their yield, a process not included in our study. Although basalt addition could substitute to conventional fertilisers, the associated benefit is expected to offset only a small share of the cost of EW (around 5%¹⁵). We added on L137 the following sentence :

Main text L45-47

Furthermore, co-benefits of crop yields from amendment with basalt have been studied in previous work, including from dedicated experiments showing: improvement of soils quality³⁷ which could enable a reduction of fertilisers use^{38,39}, improved plant health, and increased yields^{24,27,32,34,40}.

Main text L135-137

Possible co-benefits could increase the cost-effectiveness of EW¹⁵ if they can act as fertilisers, but our model does not include food systems and land use and these effects were not considered in the study.

Comment #4

As a CDR technology, the potential of EW compared with other CDR technologies (Such as BECCS in this study) and quantify the side effect on ecosystem are some noticeable points. This study makes a model not only estimation for the ESW and BECCS but also quantitative analysis the phosphorus fertilization for forest system, it is also meaningful. As we know, phosphorus fertilization by ESW at terrestrial ecosystem is not to be sniffed at. However, in this study, it is important to distinguish the inorganic carbon by basalt weathering and organic carbon by phosphorus fertilization.

Response #4

We do account for these two effects (organic carbon increase by phosphorus addition and inorganic carbon removal). We are sorry if it was not clearly explained and we made sure it is the case in the revised manuscript. In the main text, we distinguish inorganic and organic carbon sequestration by calling them geochemical and biotic CDR, respectively. For this purpose, we display organic and inorganic CDR with different colours on figure 2, and discuss the impact of each process on mitigation targets, costs See L152-164. To further highlight the distinction, we have also added the terms geochemical and biotic to the abstract.

Main text L152-164

Allowing basalt application in forests reduces the carbon price threshold above which EW becomes cost-efficient and increases the CDR potential in two ways: by expanding the area for basalt application, increasing the geochemical CDR potential, and by enabling biotic CDR in forests. As a consequence, the EW-induced CDR is almost tripled when basalt can be applied on forests, with a peak of 12.4 GtCO₂ per year and 446 GtCO₂ cumulatively in the 1.5°C with high overshoot scenario. The relative contributions of geochemical and biotic removals vary depending on scenarios; for example, in the 1.5°C with high overshoot scenario, the increase in geochemical CDR due to the additional area available is more pronounced than in other scenarios because cropland application is at full potential. Furthermore, the share of the biotic CDR is proportionally lower at high application levels because the phosphorus stimulation of forest production gradually saturates, thereby limiting the biotic CDR potential. This limit explains why biotic CDR by EW varies less than geochemical CDR by EW among different scenarios.

Abstract

Carbon dioxide removal (CDR) is deemed necessary to attain the Paris Agreement's climate objectives. While bioenergy with carbon capture and storage (BECCS) has generated substantial attention, sustainability concerns have led to increased examination of alternative strategies, including enhanced rock weathering (EW). We analyse the role of EW under cost-effective mitigation pathways,

by including the CDR potential of basalt applications from silicate weathering (geochemical CDR) and enhanced ecosystem growth and carbon storage in response to phosphorus released by basalt (biotic CDR). Using an integrated carbon cycle, climate and energy system model, we show that the application of basalt to forests could triple the level of carbon sequestration induced by EW compared to an application restricted to croplands. EW also reduces the costs of achieving the Paris Agreement targets as well as the reliance on BECCS. Further understanding requires improved knowledge of weathering rates and basalt side-effects through field testing.

Comment #5

A large number of studies have also been carried out on Enhanced Silicate Weathering, which shows that the field weathering rate of silicate(basalt, etc) is lower than the model(several orders of magnitude). Thus, this magnitude of the model is need to be discussed. Similarly, this study predict a theoretical value for ESW also hard to convince to support the result of such a large CDR efficiency. It is of great necessary to verify or adjust with field data.

Response #5

Thank you for this comment. We refer reviewer#1 to our response to Comment#1 above, regarding the possible overestimation of the weathering rate, where we gathered results from field data. We also refer reviewer#1 to the supplementary information of Goll et al. 2021, Part 2.2 “Weathering rates in laboratory and field conditions”, which reports that, in fact, slightly higher weathering rates have been observed in the field than in the laboratory (when correcting for particle size distributions), a phenomenon that might be caused by weathering-enabling biological processes which are absent in laboratory conditions¹¹.

Comment #6

Some issues, how to improve the ESW real weathering rate, become the hot points rather than to predict theoretical magnitude of ESW. It is the dilemma of the promotion and application of the ESW. Although a lot of work has been carried out(seek the suitable grain size, consider Carbonic Anhydrases and Organic acids of fungi). Thus, improving the weathering rate rather than calculating the CDR rate is the key to the large-scale application of the ESW. How do you think to increase the CDR rate of ESW?

Response #6

We agree that better knowledge of the weathering rate is critical for implementing cost effective ESW projects. In absence of large scale values for those rates, in our study we adopted an honest and large range of uncertainty for the weathering rate parameter, and propagated this uncertainty into the inferred amount of CDR. We developed the paragraph L281-289 (see Response#1) to emphasise the fact that the

assumed weathering rates are crucial to determine how important EW can be as a climate solution, for instance compared to BECCS. We also note that BECCS itself has very large uncertainties depending on the assumed yield and capture efficiencies, direct air capture has huge uncertainties for the best achievable rates, etc ... It is important to consider those uncertainties in a holistic approach when using an integrated model like ours, and we have done so by providing a systematic uncertainty analysis of our results to key parameters

Comment #7

It is a generally acknowledged truth that the weathering rate of silicate is lower than carbonate. What do you think of ECW(Enhanced Carbonate Weathering)?

Response #7

We considered basalt, hence silicates and not carbonates, because there is enough data for silicate-based enhanced weathering to be incorporated in a model (accounting for uncertainties, of course), which is less the case of ECW.

The effectiveness of enhanced carbonate weathering could be more challenging to assess with regard to measurement, reporting and verification of carbon removal, as carbonates can re-precipitate and offset a part of the CO₂ uptake, in particularly under climate and vegetation changes: Modelling found that the efficacy of enhanced carbonate weathering is a function of the capacity that rivers have for transporting the products from carbonate weathering to the oceans, rather than of the dissolution capacity of soils⁴¹.

Reviewer #2:

General Comment

Comment #1

The study develops a techno-economic module of Enhanced Rock Weathering applied to crops and natural lands, which is integrated into a coupled energy-climate model to assess the importance for achieving Paris-compatible climate targets focusing on emissions, energy use and emissions & removals. With respect to the later not only the removal due to abiotic weathering is represented, but also the enhanced soil quality due to phosphor fertilization. The results suggest that EW on natural can contribute a substantial share, particularly, in case of a 1.5°C target with a large overshoot. The authors also add an uncertainty analysis.

The study, thus, covers a broad range of disciplines and different methods. As a reviewer, I ask myself whether it would have been better to split the overall study into two, separating the technoeconomic and geological assessment from the assessment of EW in a broader energy-climate modelling framework. The advantage of a single contribution is that the reader is provided with a full package, the disadvantage is that it always seems a bit premature (which on the other hand also indicates novelty). I think the authors should seriously think about a rebalancing. I recommend them to put more emphasis on the uncertainties and highlight the exploratory nature of the study. It is a scientific value added to clearly highlight the uncertainties and discuss them, if a novel climate change mitigation option is considered to be deployed at large scale.

The authors put much emphasis on the cross-sectoral effects regarding the deployment of bioenergy with CCS and also the food prices (that are not explicitly modeled, though). The authors have put these cross effects into the abstract. I do not think that this is the major point to be discussed at that stage of the scientific discussion. It would be much more important to discuss for example the regional heterogeneity and what factors are crucial for the weathering rates and what can be said about them. Most of this has been moved to the technical appendix, but it should be presented, analyzed and discussed in the main manuscript.

Response #1

We thank reviewer#2 for the insightful review and comments. As suggested by reviewer#2, we have rebalanced the paper to highlight the exploratory nature of the study, see more details in the point-by-point answer below. We also refer reviewer#2 to the modifications in the text highlighted in the responses #1 and #2 to reviewer#1 comments, which put more emphasis on side-effects and uncertainties.

The suggestion of reviewer#2 regarding the splitting of our paper into two, as well as the remarks regarding the discussion of local factors, lead us to believe that we may not have explained the context of the study precisely enough. We choose not to split into two contributions because the Goll et al 2021 paper actually presented in detail the biogeochemical potential of CDR from basalt rocks, including local suitability factors, uncertainties on weathering rates and possible technological deployment barriers. Here, we take the next step by building a globally-aggregated model that emulates the response of the spatially-resolved model ORCHIDEE-CNP used by Goll et al 2021, and explore in a scenarized way the possible consequences of accounting for the biological carbon sequestration co-benefits of the enhanced weathering of basalt. We agree with the reviewer that cross-sectoral effects could indeed be viewed as a secondary step in the discussion compared to the impact of the amplified CDR from EW by biological carbon sequestration as opposed to abiotic effect only. However, we could not produce ex-ante scenarios with EW alone, which led us to integrate EW in a coupled climate energy carbon cycle model, which is an interesting and new endeavour, as we hope the reviewer will agree with. Further, we could not compare EW (with and without biological carbon sequestration) without a baseline, and we chose to compare it with BECCS, which is a defensible choice because BECCS, is a central, although controversial, CDR technology of many scenarios reaching a low warming level.

What we gained in understanding of the cross-sectoral effects of EW has to be tempered by the necessarily stylized approach we used, and by the fact that using a global model could not render all the subtle and interesting regional details that were highlighted in Goll et al.

More precisely, we offer the reviewer the following text added in the introduction of the paper to guide the reader to this point L90-106. Furthermore, we have modified the abstract to reduce the emphasis on cross-sectoral effects.

Main text L89-105

We also developed an EW module and coupled it with the carbon cycle module of GET-ACC2, where the dissolution of basalt directly removes atmospheric CO₂, and delivers phosphorus to the soil which stimulates the net primary production (NPP). The increase in terrestrial vegetation carbon storage is the difference between NPP and CO₂ released from heterotrophic respiration, which are resolved in ACC2 on a global-annual-mean basis. We emulated the stimulation of NPP through the release of phosphorus from basalt performed by ORCHIDEE-CNP⁴², a global biosphere model that resolves the phosphorus cycle, in order to quantify the geochemical (abiotic) and biotic CDR from basalt applied to forest ecosystems. The spatial heterogeneity of the response of ecosystems to basalt application and the local factors driving additional biological carbon sequestration have been discussed in Goll et al. (2021)⁴³. The biotic CDR was found to be highly variable across regions, strongly dependent on ecosystem type, and most effective where the natural background phosphorus availability was insufficient for plants to benefit from increasing atmospheric CO₂ and warming, notably in tropical and boreal forests⁴³. It should however be noted that this spatial heterogeneity is only implicitly represented in the emulator (see Methods), and that we only accounted for the phosphorus fertilisation effect on plants, not other effects of basalt weathering on soil microbes and soil biota, or other biotic effects such as interactions with the nitrogen cycle, plant health and resistance to pathogens¹¹.

Abstract

Carbon dioxide removal (CDR) is deemed necessary to attain the Paris Agreement's climate objectives. While bioenergy with carbon capture and storage (BECCS) has generated substantial attention, sustainability concerns have led to increased examination of alternative strategies, including enhanced rock weathering (EW). We analyse the role of EW under cost-effective mitigation pathways, by including the CDR potential of basalt applications from silicate weathering (geochemical CDR) and enhanced ecosystem growth and carbon storage in response to phosphorus released by basalt (biotic CDR). Using an integrated carbon cycle, climate and energy system model, we show that the application of basalt to forests could triple the level of carbon sequestration induced by EW compared to an application restricted to croplands. EW also reduces the costs of achieving the Paris Agreement targets as well as the reliance on BECCS. Further understanding requires improved knowledge of weathering rates and basalt side-effects through field testing.

Comment #2

An important information derived from the modeling would be, for example, the CO₂ prices at which the EW on natural lands or crops land turn competitive. A model is a useful tool for this information because the costs depend on various system wide parameters that are not independent from the climate policy, such as the carbon intensity of the energy carriers used for the EW activity. A partial analysis that sets these parameters constant and only varies them independently, misses the point of the transformative nature of a global mitigation strategy and the resulting transformation pathway. The information is

available, but has not been used to the degree that it should be used. It would be interesting to understand what the determinants (or influencing factors) of these break-even CO₂ prices are.

Response #2

We agree with reviewer#2 about the interest of analysing break-even CO₂ prices. We had incorporated this analysis into the supplementary material, but we opted not to include it in the main text due to its relatively higher complexity compared to the rest of the paper and due to length limitations. This decision was made to ensure clarity and accessibility for a broader audience. However, we have offered to the reviewer a third way for the revision: we added a more detailed analysis of the costs and break-even CO₂ prices in the supplementary materials (see section 1.3 of the SM) and the results remain in the main text L127-162 (see below).

Main text L126-164

Under each climate target case, we assessed the following three CDR portfolios: i) BECCS only (No EW), ii) BECCS with EW on croplands only (EW on CL), and iii) BECCS with EW on croplands and forest areas (EW on CL+FA). In the latter, we assumed that basalt could be spread over forest areas, yet with a significant energy and cost penalty compared to croplands. The cost-effective magnitude of CDR from EW deployment is very contrasted across climate target scenarios (**Fig. 2**): EW is more used to achieve 1.5°C than to achieve 2°C, and it is also used more in high-overshoot than in medium and no-overshoot scenarios. EW is applied when the net present value of future carbon removals, occurring in the years following basalt application and extending over decades for biotic sequestration, outweighs application costs. Cropland application costs range from \$43 to \$132 per ton basalt, increasing at higher application levels due to prioritising accessible fields first, leading to higher transport costs for more remote areas. It corresponds to \$116 to \$242 per ton of CO₂ removed, within the range of existing assessments^{13,14}. Possible co-benefits could increase the cost-effectiveness of EW¹⁵ if they can act as fertilisers, but our model does not include food systems and land use and these effects were not considered in the study.

We assume no absolute limit to the production of basalt dust although its growth rate is limited (to 10-20% per year). The maximum CDR potential by EW on croplands thus depends on the application rate of basalt (15 kg/m²), the area of suitable croplands with sufficiently warm and rainy climate, 7.9 Mkm², i.e. a third of global croplands¹³, and the weathering rate (1% to 26% per year). The maximum cropland CDR is 4.9 GtCO₂ per year, consistent with other estimates assuming unlimited basalt supply^{14,16}. Here, the EW application only on croplands approaches its full potential in the 1.5°C with high overshoot scenario with an annual CDR peak of 4.4 GtCO₂ per year and a cumulative removal of 173 GtCO₂.

Forest application is more expensive, with costs varying from \$146 to \$364 per ton of basalt, hinging primarily on the expenses associated with airborne application, and on carbon price-sensitive energy costs, constituting 20-40% of the total. However, the phosphorus effect enhances CO₂ removal efficiency, resulting in substantially reduced removal costs of \$20-\$166 per ton CO₂, especially at low application levels (see SM for a detailed analysis of removal costs). Allowing basalt application in forests reduces the carbon price threshold above which EW becomes cost-efficient and increases the CDR potential in two ways: by expanding the area for basalt application, increasing the geochemical CDR potential, and by enabling biotic CDR in forests. As a consequence, the EW-induced CDR is

almost tripled when basalt can be applied on forests, with a peak of 12.4 GtCO₂ per year and 446 GtCO₂ cumulatively in the 1.5°C with high overshoot scenario. The relative contributions of geochemical and biotic removals vary depending on scenarios; for example, in the 1.5°C with high overshoot scenario, the increase in geochemical CDR due to the additional area available is more pronounced than in other scenarios because cropland application is at full potential. Furthermore, the share of the biotic CDR is proportionally lower at high application levels because the phosphorus stimulation of forest production gradually saturates, thereby limiting the biotic CDR potential. This limit explains why biotic CDR by EW varies less than geochemical CDR by EW among different scenarios.

Comment #3

Also, the reader is very much interested into the local side effects of adding the basalt rock powder to natural lands (such as forests). What are the positive side effects and also what might be risks. In my opinion this is at this stage, more important than the side effects on food prices via bioenergy markets.

Response #3

We agree and have added a more extensive discussion of possible local side-effects in the discussion section (see L343, and also our response to Reviewer 1 Comment #2). Related to local side effects is the uncertainty of the weathering rates. Here, following the comments and advice of Reviewer 1, we better discuss the influence of this source of uncertainty on our results (see L297 and our reply to reviewer 1).

Main text, L343-372

Potential impacts of EW on human health or ecosystems^{1,2}, possible scaling constraints on basalt supply, or lower-than-expected geochemical CDR from incomplete basalt weathering are not considered in the model, which may both limit the sustainability and scalability of basalt soil amendment and restrict EW efficiency. For instance, the needed basalt extraction in the 1.5°C with high overshoot case reaches 46 Gt/year (Fig. S1), which is half of the current global material footprint and ten times the global cement production today, potentially having a large ecological and societal impact. The dust pollution associated with the aerial application of finely milled basalt could lead to silicosis and other respiratory diseases³, and must therefore be prevented, for example by mixing the dust with water to form aggregates or by pelletisation⁴. The release of metals in basalts causing toxicity for humans must also be avoided in agricultural settings by choosing carefully the right material⁵. Basalt dust potential impacts on tree canopy, possibly blocking leaf stomata and reducing tree growth^{3,6} as well as potential impacts on riverine chemistry⁷ must also be anticipated. The application of basalt in forests could alter soil geochemistry for centuries, possibly disrupting natural systems and impacting the composition of plant communities⁸. Wisely exploited, these geochemistry side-effects could serve as an additional tool for biotic CDR in addition to phosphorus fertilisation, as observed in an acid-rain impacted forest where the release of calcium through weathering of added silicate led to a biotic CDR of 3.2-3.5 tCO₂ per ton of wollastonite applied⁴. Ultimately, biotic effects may either offset the net

carbon removal in the case of soil carbon leaching to rivers⁹, or enhance it by increasing soil carbon sequestration^{10,11}. More experiments are therefore required to explore the side-effects of enhanced weathering particularly as rock material cannot be removed from the soil after its application.

At face value, a life cycle analysis comparing EW with other mitigation technologies showed that EW has the advantage to use less land than BECCS or afforestation, less energy than for direct air capture, and less water than for those three technologies¹². The application of basalt in forests is therefore a promising method for mitigating climate change, but it requires the deployment of an appropriate regulatory framework, to ensure that EW helps ecosystems sequester more carbon while minimising adverse side-effects. Furthermore, even if we explored uncertainties as comprehensively as possible, the true uncertainties cannot be wholly captured inherently and certain classes of uncertainties cannot be assessed via quantitative means, indicating a need for careful interpretation and dissemination of our results for stakeholders.

Main text, L304-322

Weathering rates in soils remain highly uncertain^{17,20,21} (see **Methods**). The weathering rates used in modelling studies are calibrated on laboratory experiments^{13,14,17-19}. Field or pot conditions experiments generally provide lower estimates than laboratory experiments^{1,20,22,23}, because the weathering rate depends on complex interplay of soil pH, temperature, hydrological conditions, and biological activity^{11,23}. Soil column²² and mesocosm²⁰ experiments have suggested surface-normalised weathering rates respectively two and three orders of magnitude slower than those used in Strefler et al. 2018¹³, which defines the high range of the weathering rates used in the present work (25% per year). However, field²⁴⁻²⁷ and forest⁴ studies have shown promising CDR rates corresponding to weathering rates exceeding this range. For instance, ref²⁴ reports a $16 \pm 6\%$ loss of cations from basalt applied on agricultural crops over 4 years. This corresponds to a mean weathering rate of 3-6% per year. As they report a grain size of 267 μm , 20 μm -sized grains could weather around ten times faster, because weathering speed scales with the reactive surface. The wide variations across experiments indicate that the weathering rate is a critical source of uncertainty. As shown in **Figure 6(e-l)**, the lower the weathering rate, the less basalt is applied and the less carbon is captured. For weathering rates below 1% per year, EW can become a viable cost-effective option only if basalt is applied on natural areas, because the supply of even very low quantities of phosphorus to phosphorus-depleted soils yields a significant biotic CDR. Thus, the high efficiency of basalt application over forest is maintained even with low weathering rates (see also the sensitivity results using weathering rates of 1% and 25% in SI 1.1.1).

Specific Comments

Comment #4

The notion that GET is a perfect foresight model is not sufficient (page 3). It must be clarified, which carbon fluxes are endogenously considered and incentivized in the policy framework. This is important because of the indirect effect of biotic carbon removal caused by the phosphorus fertilization. For the deployment of the option it is important whether the indirect effect is accounted and the corresponding revenue flow is fully internalized into the calculation that balances revenues to costs. In case the indirect is internalized it is also important to understand whether this indirect effect is considered via an average value or whether it is specific to individual pieces of land that are heterogenous. The authors are required to clarify these points and seriously consider to undertake an uncertainty analysis.

Response #4

In GET-ACC2, the perfect knowledge and foresight is assumed to extend to all physical fluxes of EW, and the phosphorus effect is internalised in the calculation.

More precisely, the biotic ‘indirect effect’ CDR, which is the net increase of land carbon stock due to phosphorus fertilisation, is endogenously optimised: the equations that related this fertilisation effect to the resulting temperature change, the equations that describe the energy system and the basalt application are solved simultaneously, iteratively, until an optimal solution is found. No revenue flows are explicitly considered in the model, Carbon fluxes related to afforestation and deforestation are not optimised in the model. We have clarified this L409 in the method section.

Main text L409-414

GET-ACC2 is fully coupled and optimised with perfect foresight, therefore the biotic CDR, which is the net increase of land carbon stock due to phosphorus fertilisation, the geochemical CDR, the basalt application, the energy system and the climate system are optimised simultaneously, reaching the global least-cost solution that meets the temperature constraint. No revenue flows are explicitly considered in the model. Carbon fluxes related to afforestation and deforestation are not optimised in the model.

Comment #5

It is not clear to me what the regional resolution of the GET model is. The supplement only says that there is a multi-region version available, but it does not say whether this version has been used. For the analysis at hand a multi-regional model would be the appropriate tool due to the geographic heterogeneity and locational specificity of the EW deployment.

Response #5

The GET model we have used is globally aggregated within a single region, although a regional analysis would be interesting to explore the locational specificity of the EW deployment. We do not have access to the regional version of the model. We have made this point clearer in the supplements (section 3.1). Even a regional version of GET would not be able to resolve the fine-scale heterogeneities of local ecosystems' responses to basalt addition within a region. Also, as we hard-link ACC2, GET and an emulator of ORCHIDEE-CNP using a version of GET would increase the running time considerably, without adding much insights.

Comment #6

Is the bioenergy supply related to GHG emissions? A short note and a reference to the literature would be very useful. In case the authors assume full carbon neutrality the authors are requested to justify this assumption and, if need be, to adjust the model assumption.

Response #6

In short, we do not include the GHG emissions used to obtain the mechanical energy for the cultivation, management and harvesting of the bioenergy crops. In the GET model, the exogenous energy demand pathways are assumed to take account of the use of energy to produce energy (e.g., the extraction of materials used to produce renewable energy). The low density of the biomass and the additional transport requirements are accounted for through a financial cost but not through an energy one. Secondly, the extension of cultivated lands to increase the biomass supply could affect the vegetation and soil carbon stocks, and N₂O can be emitted due to added fertilizers, possibly increasing land-use and land-use change emissions^{44,45}.

The variation in soil carbon stock from BECCs can be positive or negative depending on the type of previous land use and the type of BECCS crops⁴⁶. Biophysical factors can contribute to global temperature effects, so the net climate effect of bioenergy depends on the way and the regions where it is deployed. As a consequence, the governance and regulation of the land-use-sector is critical to limit land use change emissions associated with bioenergy production, notably induced direct and indirect land use change⁴⁷. Here, consistently with the perfect information, perfect foresight, optimising paradigm of the model, we initially assumed that bioenergy crop areas are chosen wisely in order to minimise land-use change emissions and we assumed the bioenergy emission factor to equal zero. However, a recent modelling study⁴⁷ have estimated that with a uniform carbon price scheme (a policy framework similar to the intertemporal optimization assumed in GET-ACC2), the average land carbon losses amounted to 5 kgCO₂ per GJ of primary energy (12 kgCO₂/GJ of biofuel, with a conversion efficiency of 41%). Since the bioenergy supply potential of the model is 210-310 EJ per year, it would represent 1-1.5 GtCO₂ per year at most. We have modified the model in consequence. This is clarified in Supplement section 3.2.2. Furthermore, the N₂O emissions from bioenergy supply are included with an emission factor of 0.01MtN/EJ.

Comment #7

Why is the CO₂ price decreasing after 2090 with a remarkable peak in the cases with high overshoot (Figure 3)? The time profile of the prices is also reflected in the emissions trajectory (Figure 2) with the decreasing emissions. These changes are remarkable and seem not to follow a specific argument, but are due to numerical issues.

Response #7

Carbon prices are derived from the dual solution of the optimization and not an exogenous assumption. The carbon price decrease after 2090 in overshoot scenarios is due to the combination of three factors: the climate constraint effective only from 2100 onwards, the inertia of the climate system, and the 10-years time-step of the energy system model GET. Until 2090, carbon prices increase because of the need to reduce temperature through negative emissions. From 2100 onwards, the temperature has reached the target and the model only needs to stabilise it, which requires less net negative emissions and therefore lower carbon prices than in 2090. Carbon prices between 2090 and 2100 are linearly interpolated between GET-timesteps, which explains that the decline starts after 2090.

Comment #8

How is the rock powder spread over the forest areas? The authors state that the phosphor effect saturates (page 4). This suggests that the rock powder is spread evenly across the forest areas, rather than square km by square km. The difference is that the logistic effort varies between both approaches. Obviously, the even spreading of the rock powder would be better to benefit from the phosphor stimulation of NPP, however, it implicitly assumes a much larger logistical effort that is not represented. At least this is how I understand the implementation, the results and the conclusions drawn on that basis. The SI material is not very useful to understand how the spreading is implemented into the GET model and how it relates to

Response #8

Thank you for this comment. We have changed the model to account for the increased cost of evenly spreading basalt powder. In the previous version, the surface of application was constant during each simulation, and at each time step basalt was applied everywhere simultaneously. On the one hand, this might have been underestimating the logistic effort. On the other, it means applying basalt several times over the same area, which is not optimal with regard to the saturation of the phosphorus effect in some areas.

In the new version, the global area is split into a given number of land response classes based on the NPP response to basalt addition (5 in the current setting, with a priority of use for the most responsive classes, see L459-480 & L540-569). The evolution of basalt and phosphorus addition is tracked independently in each land response class. Basalt remains applied evenly in a given class, but the cost and energy intensity of application in the revised model depends on the time taken to apply 1 ton of rock,

which is inversely proportional to the desired application rate of rock in kg/m² over the fixed area of a given land response class. As a consequence, in the new model, basalt is applied successively in the priority order of different land response classes -because of the long-term effect of phosphorus- instead of being applied several times in the same class, see supplementary section SI 1.1.3. We assume that each land response class has the same accessibility and costs for aircraft deployment infrastructure. The time it takes to apply a fixed amount of basalt depends only on the application rate and is independent of the class considered. This is clarified in the revised supplementary section 1.2.2.

Main text, L459-480

Global forests are divided into five land response classes based on the net primary productivity (NPP) response to basalt addition. Basalt is applied evenly in a given class, but the cost and energy intensity of application depends on the time taken to apply 1 ton of rock, which is inversely proportional to the desired application rate of rock in kg/m² over the fixed area of a given land response class.

We considered that small agricultural aeroplanes such as the AirTractor 802 (AT802), which can be equipped with a dust spreader, could be used to spread basalt dust. This kind of aircraft is commonly used to spread limestone⁶⁹⁻⁷¹ although issues with rock discharge have been reported, due to the wide range of the particle size distribution⁶⁹. Details of rock dust discharge are beyond the scope of this analysis, and more research would be needed on how to spread large quantities of basalt dust by air. An AT802 burns 330 litres of kerosene per hour, flies at 306 km/h, can carry 4.3t of rocks (we assume that rock dust can be spread without mixing it with water, either as free-flowing particles or as pellets) and costs USD 1.8 million. Using an open-source map of airports, we estimate that the mean distance per flight ranges between 160 and 240 km. If one adds 10 minutes for spreading operations, the average flight should last between 41 and 57 minutes, lasting longer if the application rate (in kg/m²) is lower, and thereby increasing the application cost per ton of rock. This represents an energy use of 1.8 to 2.5 GJ/t_{rock} for a spreading duration of 10 minutes, but it could virtually be infinite for infinitely low application rates. Assuming 20 minutes of ground operations per flight, 10 hours of use per day, five days out of seven, the capacity cost for spreading one ton per year is \$170-210. Including ground operation and maintenance, pilot fees, insurance and housing (see SI 1.3), the non-energy cost per ton applied in forests is \$110-170, and total costs (including energy costs) reach \$142-355 per ton of basalt depending notably on the energy prices. We do not consider the non-CO₂ climate effect of aerial application⁷², nor those of diesel combustion⁷³.

Main text, L557-586

In the spatially explicit land surface model ORCHIDEE-CNP, the increase of NPP due to phosphorus release depends on the soil, biome and climate and saturates with increasing basalt additions. Application pixels are ranked according to their NPP stimulation from high to low, and grouped in M compartments of areas a_i . In the current setting, M=5 (more details on compartments in the SM). A function of the rock application rate $c_{B,i}$ [Gt.Mkm⁻²] is used to fit the mean NPP response in each compartment i during the forty years that follow basalt application, $\delta\bar{NPP}_i$ (E.5). These classes are an

implicit representation of the spatial heterogeneity of the response of forest ecosystems to phosphorus addition.

$$\delta N\bar{P}P_i = \delta N\bar{P}P_{i,max} \left(1 - e^{-\alpha_i c_{B,i}}\right) \quad (E.5)$$

The emulator is based on the following assumptions: the increase in NPP in compartment i responds to the increase $\delta c_{p,i}$ in soil phosphorus concentration [Gt.Mkm⁻²], which is proportional to the application rate of basalt, and decreases over time (E.6).

$$\delta NPP_i(t) = \delta NPP_{i,max} \left(1 - e^{-\alpha_i \delta c_{p,i}(t)}\right) \quad (E.6)$$

The dynamic evolution of the soil phosphorus concentration $\delta c_{p,i}$ is designed to reproduce the results of ORCHIDEE-CNP. It is modelled with an auxiliary pool of phosphorus which is unavailable to plants, exchanges phosphorus with the soil concentration with exchange times $\tau_{p,i}$ and $\tau_{u,i}$, and is leached to inland waters with a time $\tau_{l,i}$. Noting B_i the undissolved basalt in compartment i , and $\delta u_{p,i}$ the concentration of unavailable phosphorus, we calibrate the exchange times on the ORCHIDEE-CNP outputs using the following system of equations (see SM for more details on the calibration procedure).

$$\frac{d\delta c_{p,i}}{dt} = \frac{\lambda w_r B_i}{a_{B,i}} - \frac{\delta c_{p,i}}{\tau_{p,i}} + \frac{\delta u_{p,i}}{\tau_{u,i}} \quad (E.7a)$$

$$\frac{d\delta u_{p,i}}{dt} = \frac{\delta c_{p,i}}{\tau_{p,i}} - \frac{\delta u_{p,i}}{\tau_{u,i}} - \frac{\delta u_{p,i}}{\tau_{l,i}} \quad (E.7b)$$

Figure S.6 shows the comparison of the emulator with ORCHIDEE-CNP data. Finally, the total NPP increase is the sum of the increase over all compartments (E.8)

$$\delta NPP(t) = \sum_i \delta NPP_i(t) \quad (E.8)$$

Comment #9

In that context, I sense the following comment is appropriately placed. I do not understand Figure S1.3.4. What is the y-axis telling me? Is land somehow sorted to some criterion? and why these uncommon patches? What does the width of these patches tell the reader? I simply do not get what this figure tries to communicate.

Response #9

As we have changed the model, the application surface is not optimised in the same way as before, and we therefore removed this figure.

Comment #10

It is not clear to me how the climate model responds to CO₂ emissions. Investigating Figures 2 and 3, it appears to me that the global mean temperature remains constant at annual CO₂ emissions of around 10 GtCO₂. There is some variation about the emissions that are consistent with a constant global mean temperature, but 10 GtCO₂/yr seem above that level. The Supplement is not saying something about this point. The authors have to clarify and, if need be, adjust the model parameters.

Response #10

Thank you for offering us the possibility to clarify this point. The land-use change emissions are exogenous and were not included to the figure 2 (brown bars = land-use). With land-use emissions, assumed to be negative after 2070, the temperature remains constant at annual net emissions of 5 GtCO₂/year (not 10 GtCO₂/year) which is in the uncertainty range of emissions levels consistent with a constant global mean temperature. This is clarified in the caption of Fig 2.

is limited to 2°C. **2°C with high overshoot:** the temperature change is limited to 2°C after 2100. 'EW: supply chain' (in orange) represents the emissions from fossil fuels used to apply EW. The **red lines** represent the net CO₂ emissions.

Comment #11

In that context the following point is also important. What are the levels of emissions of non-CO₂ GHG? The supplement does not say anything about it. This is important to understand the CO₂ emissions and the temperature change. This information should be reported in the Supplement.

Response #11

Thank you for this comment. Non-CO₂ emissions are important (and included) in the study. We have added the emission pathways figures for non-CO₂ GHG in supplementary figure 3.3.2 and mention in the Method section L400 that non-CO₂ emissions are included, endogenous for CH₄ and N₂O and exogenous for other forcers based on SSP scenarios corresponding to each target. Furthermore, we have added the marginal abatement cost curves used to compute the abatement of CH₄ and N₂O in the SI 3.2.3 .

Main text L400-411

The anthropogenic CH₄, N₂O emissions and net energy-related CO₂ emissions are calculated in GET7.1 and are transferred to ACC2 for temperature calculations. The temperature calculations also use exogenous non-energy-related CO₂ emissions and other greenhouse gas and pollutant emissions, which are assumed to follow SSP1-1.9 and SSP1-2.6 for the 1.5°C and 2°C target cases, respectively. In ACC2, a box model represents oceanic and terrestrial carbon cycles. The ocean CO₂ uptake is represented by a four-layer box model, with the uppermost representing the atmosphere and ocean mixed layer and the others the ocean's inorganic carbon storage capacity, and the land CO₂ uptake model consists of four reservoirs connected with the atmosphere. The atmospheric concentrations of multiple greenhouse gases respond dynamically to their emissions and removals e.g. chemical sinks. The resulting radiative forcing is an input to a heat diffusion model, which further calculates the global temperature. The temperature change in turn affects the carbon cycle through soil respiration and ocean-atmosphere carbon flux. ACC2 is comprehensively described in ref^{s1}.

Comment #12

The overshoot qualified as “low” in the 1.5°C case is not small given the IPCC classification. The C1 category allow for a 0.1°C overshoot, while there it is twice that temperature difference. The authors may want to revisit the scenario definition.

Response #12

We agree and have changed the scenario name for “1.5°C case with a medium overshoot”.

Comment #13

The reaction speed of weathering very much depends on the grain size, which in turn has a strong effect on the electricity required for the grinding. This has been shown in the paper by Strefler et al. (2018, Figure 1). The authors should depict the assumptions against this backdrop so that the reader better understands how the assumptions compare. Providing a range of 0.06-0.6EJ/Gt is too little information (Sec 1.1 in Supplement).

Response #13

The range used is 0.07 to 0.6 GJ/t. To define this range, we used a triangular distribution with a central value of 0.2 GJ/t for grinding the grains at a size of 20µm which corresponds to the SI of Strefler et al. 2018 (table J-5). We have chosen this grain size for consistency with Goll et al, 2021, which used the same size in their simulations: a grain size of 20µm corresponds to a near-complete dissolution within a decade, which is the timestep of the model, and is optimal for carbon prices around 270\$/tCO₂ based of Strefler et al. 2018, which is close to the carbon prices obtained in the model. Although it could be possible to let the model optimise the grain size, if the grain size evolves with time the different vintages of basalt must be tracked separately, increasing the computational burden. We have modified the main text accordingly in L420-422

Main text L423-425

The electricity for grinding basalt is 0.2 EJ/Gt (central value) and ranges from 0.07 to 0.6 EJ/Gt in the uncertainty analysis, **which is the range provided in ref¹³ for grain size of 20µm.**

Comment #14

The weathering process involves the reaction of CO₂ dissolved in rain water. Thus, precipitation is an important natural parameter for the speed of weathering. In forest areas it is also important how much rain water gets to the soil where the rock powder will be located. In relatively mature forests with frequent, but

small rain falls, the share of rain water that hits the ground will be relatively small. I did not find that the authors have represented this point in their parameterization. The authors are required to clarify.

Response #14

In the ORCHIDEE model (with phosphorus fertilization) which is emulated in this study, no hydrological effect is considered for the abiotic weathering processes which depend only on grain size and temperature, but hydrology affecting plant NPP is included in the indirect phosphorus fertilisation biotic pathway. This is explained in L470-474 of the manuscript.

Main text L472-476

Once applied to soils, basalt is assumed to have a constant dissolution rate, referred to in this study as the weathering rate. This simplistic approach reflects current understanding and data availability to parameterize weathering rates, and does not account for certain phenomena, such as the reduction of reactive surface over time, as well as the soils, plants and hydrological processes that can potentially influence weathering.

Comment #15

What are the variable costs of spreading by airplane? The authors only mention the maintenance costs of 5% of the initial investment costs. However, an air plane needs a start and landing base, a ground crew and a pilot. Are these costs accounted? What about the wages for truck drivers? (This refers to the techno-economic parameters in the Supplement).

Response #15

The variable costs of spreading by airplane are \$65-113 per ton of rock, including operation and maintenance costs and pilot fees. Adding the housing and cost of buying and insuring the aircraft, the costs reach \$84-137 per ton. The assessment of the variable costs of spreading by aircraft was incomplete in the previous version, as we did not include pilot fees nor insurance costs, and amounted only to \$20-25 per ton. We have updated them based on techno-economic assessments of agricultural aviation⁵⁴, they are assumed to be globally uniform (see supplementary 1.2.2). It appears that the pilot wages as well as the ground operations and maintenance represent a significant share of the costs. The truck drivers wages are accounted for within the “financial costs” mentioned in the SI: they are derived from the OPEX for road freight listed in ref⁹.

Comment #16

The authors talk rather generally about the ranking of the CDR quantities as differences between scenarios are more or less. However, Figure 2 clearly shows that for the same long-term temperature target there is an intersection of CDR deployment over time, because the low overshoot case requires stronger-near-term CDR deployment while the high overshoot case delays CDR deployment

Response #16

We agree and already discussed the delay of abatement measures in the part dedicated to policy costs. We have modified the text L181 to mention this effect.

Main text L181-183

Abatement, including EW deployment (Fig. 2), is mainly delayed in OS scenarios: in our forward-looking optimisation model, the greater the future opportunity for negative emissions is, the lower the near-term abatement and the discounted costs.

Comment #17

The strong variations in the ratio between biotic and abiotic cumulative carbon removal in figure 2 (top row) suggests that there is a certain and limited potential of biotic carbon removal that is largely independent of the amount of abiotic. It is important to understand the coupling between the biotic and abiotic carbon removal. The authors are required to investigate this somewhat more.

Response #17

Thank you for this comment. The coupling between biotic and abiotic carbon removal has been investigated in Goll et al, 2021 (see Figures 1, 2 and 3). The potential of biotic carbon removal depends on the regions, basalt application, other plant nutrients, climate change and atmospheric CO₂ concentration. While most modelling assumes there is no upper limit to abiotic CDR, practically there might be important constraints on how much rock dust can be applied.

We have emphasised this important point on L157 of the revised manuscript:

Main text L157-162

The relative contributions of geochemical and biotic removals vary depending on scenarios; for example, in the 1.5°C with high overshoot scenario, the increase in geochemical CDR due to the additional area available is more pronounced than in other scenarios because cropland application is at full potential. Furthermore, the share of the biotic CDR is proportionally lower at high application levels because the phosphorus stimulation of forest production gradually saturates, thereby limiting the

biotic CDR potential. This limit explains why biotic CDR by EW varies less than geochemical CDR by EW among different scenarios.

Comment #18

The uncertainty range in Figure 3 for the global mean temperature anomaly is small. This is because the ECS is not varied. However, it makes sense to assume the same uncertainties for all four scenarios.

Response #18

As explained in the text, large ECS values make the most ambitious climate targets out of reach, particularly in the near term. ECS is varied in the model (see L99) with a log-normal distribution from values of 2°C and 4.7^{o55}.

Comment #19

Figure 1.3.5 in the supplement is confusing. Why are the energy costs higher than the marginal costs? How can the marginal revenue be higher than the carbon price (1.5K, low overshoot case in 2030)? What is the share representing (bottom, right hand side)? What are the ranges representing. The 25-75% is too little info. How can the solid lines be outside the ranges?

Response #19

The energy costs curve represents the share of the rock application costs as a percentage of total application costs, and is associated with the right y-axis, which scales from 0 to 1, hence it is not larger than the marginal cost.

The marginal revenue per ton of rock can be higher by the carbon price for two reasons : First, because a ton of rock sequesters more than a ton of CO₂ on forests. Second, because CO₂ removals are delayed compared to the time of application, meaning that the net present value of future benefits will be higher if carbon prices increase very fast. The range represents the distribution of the benefits and costs over the sample of parameters. We have modified the figure to represent both the 25-75% and the 5-95% ranges. Solid lines were outside the range because we displayed the means which can be strongly affected by outliers. We have changed the figure to display the medians.

SI 1.3.2 figure 1.3.5

Fig 1.3.5 Marginal revenues and costs of basalt application in the different climate policy scenarios. Line: median. Shaded areas: 5-95% (light) and 25-75% ranges (dark). The marginal revenue of basalt application is defined as the nominal benefit associated with the CDR resulting from the application of one ton of rock, if no financial or energy expenditure is required. The marginal cost of basalt application is the nominal cost of applying one ton of rock without any resulting CDR. The energy cost is the nominal price of the energy that would be used to apply one ton of basalt. **The dotted lines** represent the ratio of energy cost over total basalt application cost (**right y-axis**).

Comment #20

Smaller Points

Page 3: “The increase in terrestrial carbon storage is” I guess the word vegetation is missing here.

Response #20

We agree and thank reviewer#2 for spotting this.

Reviewer #3 (Remarks to the Author):

General Comments

Comment #1

The paper presents the results of calculations that show that adding basalt powder to forests could increase the efficiency of enhanced weathering as a method for carbon removal from the atmosphere. Using their integrated carbon, climate and energy system model they propose that this approach could remove 3 times the amount of carbon when compared to adding the same amount of basalt to agricultural lands. The topic is timely and I found the manuscript interesting if not entirely convincing. Adding basalt to forests has been proposed before (see for example de Oliveira Garcia, Wagner, et al. "Impacts of enhanced weathering on biomass production for negative emission technologies and soil hydrology." *Biogeosciences* 17.7 (2020): 2107-2133.) so that the main contribution of this article is an attempt to calculate this potential impact.

Response #1

We thank reviewer#3 for the review of our work. The aim of the article is indeed to assess the impact on mitigation pathways the application of basalts to forests.

Comment #2

However, the analysis suffers from a number of limitations:

The method of calculation involves a complex global model, which the authors present no attempt to calibrate or verify. While the authors do perform an uncertainty analysis using a Monte Carlo method, the overall reliability of the approach remains unclear.

Response #2

In order to assess the impacts of EW on mitigation pathways, we need to use a coupled model with three submodules: the climate model ACC2, the emulator of ORCHIDEE-CNP for representing EW with biological carbon sequestration , and the energy system model GET.

- 1) The climate model ACC2^{52,53,56-61} is calibrated on historical data⁵³, and its temperature response to greenhouse gas emissions has been validated against similar reduced-complexity climate models⁶². The carbon sequestration following an increase in net primary production in ACC2 has been validated against CMIP6 models in the context of an increase in net primary production due to CO2 fertilisation⁶⁰, but not phosphorus fertilisation which is a process absent from all CMIP6 models.
- 2) The land-surface model ORCHIDEE-CNP is well evaluated from site to global scale including nutrient leaching from terrestrial soils and the effects of elevated CO2 on primary productivity and land carbon storage^{42,63-65}. Further, the response of aboveground productivity to mineral P fertiliser addition compares well to observation-based estimates⁴³. The emulator that reproduces the NPP response of ORCHIDEE-CNP to basalt addition is described in the supplementary material section 1.1.2. The description of the emulator calibration in the supplementary information was completely rewritten. We have made the calibration process clearer and more transparent in the revised version, notably by complementing the step-by-step description of the calibration procedure with a Jupyter Notebook containing the code used for the calibration. We acknowledge the fact that the emulator, relying on a reduced set of assumptions for extrapolating the behaviour of the ORCHIDEE-CNP model, is necessarily a simplification of the original model.
- 3) The energy system model GET was developed for providing least-cost scenarios of energy transition⁶⁶⁻⁷¹. The initial state of the energy system is calibrated on IEA data regarding energy flows and production capacities, and technology costs are taken from other energy system models. This kind of forward-looking model is hard to validate⁷² because it does not intend to make predictions about the future, but rather to provide quantitatively self-consistent energy scenarios, in an idealised world where a central planner with perfect foresight coordinates the energy transition to achieve climate targets at the lowest cost. The resulting model is indeed complex and cannot be easily validated on existing data.

This choice of models and their suitability and previous peer reviewed publication history has been made clearer in Supplements 3.1.3

SI 3.1.2

The climate model ACC2^{52,53,56-61} is calibrated on historical data⁵³, and its temperature response to greenhouse gas emissions has been validated against similar reduced-complexity climate models⁶². The carbon sequestration following an increase in net primary production in ACC2 has been validated against CMIP6 models in the context of an increase in net primary production due to CO2 fertilisation⁶⁰, but not phosphorus fertilisation which is a process (to date) absent from all CMIP6 models.

The land-surface model ORCHIDEE-CNP was well evaluated from site to global scale including nutrient leaching from terrestrial soils and the effects of elevated CO2 on primary productivity and land carbon storage^{42,63-65}. Further, the response of aboveground productivity to

mineral P fertiliser addition was compared with observation-based estimates⁴³. The emulator that reproduces the NPP response of ORCHIDEE-CNP to basalt addition, relying on a reduced set of assumptions for extrapolating the behaviour of the complex ORCHIDEE-CNP model, is necessarily a simplification of the original model but we checked that it reproduces faithfully the emerging response of the complex model.

The energy system model GET was developed for providing least-cost scenarios of energy transition⁶⁶⁻⁷¹. The initial state of the energy system is calibrated on IEA data regarding energy flows and production capacities, and technology costs are taken from other energy system models. This kind of forward-looking model is hard to validate⁷² because it does not intend to make predictions about the future, but rather to provide quantitatively self-consistent energy scenarios, in an idealised world where a central planner with perfect foresight coordinates the energy transition to achieve climate targets at the lowest cost.

Comment #3

2. *The authors present no analysis of the influence of local factors such as climate, forest type, soil types, or different kinds of basalts that would need to be accounted for in order to reliably calculate the impact of enhanced weathering on both global and local scales. The absence of this analysis means that even if they are correct in some environments, in others there may not be any advantage at all to adding basalt to forests.*

Response #3

Indeed, the focus of the paper is not on the influence of local factors on enhanced weathering. A more detailed analysis of local factors has been conducted in a previous study using the ORCHIDEE-CNP model⁴³.

Comment #4

Besides the addition of phosphorus to soils by enhanced weathering, which is assumed to enhance growth, there is no discussion of the possible additional mechanisms by which CO₂ removal may be enhanced. Also, how would this work in mature forests in which increased nutrient supply may not necessarily translate into enhanced weathering. As such, the model behaves very much like a black box and is not convincing, again, without experimental verification.

Response #4

We thank reviewer#3 for pointing out this omission, and agree that additional mechanisms could play a prominent role and we have therefore added a short discussion of these mechanisms in the revised paper L348-355.

Main text, L348-355

The application of basalt in forests could alter soil geochemistry for centuries, possibly disrupting natural systems and impacting the composition of plant communities⁶². Wisely exploited, these geochemistry side-effects could serve as an additional tool for biotic CDR in addition to phosphorus fertilisation, as observed in an acid-rain impacted forest where the release of calcium through weathering of added silicate led to a biotic CDR of 3.2-3.5 tCO₂ per ton of wollastonite applied⁵⁶. Ultimately, biotic effects may either offset the net carbon removal in the case of soil carbon leaching to rivers⁶³, or enhance it by increasing soil carbon sequestration^{34,64}.

Reviewer#3 also raises the issue of the translation of increased nutrient supply into increased carbon dioxide removal in mature forests. A meta-analysis⁷³ of 652 experiments of P-additions to different types of ecosystems has shown that phosphorus limitation is globally distributed in natural ecosystems (see Figure 1 of ref⁷³), notably in forests, and that P-additions increased aboveground plant production by 34% in average in natural ecosystems, and by 78.5% in forests (CI:45.1-119.6%). Regarding mature forests, they report 23 experiments from 14 publications⁷⁴⁻⁸⁷. 10 experiments⁸¹⁻⁸⁷ out of these 23 report significant increase of aboveground productions, in tropical, subtropical and temperate areas. Furthermore, a recent study has demonstrated an increase of P-availability in 30 year old tropical rubber plantation following a wollastonite amendment, which indicates a potential for increased organic carbon sequestration⁸⁸.

Specific Comments

Comment #5

The authors use so many acronyms that the writing is dense and, at times, incomprehensible. For example, at one point they write: “Global NPP is higher in ACC2 than in ORCHIDEE. Therefore, in order to replicate the magnitude of the increase in NPP from ORCHIDEE, we scale the increase in NPP by the ratio of preindustrial NPP in ACC2 and the NPP in ORCHIDEE for the year 2018 (E.4).” If you read this aloud, which I did, it just sounds like gobbledygook.

Response #5

We acknowledge that in our pursuit of precision we used too many acronyms, to the detriment of clarity. We have tried to lighten the text by removing some acronyms: OS was replaced by overshoot, we reduced the occurrences of “GET-ACC2” (2 occurrences removed) and “NPP” (12 occurrences removed).

Additionally, L525: “Global NPP is higher in ACC2 than in ORCHIDEE. Therefore, in order to replicate the magnitude of the increase in NPP from ORCHIDEE, we scale the increase in NPP by the ratio of preindustrial NPP in ACC2 and the NPP in ORCHIDEE for the year 2018 (E.4).”

Became

“Global NPP is higher in ACC2 than in ORCHIDEE. Therefore, in order to replicate the absolute magnitude of its increase in ORCHIDEE, we scale its increase by the ratio of its respective initial values in the two models (E.4)”

Comment #6

Figure 5 and 6 are too dense, and contain too much information for the average reader to follow or understand.

Response #6

We have reduced the number of panels in figure 5 to reduce its complexity by only showing the 1.5°C with high OS and 2°C cases, and moved the others in the supplements.

We have removed one column of the top panels of figure 6 and removed the standard deviations of the elementary effects to reduce its density.

Main text, Figure 5

Figure 5| Energy use and CO₂ emissions of enhanced weathering per tCO₂ sequestered (mean across the 21st century). (a,c): enhanced weathering is applied on forest areas and croplands. (b,d): enhanced weathering is applied only on croplands. (a,b): 1.5°C case with high overshoot. (c,d): 2°C case without overshoot. **Blue labels:** The energy used to apply EW, for each energy vector, expressed in GJ per ton of CO₂ that is sequestered through EW. **Red labels:** The emissions associated with the use of each primary energy source, expressed in kg of CO₂ emitted per ton of CO₂ captured through EW. Note that the vertical scale is different in each panel.

Main text, Figure 6

Figure 6 | Sensitivity analysis. (a-d): Morris screening: mean variation of the output (columns), when the input (rows) is increased by half of its uncertainty range. The sources of uncertainty assessed are: the weathering rate, the geochemical capture rate, the phosphorus content of basalt, the baseline energy demand, the energy intensity of basalt grinding, the energy intensity of basalt application, the annual bioenergy potential, the efficiency of CCS, the efficiency of other mitigation technologies and the climate sensitivity. The outputs displayed are: ‘EW’, the cumulative CDR from EW when EW is applied on croplands and forests; ‘BECCS’, the cumulative CDR from BECCS when EW is applied on croplands and forests. **(e-h): CDR from EW depending on weathering rate. (i-l): Application of EW depending on weathering rate.** Each dot is a simulation in the sample. Solid line: median. Shaded area: 25-75% range for a given value of the weathering rate. The vertical dotted line shows the mean weathering rate considered in the rest of the paper (13% per year). Additional runs were performed to cover a wider range of weathering rates.

References

1. Buckingham, F. L., Henderson, G. M., Holdship, P. & Renforth, P. Soil core study indicates limited CO₂ removal by enhanced weathering in dry croplands in the UK. *Appl. Geochem.* **147**, 105482 (2022).
2. Dupla, X., Möller, B., Baveye, P. C. & Grand, S. Potential accumulation of toxic trace elements in soils during enhanced rock weathering. *Eur. J. Soil Sci.* **74**, e13343 (2023).
3. Taylor, L. L. *et al.* Enhanced weathering strategies for stabilizing climate and averting ocean acidification. *Nat. Clim. Change* **6**, 402–406 (2016).
4. Taylor, L. L. *et al.* Increased carbon capture by a silicate-treated forested watershed affected by acid deposition. *Biogeosciences* **18**, 169–188 (2021).
5. Cobo, S. *et al.* Sustainable scale-up of negative emissions technologies and practices: where to focus. *Environ. Res. Lett.* **18**, 023001 (2023).
6. Farmer, A. M. The effects of dust on vegetation—a review. *Environ. Pollut.* **79**, 63–75 (1993).
7. Zhang, S. *et al.* River chemistry constraints on the carbon capture potential of surficial enhanced rock weathering. *Limnol. Oceanogr.* **67**, S148–S157 (2022).
8. Vandeginste, V., Lim, C. & Ji, Y. Exploratory Review on Environmental Aspects of Enhanced Weathering as a Carbon Dioxide Removal Method. *Minerals* **14**, 75 (2024).
9. Klemme, A., Rixen, T., Müller, M., Notholt, J. & Warneke, T. Destabilization of carbon in tropical peatlands by enhanced weathering. *Commun. Earth Environ.* **3**, 1–9 (2022).
10. Buss, W., Hasemer, H., Ferguson, S. & Borevitz, J. Stabilisation of soil organic matter with rock dust partially counteracted by plants. *Glob. Change Biol.* **30**, e17052 (2024).
11. Vicca, S. *et al.* Is the climate change mitigation effect of enhanced silicate weathering governed by biological processes? *Glob. Change Biol.* **28**, 711–726 (2022).
12. Eufrazio, R. M. *et al.* Environmental and health impacts of atmospheric CO₂ removal by enhanced

- rock weathering depend on nations' energy mix. *Commun. Earth Environ.* **3**, 106 (2022).
13. Strefler, J., Amann, T., Bauer, N., Kriegler, E. & Hartmann, J. Potential and costs of carbon dioxide removal by enhanced weathering of rocks. *Environ. Res. Lett.* **13**, 034010 (2018).
 14. Beerling, D. J. *et al.* Potential for large-scale CO₂ removal via enhanced rock weathering with croplands. *Nature* **583**, 242–248 (2020).
 15. Lewis, A. L. *et al.* Effects of mineralogy, chemistry and physical properties of basalts on carbon capture potential and plant-nutrient element release via enhanced weathering. *Appl. Geochem.* **132**, 105023 (2021).
 16. Strefler, J. *et al.* Carbon dioxide removal technologies are not born equal. *Environ. Res. Lett.* **16**, 074021 (2021).
 17. Rinder, T. & von Hagke, C. The influence of particle size on the potential of enhanced basalt weathering for carbon dioxide removal - Insights from a regional assessment. *J. Clean. Prod.* **315**, 128178 (2021).
 18. Fuhrman, J. *et al.* Diverse carbon dioxide removal approaches could reduce impacts on the energy–water–land system. *Nat. Clim. Change* **13**, 341–350 (2023).
 19. Renforth, P. The potential of enhanced weathering in the UK. *Int. J. Greenh. Gas Control* **10**, 229–243 (2012).
 20. Amann, T. *et al.* Enhanced Weathering and related element fluxes – a cropland mesocosm approach. *Biogeosciences* **17**, 103–119 (2020).
 21. Cipolla, G., Calabrese, S., Noto, L. V. & Porporato, A. The role of hydrology on enhanced weathering for carbon sequestration I. Modeling rock-dissolution reactions coupled to plant, soil moisture, and carbon dynamics. *Adv. Water Resour.* **154**, 103934 (2021).
 22. Renforth, P., Pogge von Strandmann, P. A. E. & Henderson, G. M. The dissolution of olivine added to soil: Implications for enhanced weathering. *Appl. Geochem.* **61**, 109–118 (2015).
 23. Swoboda, P., Döring, T. F. & Hamer, M. Remineralizing soils? The agricultural usage of silicate rock

- powders: A review. *Sci. Total Environ.* **807**, 150976 (2022).
24. Beerling, D. J. *et al.* Enhanced weathering in the US Corn Belt delivers carbon removal with agronomic benefits. *Proc. Natl. Acad. Sci.* **121**, e2319436121 (2024).
 25. Kantola, I. B. *et al.* Improved net carbon budgets in the US Midwest through direct measured impacts of enhanced weathering. *Glob. Change Biol.* **29**, 7012–7028 (2023).
 26. Ryan, P. C. *et al.* The potential for carbon dioxide removal by enhanced rock weathering in the tropics: An evaluation of Costa Rica. *Sci. Total Environ.* **927**, 172053 (2024).
 27. Wang, F. *et al.* Wollastonite powder application increases rice yield and CO₂ sequestration in a paddy field in Northeast China. *Plant Soil* (2024) doi:10.1007/s11104-024-06570-5.
 28. Vanderkloot, E. & Ryan, P. Quantifying the effect of grain size on weathering of basaltic powders: Implications for negative emission technologies via soil carbon sequestration. *Appl. Geochem.* **155**, 105728 (2023).
 29. Berge, H. F. M. ten *et al.* Olivine Weathering in Soil, and Its Effects on Growth and Nutrient Uptake in Ryegrass (*Lolium perenne* L.): A Pot Experiment. *PLOS ONE* **7**, e42098 (2012).
 30. Dietzen, C., Harrison, R. & Michelsen-Correa, S. Effectiveness of enhanced mineral weathering as a carbon sequestration tool and alternative to agricultural lime: An incubation experiment. *Int. J. Greenh. Gas Control* **74**, 251–258 (2018).
 31. Larkin, C. S. *et al.* Quantification of CO₂ removal in a large-scale enhanced weathering field trial on an oil palm plantation in Sabah, Malaysia. *Front. Clim.* **4**, 959229 (2022).
 32. Haque, F., Santos, R. M. & Chiang, Y. W. CO₂ sequestration by wollastonite-amended agricultural soils – An Ontario field study. *Int. J. Greenh. Gas Control* **97**, 103017 (2020).
 33. Reershemius, T. *et al.* Initial Validation of a Soil-Based Mass-Balance Approach for Empirical Monitoring of Enhanced Rock Weathering Rates. *Environ. Sci. Technol.* **57**, 19497–19507 (2023).
 34. Kelland, M. E. *et al.* Increased yield and CO₂ sequestration potential with the C₄ cereal *Sorghum bicolor* cultivated in basaltic rock dust-amended agricultural soil. *Glob. Change Biol.* **26**, 3658–3676

(2020).

35. West, L. J., Banwart, S. A., Martin, M. V., Kantzas, E. & Beerling, D. J. Making mistakes in estimating the CO₂ sequestration potential of UK croplands with enhanced weathering. *Appl. Geochem.* **151**, 105591 (2023).
36. Buckingham, F. L., Henderson, G. M. & Renforth, P. Response to Comment from West et al. on, “Soil core study indicates limited CO₂ removal by enhanced weathering in dry croplands in the UK”. *Appl. Geochem.* **152**, 105622 (2023).
37. Beerling, D. J. *et al.* Farming with crops and rocks to address global climate, food and soil security. *Nat. Plants* **4**, 138–147 (2018).
38. Kantzas, E. P. *et al.* Substantial carbon drawdown potential from enhanced rock weathering in the United Kingdom. *Nat. Geosci.* **15**, 382–389 (2022).
39. Kantola, I. B., Masters, M. D., Beerling, D. J., Long, S. P. & DeLucia, E. H. Potential of global croplands and bioenergy crops for climate change mitigation through deployment for enhanced weathering. *Biol. Lett.* **13**, 20160714 (2017).
40. Haque, F., Santos, R. M., Dutta, A., Thimmanagari, M. & Chiang, Y. W. Co-Benefits of Wollastonite Weathering in Agriculture: CO₂ Sequestration and Promoted Plant Growth. *ACS Omega* **4**, 1425–1433 (2019).
41. Knapp, W. J. & Tipper, E. T. The efficacy of enhancing carbonate weathering for carbon dioxide sequestration. *Front. Clim.* **4**, (2022).
42. Goll, D. S. *et al.* A representation of the phosphorus cycle for ORCHIDEE (revision 4520). *Geosci. Model Dev.* **10**, 3745–3770 (2017).
43. Goll, D. S. *et al.* Potential CO₂ removal from enhanced weathering by ecosystem responses to powdered rock. *Nat. Geosci.* **14**, 545–549 (2021).
44. Daioglou, V. *et al.* Greenhouse gas emission curves for advanced biofuel supply chains. *Nat. Clim. Change* **7**, 920–924 (2017).

45. Hanssen, S. V. *et al.* The climate change mitigation potential of bioenergy with carbon capture and storage. *Nat. Clim. Change* **10**, 1023–1029 (2020).
46. Wang, J. *et al.* Temperature Changes Induced by Biogeochemical and Biophysical Effects of Bioenergy Crop Cultivation. *Environ. Sci. Technol.* (2023) doi:10.1021/acs.est.2c05253.
47. Merfort, L. *et al.* Bioenergy-induced land-use-change emissions with sectorally fragmented policies. *Nat. Clim. Change* (2023) doi:10.1038/s41558-023-01697-2.
48. M. C. E. Grafton, I. J. Yule, C. E. Davies, R. B. Stewart, & J. R. Jones. Resolving the Agricultural Crushed Limestone Flow Problem from Fixed-Wing Aircraft. *Trans. ASABE* **54**, 769–775 (2011).
49. Bošefa, M. & Šebeň, V. Analysis of the aerial application of fertilizer and dolomitic limestone. *J. For. Sci.* **56**, 47–57 (2018).
50. Clair, T. A. & Hindar, A. Liming for the mitigation of acid rain effects in freshwaters: A review of recent results. *Environ. Rev.* **13**, 91–128 (2005).
51. Fuglestvedt, J., Lund, M. T., Kallbekken, S., Samset, B. H. & Lee, D. S. A “greenhouse gas balance” for aviation in line with the Paris Agreement. *WIREs Clim. Change* **n/a**, e839.
52. Tanaka, K., Lund, M. T., Aamaas, B. & Berntsen, T. Climate effects of non-compliant Volkswagen diesel cars. *Environ. Res. Lett.* **13**, 044020 (2018).
53. Tanaka, K. *et al.* Aggregated Carbon cycle, atmospheric chemistry and climate model (ACC2): description of forward and inverse mode. 14069106 (2007) doi:10.17617/2.994422.
54. Moraes, T. R., Cornago Junior, V. M., Araújo, V. C. R. de, Esperancini, M. S. T. & Antuniassi, U. R. COST OF AERIAL AND GROUND SPRAYINGS AND TECHNOLOGICAL REPLACEMENT POINT: A CASE STUDY IN THE REGION OF MINEIROS, GO, BRAZIL. *Eng. Agric.* **41**, 359–367 (2021).
55. Sherwood, S. C. *et al.* An Assessment of Earth’s Climate Sensitivity Using Multiple Lines of Evidence. *Rev. Geophys.* **58**, (2020).
56. Tanaka, K., Raddatz, T., O’Neill, B. C. & Reick, C. H. Insufficient forcing uncertainty underestimates

- the risk of high climate sensitivity. *Geophys. Res. Lett.* **36**, L16709 (2009).
57. Tanaka, K., Johansson, D. J. A., O'Neill, B. C. & Fuglestedt, J. S. Emission metrics under the 2 °C climate stabilization target. *Clim. Change* **117**, 933–941 (2013).
 58. Tanaka, K. *et al.* Cost-effective implementation of the Paris Agreement using flexible greenhouse gas metrics. (2020).
 59. Tanaka, K. & O'Neill, B. C. The Paris Agreement zero-emissions goal is not always consistent with the 1.5 °C and 2 °C temperature targets. *Nat. Clim. Change* **8**, 319–324 (2018).
 60. Melnikova, I., Ciais, P., Boucher, O. & Tanaka, K. Assessing carbon cycle projections from complex and simple models under SSP scenarios. *Clim. Change* **176**, 168 (2023).
 61. Kvale, K. *et al.* Carbon Dioxide Emission Pathways Avoiding Dangerous Ocean Impacts. *Weather Clim. Soc.* **4**, 212–229 (2012).
 62. Nicholls, Z. R. J. *et al.* Reduced Complexity Model Intercomparison Project Phase 1: introduction and evaluation of global-mean temperature response. *Geosci. Model Dev.* **13**, 5175–5190 (2020).
 63. Goll, D., Joetzjer, E., Huang, M. & Ciais, P. Low Phosphorus Availability Decreases Susceptibility of Tropical Primary Productivity to Droughts. *Geophys. Res. Lett.* **45**, 8231–8240 (2018).
 64. Sun, Y. *et al.* Global evaluation of the nutrient-enabled version of the land surface model ORCHIDEE-CNP v1.2 (r5986). *Geosci. Model Dev.* **14**, 1987–2010 (2021).
 65. Friedlingstein, P. *et al.* Global Carbon Budget 2019. *Earth Syst. Sci. Data* **11**, 1783–1838 (2019).
 66. Azar, C. A., Lindgren, K. & Andersson, B. Hydrogen Or Methanol in the Transportation Sector? (2000).
 67. Azar, C., Lindgren, K., Larson, E. & Möllersten, K. Carbon Capture and Storage From Fossil Fuels and Biomass – Costs and Potential Role in Stabilizing the Atmosphere. *Clim. Change* **74**, 47–79 (2006).
 68. Azar, C. & Lindgren, K. Global energy scenarios meeting stringent CO₂ constraints— cost-effective fuel choices in the transportation sector. *Energy Policy* **16** (2003).

69. Azar, C., Johansson, D. J. A. & Mattsson, N. Meeting global temperature targets—the role of bioenergy with carbon capture and storage. *Environ. Res. Lett.* **8**, 034004 (2013).
70. Hedenus, F., Karlsson, S., Azar, C. & Sprei, F. Cost-effective energy carriers for transport – The role of the energy supply system in a carbon-constrained world. *Int. J. Hydrog. Energy* **35**, 4638–4651 (2010).
71. Johansson, D. J. A., Azar, C., Lehtveer, M. & Peters, G. P. The role of negative carbon emissions in reaching the Paris climate targets: The impact of target formulation in integrated assessment models. *Environ. Res. Lett.* **15**, 124024 (2020).
72. Wilson, C. *et al.* Evaluating process-based integrated assessment models of climate change mitigation. *Clim. Change* **166**, 3 (2021).
73. Hou, E. *et al.* Global meta-analysis shows pervasive phosphorus limitation of aboveground plant production in natural terrestrial ecosystems. *Nat. Commun.* **11**, 637 (2020).
74. Fisher, J. B. *et al.* Nutrient limitation in rainforests and cloud forests along a 3,000-m elevation gradient in the Peruvian Andes. *Oecologia* **172**, 889–902 (2013).
75. van der Waal, C. *et al.* Scale of nutrient patchiness mediates resource partitioning between trees and grasses in a semi-arid savanna. *J. Ecol.* **99**, 1124–1133 (2011).
76. Sarmiento, G., Silva, M. P. da, Naranjo, M. E. & Pinillos, M. Nitrogen and phosphorus as limiting factors for growth and primary production in a flooded savanna in the Venezuelan Llanos. *J. Trop. Ecol.* **22**, 203–212 (2006).
77. Newbery, D. M. *et al.* Does low phosphorus supply limit seedling establishment and tree growth in groves of ectomycorrhizal trees in a central African rainforest? *New Phytol.* **156**, 297–311 (2002).
78. Homeier, J. *et al.* Tropical Andean forests are highly susceptible to nutrient inputs—rapid effects of experimental N and P addition to an Ecuadorian montane forest. *PloS One* **7**, e47128 (2012).
79. Alvarez-Clare, S., Mack, M. C. & Brooks, M. A direct test of nitrogen and phosphorus limitation to net primary productivity in a lowland tropical wet forest. *Ecology* **94**, 1540–1551 (2013).

80. Vitousek, P. M., Walker, L. R., Whiteaker, L. D. & Matson, P. A. Nutrient limitations to plant growth during primary succession in Hawaii Volcanoes National Park. *Biogeochemistry* **23**, 197–215 (1993).
81. VITOUSEK, P. M. & FARRINGTON, H. Nutrient limitation and soil development: Experimental test of a biogeochemical theory. *Biogeochemistry* **37**, 63–75 (1997).
82. Bucci, S. J. *et al.* Nutrient availability constrains the hydraulic architecture and water relations of savannah trees. *Plant Cell Environ.* **29**, 2153–2167 (2006).
83. Harrington, R. A., Fownes, J. H. & Vitousek, P. M. Production and Resource Use Efficiencies in N- and P-Limited Tropical Forests: A Comparison of Responses to Long-term Fertilization. *Ecosystems* **4**, 646–657 (2001).
84. Tanner, E. V. J., Kapos, V., Freskos, S., Healey, J. R. & Theobald, A. M. Nitrogen and Phosphorus Fertilization of Jamaican Montane Forest Trees. *J. Trop. Ecol.* **6**, 231–238 (1990).
85. Simms, E. L. The effect of nitrogen and phosphorus addition on the growth, reproduction, and nutrient dynamics of two ericaceous shrubs. *Oecologia* **71**, 541–547 (1987).
86. Turrion, M. B., Gallardo, J. F. & Gonzalez, M. I. Distribution of P Forms in Natural and Fertilized Forest Soils of the Central Western Spain: Plant Response to Superphosphate Fertilization. *Arid Soil Res. Rehabil.* **14**, 159–173 (2000).
87. Raich, J. W., Russell, A. E., Crews, T. E., Farrington, H. & Vitousek, P. M. Both nitrogen and phosphorus limit plant production on young Hawaiian lava flows. *Biogeochemistry* **32**, 1–14 (1996).
88. Bi, B. *et al.* Enhanced rock weathering increased soil phosphorus availability and altered root phosphorus-acquisition strategies. *Glob. Change Biol.* **30**, e17310 (2024).

Response to reviewers

Manuscript ID: NCOMMS-23-32794-T

Title: *Leveraging ecosystems responses to enhanced rock weathering in mitigation scenarios*

Journal: *Nature Communications*

In the following boxes, **green text** corresponds to the changes made in the manuscript, and black text was already in the original manuscript. We reproduced the changes made in the supplements when they directly addressed a reviewer's comment.

Reviewer #3 :

General Comments

Comment #1

In the new manuscript, the authors have satisfactorily answered the questions I raised and I think the paper is suitable for publication.

Response #1

We thank again reviewer#3 for the review of our work.

Reviewer #4:

General Comments

Comment #1

Note: This is not a de novo review of the paper. The editor requested an assessment of how well the authors had responded to the suggestions in the first round reviewer reports.

This study quantifies the global potential for the combination of geochemical CDR as enhanced rock weathering of basalt and biotic CDR (increased biomass production) using a coupled system of models: the global energy transition model, the aggregated carbon cycle, atmospheric chemistry, and climate model, and the ORCHIDEE-CNP land-surface model. The analysis covers a significant set of scenarios including four different climate targets and three CDR portfolios and includes an uncertainty analysis of ten parameters, many of which were used to quantify enhanced weathering. Most notably, the study suggests that potential for biotic CDR from the application of basalt to forests could be a significant source of CDR under all four climate targets. The revised manuscript adequately responds to most of the reviewers' comments. For example, the authors made significant additions in response to reviewer one's first and second comments that enhanced the quality of the manuscript.

Response #1

We thank reviewer#4 for taking over the review of the paper and appreciating that we have responded adequately to the first reviewers.

Comment #2

One remaining concern is that comments five and three, from reviewers two and three respectively, about the justification of an aggregated global analysis versus a regional analysis could be more fully addressed in the revised manuscript. It is likely that the global potential presented by this analysis overestimates the reasonably achievable deployment of EW CDR when additional regional factors are considered. It is recommended that the authors consider addressing more directly some of the constraints that may arise at the local scale that could reduce the aggregate potential for EW. These constraints can arise from the potential health and ecosystem services concerns (briefly mentioned in the current draft on lines 326 et seq. - maybe the authors would want to cite Choi et al 2021 here), but also from institutional frictions in shifting forest management practices. We are not suggesting that modeling these things are within the scope of the current paper but that the paper acknowledges the magnitude of the potential affect that these concerns may have in constraining the potential for forest EW. What does the modeling effort in this paper suggest about specific issues that need to be addressed to understand how region and locality specific constraints may limit the potential for EW? Given the potential importance of these factors, some additional comment on this point is warranted.

One relevant paper the authors do cite (although for a different purpose) is:

Fuhrman, J. et al. Diverse carbon dioxide removal approaches could reduce impacts on the energy–water–land system. *Nat. Clim. Chang.* 13, 341–350 (2023)

This paper, and others mentioned in the references below are examples of modeling exercises exploring regional issues in CDR deployment. It is not that these papers need to be cited, but that the implications of regional analysis on aggregate potential be fairly (if briefly) explored so as not to give an overly optimistic assessment of actual potential for forest EW. This has implications for the marginal cost estimates given in the “Enhanced weathering deployment” subsection.

References:

Environmental Risks in Atmospheric CO₂ Removal Using Enhanced Rock Weathering Are Overlooked
WJ Choi, HJ Park, Y Cai, SX Chang. *Environmental Science & Technology* 55 (14), 9627-9629

Some regionalized modeling of CDR include:

Fuhrman et al. (2023) already discussed.

Regional implications of carbon dioxide removal in meeting net zero targets for the United States
Chloé Fauvel et al 2023 *Environ. Res. Lett.* 18 094019

Response #2

We thank Reviewer#4 for his detailed comment, and we agree with his interpretation of the results: our aggregated modelling exercise does not reflect the full range of local constraints. It ties in with the more general question of the realism of least-cost mitigation scenarios, which are identified among the space of technically possible scenarios. Yet, such space includes but is not equal to the space of politically, socio-economically or institutionally possible scenarios. Thus, beyond the limitations of the model, this approach is inherently limited to estimating the physical and technical potential of a given CDR method, which is higher than the plausible deployment level when other constraints are considered. Representing institutional constraints at the global, centennial scale is out of scope for this kind of model, although there have been some recent explorations in this direction at a supra-national level of detail¹. The two modelling studies^{2,3} quoted by Reviewer#4 (as well as ref.⁴) use the model GCAM to assess regional deployment of CDR technologies. However, the main regional constraints included in the model concern DACCS (regional geologic carbon storage capacity) and BECCS (arable land availability and geologic carbon storage capacity), and enhanced weathering is only constrained by available application area and application cost, as in our model. A typical resource constraint could be the availability of good-quality basalt feedstock, but basalt has not been primarily exploited for enhanced weathering so far, and thus the most suitable basalt resources are still unknown, whereas basalt in general is a widely abundant material. In our case, large scale modelling studies will need to be complemented by more granular analyses, not only of biological and geochemical processes, but also of the institutional, socio-economic and political contexts that may favour or hinder the adoption of new agricultural and forest management practices.

Based on the existing literature, we can already identify certain factors that may illuminate the gap between modelling and local reality, as well as the expected constraints on the deployment of these new forest and agricultural management methods. These are of different nature:

1/ There are important remaining biophysical uncertainties regarding the magnitude of the response of forest ecosystems to basalt applications. The number of field studies is expanding rapidly, but experimental results remain scarce, which is why we need to rely on the land-surface model predictions. The geochemical CDR is also uncertain as its efficiency can be reduced by different mechanisms, including secondary minerals formation instead of alkalinity exports⁵.

2/ The health and environmental effects could justify local opposition to EW deployment. We discuss the health and environmental effects, but we disregarded some of the effects discussed in Choi et al, 2021⁶ (non peer-reviewed), because they concern industrial slags which is a feedstock that we are not considering. Furthermore, non-modelled local ecosystems vulnerability to basalt application or soil pH change (ref), which should prevent basalt application, must be investigated with small scale experiments.

3/ The microeconomics of EW application could also be refined for smaller-scale study, depending on the local context. For instance, MRV costs were not considered in our assessment, nor the incentive framework that could promote the deployment of EW applications without creating perverse incentives. Incorporating MRV costs is likely to increase the marginal costs of EW, in particular for the permanent geochemical CDR⁷. However, how MRV costs will propagate into EW costs is unknown as it will depend on the incentive framework.

4/ At the local scale, behavioral, economical and institutional barriers from changing forest management practices could certainly reduce the pace and the scale of EW deployment, similarly with other land-based CDR practices⁸. Basalt application in forests could be incorporated in future schemes of carbon sequestration through forestry management (such as REDD+).

5/ The lack of local support for EW could also limit its deployment. Recent studies have shown that EW has a lower social acceptability level compared to other CDR technologies in the global south, notably because of the required increase of extractive activities^{9,10}. It is also possible that airborne basalt applications receive low social support.

We have therefore expanded the discussion of what are plausibly the most important sources of region-specific constraints to the deployment of EW in the text, but we are not able to quantify the magnitude of these constraints.

Main text, L330-341

This global, centennial-scale assessment of the technical and physical potential of enhanced weathering (EW) does not capture all the local factors that could constrain its real-world efficiency, scalability, and sustainability. Specifically, the model does not account for lower-than-expected geochemical carbon dioxide removal (CDR) resulting from incomplete basalt weathering or the formation of secondary minerals⁵, socio-cultural and institutional barriers⁸, or potential impacts of EW on human health or ecosystems^{11,12}. Monitoring, reporting, and verification (MRV) can be expected to

increase marginal costs⁷. The cost of MRV will depend on the specific technologies under development, which may vary between biotic and geochemical CDR pathways. However, how MRV costs could propagate to the marginal costs depends on the local regulatory framework. At the local scale, behavioral and institutional barriers, along with low social acceptance, may limit EW deployment as with other land-based CDR practices⁸. EW also faces low acceptability compared to other CDR methods, notably because of increased extractive activities^{9,10}. For instance, the needed basalt extraction in the 1.5°C with high overshoot case reaches 46 Gt/year (**Fig. S1**), which is half of the current global material footprint and ten times the global cement production today, potentially having a large ecological and societal impact. The dust pollution associated with the aerial application of finely milled basalt could lead to silicosis and other respiratory diseases¹³, and must therefore be prevented, for example by mixing the dust with water to form aggregates or by pelletisation¹⁴. The release of metals in basalts causing toxicity for humans must also be avoided in agricultural settings by choosing carefully the right material¹⁵. Basalt dust potential impacts on tree canopy, possibly blocking leaf stomata and reducing tree growth^{13,16} as well as potential impacts on riverine chemistry¹⁷ must also be anticipated. The application of basalt in forests could alter soil geochemistry for centuries, possibly disrupting natural systems and impacting the composition of plant communities¹⁸. Wisely exploited, these geochemistry side-effects could increase the potential for biotic CDR in addition to phosphorus fertilisation, as observed in an acid-rain impacted forest where the release of calcium through weathering of added silicate led to a biotic CDR of 3.2-3.5 tCO₂ per ton of wollastonite applied¹⁴. Ultimately, biotic effects may either offset the net carbon removal in the case of soil carbon leaching to rivers¹⁹, or enhance it by increasing soil carbon sequestration^{20,21}. More experiments are therefore required to explore the side-effects of EW particularly as rock material cannot be removed from the soil after its application.

Specific Comments

Comment #3

Two small edits: On line 288, “Field or pot conditions” should probably read “Field or plot conditions”

The comma on line 38 should probably be deleted.

Signed,

William M. Shobe, Research Professor of Public Policy (Emeritus), University of Virginia

Response #3

We thank reviewer #4 for these suggestions. The “pot conditions” refer to pot experiments. The comma on line 38 has been deleted.

Reviewer #5:

Reviewer #6:

General Comments

Comment #1

The manuscript analysed the role of EW under cost-effective mitigation pathways, by including the CDR potential of basalt applications from silicate weathering (geochemical CDR) and enhanced ecosystem growth and carbon storage in response to phosphorus released by basalt (biotic CDR). Using an integrated carbon cycle, climate and energy system model, the authors assessed the potential of considering enhanced weathering applications for achieving the Paris Agreement targets. The study showed that the application of basalt to forests could triple the level of carbon sequestration induced by EW compared to an application restricted to croplands. I reviewed the comments and suggestions from previous reviewers and the authors' responses. I agree with most of the comments. The authors have addressed some of the comments. Furthermore, I have some other concerns as follows:

Response #1

We thank reviewer #6 for reviewing our work and for providing further suggestions.

Specific Comments

Comment #2

The research involves coupling several complex models in very different ways. As previous reviewers mentioned, ensuring the reliability of the simulations from your coupled model is critical. In this revision, the authors have paid significant attention to the uncertainties of the model. Using Morris methods, they have discussed the uncertainties of key model parameters and calibrated the weathering rates. Beyond these efforts, if possible, I suggest evaluating their assessments based on the coupled model by comparing them to existing enhanced weathering application studies.

Response #2

We thank reviewer #6 for the suggestion. Existing global-scale studies on EW application have so far been limited to the case of cropland application^{3,22-24}. We find similar results for the case of cropland application, both regarding the application costs and the maximum annual EW CDR. Ref^{22,23}, which were used to parameterize our cropland application module, reports a global annual potential of 4 GtCO₂/year, limited by suitable croplands area, at a cost of 200\$/tCO₂. Ref²⁴ estimated a median technical potential of 3.2GtCO₂/year (uncertainty range of 2-5 GtCO₂/year) under 2°C scenario, considering application only in China, India, United States, Canada, Indonesia, Mexico, Brazil, France, Germany, Italy, Spain and Poland, and reports country-specific median costs ranging from 60 to 420\$/tCO₂ depending on the

country. Ref⁸ have produced cost-effective mitigation scenarios involving a large CDR portfolio that includes EW, and report an annual EW CDR of 4GtCO₂/year, consistently with Ref^{22,23}.

The only existing assessment of EW application over forests¹³ did not include the biotic CDR, and reported application cost range over forests close to the one we found.

We have made these points clearer in the discussion and the supplementary information

Main text, L310-313

We showed that the CDR potentials of EW under cost-effective mitigation pathways can be larger than previously thought by additionally considering the potentials associated with the phosphorus fertilisation, or ‘biotic’ effect of EW, while our results align with existing studies on application costs and geochemical removal potential over croplands^{3,22-24}.

S.I. 1.3.1

Adding the energy and non-energy costs, the total costs of EW on croplands range from \$43 to \$132 per ton, and the cost of the first ton applied ranges from \$43 to \$58. The total costs of enhanced weathering on forests range from \$146 to \$364 per ton, in line with a previous assessment (162-325\$/t, see ref¹³), and the share of energy costs within total costs is primarily driven by the carbon price (**Figure S1.3.2**), besides exogenous parameters variation.

Our results align with the few existing global-scale studies on EW application, which have so far been limited to the case of cropland application^{3,22-24}. Ref^{22,23} reports a global annual potential of 4 GtCO₂/year, limited by suitable croplands area, at a cost of 200\$/tCO₂. Notably, the EW module in ref⁸ as well as our cropland application module have been designed following ref^{22,23}, and we obtain similar CDR magnitude and application costs. Ref²⁴ estimated a median technical potential of 3.2 GtCO₂/year (uncertainty range of 2-5 GtCO₂/year) under 2°C scenario, considering application only in China, India, United States, Canada, Indonesia, Mexico, Brazil, France, Germany, Italy, Spain and Poland, and reports country-specific median costs ranging from 60 to 420\$/tCO₂ depending on the country. This larger cost range reflects their bottom-up approach, which does not assume a globally uniform marginal carbon price.

We have also compared the assumptions of our model with existing experimental studies, listed in the SI 1.1.1.

Comment #3

I suggest the author reorganize an easy-read Introduction section by moving sentences/review for methods to the method sections, highlighting importance of EW application at the begin, and stating your objectives and mainly methods used at the last of introduction. The paper is very multidisciplinary research, and thus this is important for readers from a broad field.

Response #3

Following the suggestion of reviewer#6, we have moved some sentences from the last paragraphs of the introduction to the method section for more clarity.

Here are the new last paragraphs of the introduction:

Main text, L79-101

We developed a new version of the partial-equilibrium energy model GET7.1^{25,26}, which we integrated with the aggregated carbon cycle, atmospheric chemistry and climate model (ACC2^{27,28}). The carbon cycle component of the resulting GET-ACC2 model was coupled to a newly created EW module, in which the dissolution of basalt directly removes atmospheric CO₂ and delivers phosphorus to the soil, fertilizing forest growth and stimulating additional CO₂ sequestration in forest biomass (Fig. 1). We emulate ORCHIDEE-CNP²⁹, a global biosphere model that resolves the phosphorus cycle, to quantify both geochemical (abiotic) and biotic carbon dioxide removal (CDR) from basalt applied to forest ecosystems. GET-ACC2 is thus the first forward-looking model accounting for this biotic CDR pathway of EW. GET-ACC2 quantifies least-cost pathways where low-carbon technologies, CDR, and abatement measures for CH₄ and N₂O are deployed to mitigate climate change. The net present value of the social surplus (i.e. the sum of consumers surplus minus the energy costs, discounted at a 5% rate) is maximised with perfect foresight, leading to a preference for late spending, including late abatement.

The spatial heterogeneity of the response of ecosystems to basalt application and the local factors driving additional biological carbon sequestration have been discussed in Goll et al. (2021)³⁰. The biotic CDR was found to be highly variable across regions, strongly dependent on ecosystem type, and most effective where the natural background phosphorus availability was insufficient for plants to benefit from increasing atmospheric CO₂ and warming, notably in tropical and boreal forests³⁰. It should however be noted that this spatial heterogeneity is only implicitly represented in the emulator (see Methods), and that we only accounted for the phosphorus fertilisation effect on plants, not other effects of basalt weathering on soil microbes and soil biota, or other biotic effects such as interactions with the nitrogen cycle, plant health and resistance to pathogens²¹.

Comment #4

The authors said "application of basalt to forests could triple the level of carbon sequestration induced by EW compared to an application restricted to croplands". Do you mean for all forest types? From the perspective of biogeography and climate, different forest types (or forests with much different hydrologic

and climate conditions) exhibit distinct different biogeochemical effects. I did not see how did you consider this important problem.

Response #4

We mean that enabling basalt application to forests increases the CDR. It does not mean that basalt is applied to all forests. Likewise, basalt application to croplands does not mean that basalt is applied to all croplands. We have considered the issue of forest heterogeneity at the calibration step, which is described both in the main text and in the supplements. We have further emphasized this point in the text.

Main text, L92-96

The spatial heterogeneity of the response of ecosystems to basalt application and the local factors driving additional biological carbon sequestration have been discussed in Goll et al. (2021)³⁰. The biotic CDR was found to be highly variable across regions, strongly dependent on ecosystem type, and most effective where the natural background phosphorus availability was insufficient for plants to benefit from increasing atmospheric CO₂ and warming, notably in tropical and boreal forests³⁰. **GET-ACC2 thus prioritizes the most responsive areas for basalt application**, but this spatial heterogeneity is only implicitly captured in the emulator (see Methods).

Main text, L543-551

In the spatially explicit land surface model ORCHIDEE-CNP, the increase of NPP due to phosphorus release depends on the soil, biome and climate and saturates with increasing basalt additions. Application pixels are ranked according to their NPP stimulation from high to low, and grouped in M land response classes of areas a_i . In the current setting, M=5 (more details on classes in the SM). A function of the rock application rate $c_{B,i}$ [Gt.Mkm⁻²] is used to fit the mean NPP response in each class i during the forty years that follow basalt application, $\delta\bar{NPP}_i$ (E.5). These classes are an implicit representation of the spatial heterogeneity of the response of forest ecosystems to phosphorus addition.

$$\delta\bar{NPP}_i = \delta NPP_{i,max} \left(1 - e^{-\alpha_i c_{B,i}}\right) \quad (E.5)$$

We have also modified a sentence of the introduction

Main text L52-53

Here we explore how the application of EW on **suitable** forests and crop fields could affect mitigation pathways [...]

Comment #5

In addition to the high costs, large-scale applications of EW for natural ecosystems may induce ecological risk. Discussing where is suitable for application EW is important.

Response #5

We thank reviewer#6 for their suggestion. In addition to the discussion of possible risks to natural ecosystems, we have emphasized the importance of applying basalt in the most suitable areas with regards to side-effects. However, the 'ecologically safe' spatial distribution of the EW application is not within the scope of our model, and we have no additional results to contribute to the discussion.

Main text, L350-359

The application of basalt in forests could alter soil geochemistry for centuries, possibly disrupting natural systems and impacting organisms among all trophic levels¹⁸. However, wisely exploited, these geochemistry side-effects could increase the potential for biotic CDR in addition to phosphorus fertilisation, as observed in an acid-rain impacted forest where the release of calcium through weathering of added silicate led to a biotic CDR of 3.2-3.5 tCO₂ per ton of wollastonite applied¹⁴. Ultimately, biotic effects may either offset the net carbon removal in the case of soil carbon leaching to rivers¹⁹, or enhance it by increasing soil carbon sequestration^{20,21}. More experiments are therefore required to explore the side-effects of enhanced weathering, and to determine the most suitable areas for basalt application particularly as rock material cannot be removed from the soil after its application.

Comment #6

Please revise the entire text according to the journal's formatting requirements, such as using abbreviations and full names for the journal titles and figures (e.g., "Fig." for figures in abbreviation and "Figure" in full).

Response #6

We have proceeded as recommended by the reviewer.

Comment #7

In the results section, some figures do not fully demonstrate the data due to the attempt to maintain uniform coordinate axes, while others concentrate within a narrow range, making it difficult to discern trends, as exemplified in Figure 4g. Consider adjusting these figures to better illustrate the differences between the data.

Response #7

We have done as suggested.

Main text, figure 4

Comment #8

The background section lacks detailed knowledge on the application of the model used in this paper to the study of rock weathering carbon sequestration. The suitability of using this model is not apparent from the context provided. In addition, what are the advantages of the applied model compared with other models?

Response #8

Our model is so far the only one that integrates the biotic effect of the phosphorus released by basalt application within a fully coupled climate-energy-economy model, which allows us to explore the main relevant feedbacks of EW on climate mitigation strategies: carbon-cycle feedbacks, energy demand feedbacks, substitution with other mitigation technologies, delay of the early emission reductions in overshoot cases, and mitigation costs reductions. We have made this point clearer in the introduction.

Main text, L79-91

We developed a new version of the partial-equilibrium energy model GET7.1^{25,26}, which we integrated with the aggregated carbon cycle, atmospheric chemistry and climate model (ACC2^{27,28}). The carbon cycle component of the resulting GET-ACC2 model was coupled to a newly created EW module (**Fig. 1**), in which the dissolution of basalt directly removes atmospheric CO₂ and delivers phosphorus to the soil, stimulating CO₂ sequestration in forest biomass. GET-ACC2 emulates ORCHIDEE-CNP²⁹, a global biosphere model that resolves the phosphorus cycle, to quantify both geochemical (abiotic) and biotic carbon dioxide removal (CDR) from basalt applied to forest ecosystems. GET-ACC2 is thus the first forward-looking model accounting for this biotic CDR pathway of EW. GET-ACC2 quantifies least-cost pathways where low-carbon technologies, CDR, and abatement measures for CH₄ and N₂O are deployed to mitigate climate change. The net present value of the social surplus (i.e. the sum of consumers surplus minus the energy costs, discounted at a 5% rate) is maximised with perfect foresight, leading to a preference for late spending, including late abatement.

Comment #9

Why choose basalt for rock weathering research, and what impact does it have on climate change?

Response #9

Basalt is chosen because it is a very abundant feedstock, which has less negative side-effects than other feedstocks such as olivine as it contains much less heavy metals. We have completed the introduction.

Main text, L36-42

Enhanced weathering of basalt (EW) is an emerging and promising CDR that consists in amending soils with basalt dust^{24,31-33}. Basalt is an abundant volcanic rock containing less harmful trace elements than alternative rocks considered for EW³⁴. As basalt erodes, the minerals released react with CO₂ and sequester carbon for at least several hundred years³⁵, a process called ‘geochemical CDR’. Current research focuses on basalt for demonstration purposes of EW. Basalt encompasses a wide range of rock material with varying CDR potential and other feedstocks might be more suited.

Comment #10

The results section of this paper cites numerous similar studies. What distinguishes this section from the discussion? It is recommended that the results section solely present the findings of this paper, leaving in-depth exploration and discussion of the relevant results to the discussion section.

Response #10

The results section presents our findings on EW potential and costs and includes a quantitative comparison with existing studies, showing alignment with previous research. This approach provides context for our results within the broader literature. The discussion section, by contrast, focuses on interpreting the significance of our findings, considering their broader implications, and addressing the limitations and potential blind spots of the study.

Comment #11

What is the conclusion of this paper and does it conform to the hypothesis? What are the implications of this study's findings for climate change mitigation.

Response #11

The conclusion of the paper can be found in the discussion section. In particular:

Main text, L312-329

We showed that the CDR potentials of EW under cost-effective mitigation pathways can be larger than previously thought by additionally considering the potentials associated with the phosphorus fertilisation, or ‘biotic’ effect of EW, **while our results align with existing studies on application costs and geochemical removal potential over croplands** ^{3,22-24}. EW neither accelerates climate change mitigation nor reduces temperature overshoot in our cost-effectiveness analysis, yet its potential for lowering peak temperatures to mitigate near-term climate damage could be further assessed elsewhere through cost-benefit analyses. Deploying EW in addition to BECCS reduces the willingness to pay for biomass and could thereby lower the pressure on land conversion as well as on food prices, although the reliance on bioenergy remains significant.

We further demonstrated that under mitigation pathways, in particular, for the 1.5°C warming target, the use of EW reduces the total mitigation cost, lowers the peak carbon price, and replaces a larger amount of BECCS when the ‘biotic’ effect is included, even if we account for the high costs for EW application over forest areas by aeroplanes. These findings are robust under a range of uncertainties considered, unless weathering rates are in the 0-1%/year range. Such benefits of EW were found to be more pronounced under pathways with high temperature overshoot than those with medium or no overshoot. Nevertheless, in high overshoot pathways, EW is used to compensate for higher emissions for the upcoming decades, which is a risky strategy, given the increased likelihood of climate disasters at high overshoot levels.

and

Main text, L358-366

At face value, a life cycle analysis comparing EW with other mitigation technologies showed that EW has the advantage to use less land than BECCS or afforestation, less energy than for direct air capture, and less water than for those three technologies³⁶. The application of basalt in forests is therefore a promising method for mitigating climate change, but it requires the deployment of an appropriate regulatory framework, to ensure that EW helps ecosystems sequester more carbon while minimising adverse side-effects. Furthermore, even if we explored uncertainties as comprehensively as possible, the true uncertainties cannot be wholly captured inherently and certain classes of uncertainties cannot be assessed via quantitative means, indicating a need for careful interpretation and dissemination of our results for stakeholders.

References

1. Bertram, C. *et al.* Feasibility of peak temperature targets in light of institutional constraints. *Nat. Clim. Change* **14**, 954–960 (2024).
2. Fauvel, C. *et al.* Regional implications of carbon dioxide removal in meeting net zero targets for the United States. *Environ. Res. Lett.* **18**, 094019 (2023).
3. Fuhrman, J. *et al.* Diverse carbon dioxide removal approaches could reduce impacts on the energy–water–land system. *Nat. Clim. Change* **13**, 341–350 (2023).
4. Javadi, P. *et al.* The impact of regional resources and technology availability on carbon dioxide removal potential in the United States. *Environ. Res. Energy* **1**, 045007 (2024).
5. Niron, H., Vienne, A., Frings, P., Poetra, R. & Vicca, S. Exploring the synergy of enhanced weathering and *Bacillus subtilis*: A promising strategy for sustainable agriculture. *Glob. Change Biol.* **30**, e17511 (2024).

6. Choi, W.-J., Park, H.-J., Cai, Y. & Chang, S. X. Environmental Risks in Atmospheric CO₂ Removal Using Enhanced Rock Weathering Are Overlooked. *Environ. Sci. Technol.* **55**, 9627–9629 (2021).
7. Mercer, L., Burke, J. & Rodway-Dyer, S. Towards improved cost estimates for monitoring, reporting and verification of carbon dioxide removal.
8. Perkins, O. *et al.* Toward quantification of the feasible potential of land-based carbon dioxide removal. *One Earth* **6**, 1638–1651 (2023).
9. Baum, C. M., Fritz, L., Low, S. & Sovacool, B. K. Public perceptions and support of climate intervention technologies across the Global North and Global South. *Nat. Commun.* **15**, 2060 (2024).
10. Low, S., Fritz, L., Baum, C. M. & Sovacool, B. K. Public perceptions on carbon removal from focus groups in 22 countries. *Nat. Commun.* **15**, 3453 (2024).
11. Buckingham, F. L., Henderson, G. M., Holdship, P. & Renforth, P. Soil core study indicates limited CO₂ removal by enhanced weathering in dry croplands in the UK. *Appl. Geochem.* **147**, 105482 (2022).
12. Dupla, X., Möller, B., Baveye, P. C. & Grand, S. Potential accumulation of toxic trace elements in soils during enhanced rock weathering. *Eur. J. Soil Sci.* **74**, e13343 (2023).
13. Taylor, L. L. *et al.* Enhanced weathering strategies for stabilizing climate and averting ocean acidification. *Nat. Clim. Change* **6**, 402–406 (2016).
14. Taylor, L. L. *et al.* Increased carbon capture by a silicate-treated forested watershed affected by acid deposition. *Biogeosciences* **18**, 169–188 (2021).
15. Cobo, S. *et al.* Sustainable scale-up of negative emissions technologies and practices: where to focus. *Environ. Res. Lett.* **18**, 023001 (2023).
16. Farmer, A. M. The effects of dust on vegetation—a review. *Environ. Pollut.* **79**, 63–75 (1993).
17. Zhang, S. *et al.* River chemistry constraints on the carbon capture potential of surficial enhanced rock weathering. *Limnol. Oceanogr.* **67**, S148–S157 (2022).
18. Vandeginste, V., Lim, C. & Ji, Y. Exploratory Review on Environmental Aspects of Enhanced

- Weathering as a Carbon Dioxide Removal Method. *Minerals* **14**, 75 (2024).
19. Klemme, A., Rixen, T., Müller, M., Notholt, J. & Warneke, T. Destabilization of carbon in tropical peatlands by enhanced weathering. *Commun. Earth Environ.* **3**, 1–9 (2022).
 20. Buss, W., Hasemer, H., Ferguson, S. & Borevitz, J. Stabilisation of soil organic matter with rock dust partially counteracted by plants. *Glob. Change Biol.* **30**, e17052 (2024).
 21. Vicca, S. *et al.* Is the climate change mitigation effect of enhanced silicate weathering governed by biological processes? *Glob. Change Biol.* **28**, 711–726 (2022).
 22. Strefler, J. *et al.* Carbon dioxide removal technologies are not born equal. *Environ. Res. Lett.* **16**, 074021 (2021).
 23. Strefler, J., Amann, T., Bauer, N., Kriegler, E. & Hartmann, J. Potential and costs of carbon dioxide removal by enhanced weathering of rocks. *Environ. Res. Lett.* **13**, 034010 (2018).
 24. Beerling, D. J. *et al.* Potential for large-scale CO₂ removal via enhanced rock weathering with croplands. *Nature* **583**, 242–248 (2020).
 25. Azar, C., Johansson, D. J. A. & Mattsson, N. Meeting global temperature targets—the role of bioenergy with carbon capture and storage. *Environ. Res. Lett.* **8**, 034004 (2013).
 26. Johansson, D. J. A., Azar, C., Lehtveer, M. & Peters, G. P. The role of negative carbon emissions in reaching the Paris climate targets: The impact of target formulation in integrated assessment models. *Environ. Res. Lett.* **15**, 124024 (2020).
 27. Tanaka, K. *et al.* Aggregated Carbon cycle, atmospheric chemistry and climate model (ACC2): description of forward and inverse mode. 14069106 (2007) doi:10.17617/2.994422.
 28. Tanaka, K. & O'Neill, B. C. The Paris Agreement zero-emissions goal is not always consistent with the 1.5 °C and 2 °C temperature targets. *Nat. Clim. Change* **8**, 319–324 (2018).
 29. Goll, D. S. *et al.* A representation of the phosphorus cycle for ORCHIDEE (revision 4520). *Geosci. Model Dev.* **10**, 3745–3770 (2017).
 30. Goll, D. S. *et al.* Potential CO₂ removal from enhanced weathering by ecosystem responses to

- powdered rock. *Nat. Geosci.* **14**, 545–549 (2021).
31. Renforth, P. The potential of enhanced weathering in the UK. *Int. J. Greenh. Gas Control* **10**, 229–243 (2012).
 32. Hartmann, J. *et al.* Enhanced chemical weathering as a geoengineering strategy to reduce atmospheric carbon dioxide, supply nutrients, and mitigate ocean acidification. *Rev. Geophys.* **51**, 113–149 (2013).
 33. Moosdorf, N., Renforth, P. & Hartmann, J. Carbon Dioxide Efficiency of Terrestrial Enhanced Weathering. *Environ. Sci. Technol.* **48**, 4809–4816 (2014).
 34. Beerling, D. J. *et al.* Farming with crops and rocks to address global climate, food and soil security. *Nat. Plants* **4**, 138–147 (2018).
 35. Köhler, P. Anthropogenic CO₂ of High Emission Scenario Compensated After 3500 Years of Ocean Alkalinization With an Annually Constant Dissolution of 5 Pg of Olivine. *Front. Clim.* **2**, (2020).
 36. Eufrazio, R. M. *et al.* Environmental and health impacts of atmospheric CO₂ removal by enhanced rock weathering depend on nations' energy mix. *Commun. Earth Environ.* **3**, 106 (2022).

Response to reviewer

Please find below the response to the last reviewer's comment.

Reviewer's comment

Enhanced rock weathering (EW) can consume atmospheric carbon dioxide. The authors analysed the role of EW under cost-effective mitigation pathways, 11 by including the CDR potential of basalt applications from silicate weathering and 12 enhanced ecosystem growth and carbon storage in response to phosphorus released by basalt. However, the side effects of basalt applications on croplands should also be fully considered. Especially, the risks of accumulation of heavy metals and large scale basalt mining should be fully considered.

Response

We have made the following additions to the text (added parts in blue):

1/ regarding mining impacts:

L306: This scale of mining could drive deforestation, disrupt ecosystems, and pose significant ecological and societal challenges¹.

2/ regarding heavy metals:

L310: The release of metals in basalts causing toxicity for humans must also be avoided in agricultural settings by choosing carefully the right material, and long-term studies on metal bioavailability and accumulation in soils, crops, and water systems are needed to assess potential health risks and inform regulatory guidelines.

Relevant reference

1. Edwards, D. P. et al. Climate change mitigation: potential benefits and pitfalls of enhanced rock weathering in tropical agriculture. *Biology Letters* 13, 20160715 (2017).

General assessment

The study develops a techno-economic module of Enhanced Rock Weathering applied to crops and natural lands, which is integrated into a coupled energy-climate model to assess the importance for achieving Paris-compatible climate targets focusing on emissions, energy use and emissions & removals. With respect to the later not only the removal due to abiotic weathering is represented, but also the enhanced soil quality due to phosphor fertilization. The results suggest that EW on natural can contribute a substantial share, particularly, in case of a 1.5°C target with a large overshoot. The authors also add an uncertainty analysis.

The study, thus, covers a broad range of disciplines and different methods. As a reviewer, I ask myself whether it would have been better to split the overall study into two, separating the techno-economic and geological assessment from the assessment of EW in a broader energy-climate modeling framework. The advantage of a single contribution is that the reader is provided with a full package, the disadvantage is that it always seems a bit premature (which on the other hand also indicates novelty). I think the authors should seriously think about a rebalancing. I recommend them to put more emphasis on the uncertainties and highlight the exploratory nature of the study. It is a scientific value added to clearly highlight the uncertainties and discuss them, if a novel climate change mitigation option is considered to be deployed at large scale.

The authors put much emphasis on the cross-sectoral effects regarding the deployment of bioenergy with CCS and also the food prices (that are not explicitly modeled, though). The authors have put these cross effects into the abstract. I do not think that this is the major point to be discussed at that stage of the scientific discussion. It would be much more important to discuss for example the regional heterogeneity and what factors are crucial for the weathering rates and what can be said about them. Most of this has been moved to the technical appendix, but it should be presented, analyzed and discussed in the main manuscript.

An important information derived from the modeling would be, for example, the CO₂ prices at which the EW on natural lands or crops land turn competitive. A model is a useful tool for this information because the costs depend on various system wide parameters that are not independent from the climate policy, such as the carbon intensity of the energy carriers used for the EW activity. A partial analysis that sets these parameters constant and only varies them independently, misses the point of the transformative nature of a global mitigation strategy and the resulting transformation pathway. The information is available, but has not been used to the degree that it should be used. It would be interesting to understand what the determinants (or influencing factors) of these break-even CO₂ prices are.

Also, the reader is very much interested into the local side effects of adding the basalt rock powder to natural lands (such as forests). What are the positive side effects and also what might be risks. In my opinion this is at this stage, more important than the side effects on food prices via bioenergy markets.

Main points

The notion that GET is a perfect foresight model is not sufficient (page 3). It must be clarified, which carbon fluxes are endogenously considered and incentivized in the policy framework. This is important because of the indirect effect of biotic carbon removal caused by the phosphor fertilization. For the deployment of the option it is important whether the indirect effect is accounted and the corresponding revenue flow is fully internalized into the calculation that balances revenues to costs. In case the indirect is internalized it is also important to understand whether this

indirect effect is considered via an average value or whether it is specific to individual pieces of land that are heterogenous. The authors are required to clarify these points and seriously consider to undertake an uncertainty analysis.

It is not clear to me what the regional resolution of the GET model is. The supplement only says that there is a multi-region version available, but it does not say whether this version has been used. For the analysis at hand a multi-regional model would be the appropriate tool due to the geographic heterogeneity and locational specificity of the EW deployment.

Is the bioenergy supply related to GHG emissions? A short note and a reference to the literature would be very useful. In case the authors assume full carbon neutrality the authors are requested to justify this assumption and, if need be, to adjust the model assumption.

Why is the CO₂ price decreasing after 2090 with a remarkable peak in the cases with high overshoot (Figure 3)? The time profile of the prices is also reflected in the emissions trajectory (Figure 2) with the decreasing emissions. These changes are remarkable and seem not to follow a specific argument, but are due to numerical issues.

How is the rock powder spread over the forest areas? The authors state that the phosphor effect saturates (page 4). This suggests that the rock powder is spread evenly across the forest areas, rather than square km by square km. The difference is that the logistic effort varies between both approaches. Obviously, the even spreading of the rock powder would be better to benefit from the phosphor stimulation of NPP, however, it implicitly assumes a much larger logistical effort that is not represented. At least this is how I understand the implementation, the results and the conclusions drawn on that basis. The SI material is not very useful to understand how the spreading is implemented into the GET model and how it relates to the Orchidee model. The authors are required to clarify this point, communicate the implementation in plain language to the reader and, if need be, to adjust the implementation accordingly.

In that context, I sense the following comment is appropriately placed. I do not understand Figure S1.3.4. What is the y-axis telling me? Is land somehow sorted to some criterion? and why these uncommon patches? What does the width of these patches tell the reader? I simply do not get what this figure tries to communicate.

It is not clear to me how the climate model responds to CO₂ emissions. Investigating Figures 2 and 3, it appears to me that the global mean temperature remains constant at annual CO₂ emissions of around 10 GtCO₂. There is some variation about the emissions that are consistent with a constant global mean temperature, but 10 GtCO₂/yr seem above that level. The Supplement is not saying something about this point. The authors have to clarify and, if need be, adjust the model parameters.

In that context the following point is also important. What are the levels of emissions of non-CO₂ GHG? The supplement does not say anything about it. This is important to understand the CO₂ emissions and the temperature change. This information should be reported in the Supplement.

The overshoot qualified as “low” in the 1.5°C case is not small given the IPCC classification. The C1 category allow for a 0.1°C overshoot, while there it is twice that temperature difference. The authors may want to revisit the scenario definition.

The reaction speed of weathering very much depends on the grain size, which in turn has a strong effect on the electricity required for the grinding. This has been shown in the paper by Strefler et al. (2018, Figure 1). The authors should depict the assumptions against this backdrop so that the reader

better understands how the assumptions compare. Providing a range of 0.06-0.6EJ/Gt is too little information (Sec 1.1 in Supplement).

The weathering process involves the reaction of CO₂ dissolved in rain water. Thus, precipitation is an important natural parameter for the speed of weathering. In forest areas it is also important how much rain water gets to the soil where the rock powder will be located. In relatively mature forests with frequent, but small rain falls, the share of rain water that hits the ground will be relatively small. I did not find that the authors have represented this point in their parameterization. The authors are required to clarify.

What are the variable costs of spreading by airplane? The authors only mention the maintenance costs of 5% of the initial investment costs. However, an air plane needs a start and landing base, a ground crew and a pilot. Are these costs accounted? What about the wages for truck drivers? (This refers to the techno-economic parameters in the Supplement)

The authors talk rather generally about the ranking of the CDR quantities as differences between scenarios are more or less. However, Figure 2 clearly shows that for the same long-term temperature target there is an intersection of CDR deployment over time, because the low overshoot case requires stronger-near-term CDR deployment while the high overshoot case delays CDR deployment

The strong variations in the ratio between biotic and abiotic cumulative carbon removal in figure 2 (top row) suggests that there is a certain and limited potential of biotic carbon removal that is largely independent of the amount of abiotic. It is important to understand the coupling between the biotic and abiotic carbon removal. The authors are required to investigate this somewhat more.

The uncertainty range in Figure 3 for the global mean temperature anomaly is small. This is because the ECS is not varied. However, it makes sense to assume the same uncertainties for all four scenarios.

Figure 1.3.5 in the supplement is confusing. Why are the energy costs higher than the marginal costs? How can the marginal revenue be higher than the carbon price (1.5K, low overshoot case in 2030)? What is the share representing (bottom, right hand side)? What are the ranges representing. The 25-75% is too little info. How can the solid lines be outside the ranges?

Smaller Points

Page 3: "The increase in terrestrial carbon storage is" I guess the word vegetation is missing here.